# On the Thermodynamic and Kinetic Aspects of Immersion Ice Nucleation

Donifan Barahona[1]

[1]NASA Goddard Space Flight Center, Greenbelt, MD, USA

*Correspondence to:* Donifan Barahona (donifan.o.barahona@nasa.gov)

**Abstract.**

Heterogeneous ice nucleation initiated by particles immersed within droplets is likely the main pathway of ice formation in the atmosphere. Theoretical models commonly used to describe this process assume that it mimics ice formation from the vapor, neglecting interactions unique to the liquid phase. This work introduces a new approach that accounts for such interactions by linking the ability of particles to promote ice formation to the modification of the properties of water near the particle-liquid interface. It is shown that the same mechanism that lowers the thermodynamic barrier for ice nucleation also tends to decrease the mobility of water molecules, hence the ice-liquid interfacial flux. Heterogeneous ice nucleation in the liquid phase is thus determined by the competition between thermodynamic and kinetic constraints to the formation and propagation of ice. At the limit, ice nucleation may be mediated by kinetic factors instead of the nucleation work. This new ice nucleation regime is termed spinodal ice nucleation. Comparison of predicted nucleation rates against published data suggests that some materials of atmospheric relevance may nucleate ice in this regime.

## 1 Introduction

Ice nucleation in cloud droplets and aerosol particles leads to cloud formation and glaciation at low temperature. It is often initiated by certain aerosol species known as ice nucleating particles (INP) (DeMott et al., 2003; Cziczo et al., 2013; Barahona et al., 2017). These include dust, biological particles, metals, effloresced sulfate and sea salt, organic material and soot (Murray et al., 2012; Hoose and Möhler, 2012). Background INP concentrations may be influenced by aerosol emissions (Lohmann and Feichter, 2005), altering the formation and evolution of ice clouds and leading to an indirect effect on climate. The assessment of the role of INP on climate is challenging due to the complexity of the atmospheric processes involving ice and the limited understanding of the ice nucleation mechanism of INP (Myhre et al., 2013). Ice formation promoted by a particle completely immersed within the liquid phase, referred as "immersion freezing", is likely the most common cloud glaciation pathway in the atmosphere (DeMott et al., 2003; Wiacek et al., 2010). Immersion freezing is involved in the initiation of precipitation and determines to a large extent the phase partitioning in convective clouds (Diehl and Wurzler, 2004; Wiacek et al., 2010; Lance et al., 2011; Murray et al., 2012).

The accurate representation of immersion ice nucleation is critical for the correct modeling of cloud processes in the atmosphere (Hoose and Möhler, 2012; Murray et al., 2012; Tan et al., 2016). Field campaign data have been used to develop

empirical formulations relating the INP concentration to the cloud temperature, $T$, and saturation ratio, $S_\mathrm{i}$ (e.g, Bigg, 1953; Fletcher, 1959; Meyers et al., 1992), and more recently to the ambient aerosol size and composition (e.g., DeMott et al., 2010; Niemand et al., 2012; Phillips et al., 2013). Empirical formulations provide a simple way to parameterize ice nucleation in atmospheric models (e.g., Gettelman et al., 2012; Barahona et al., 2014). However they may not be valid outside the conditions used in their development, particularly as different experimental techniques may result on a wide range of measured ice nucleation efficiencies (Hiranuma et al., 2015). Alternatively, the ice nucleation efficiency can be empirically parameterized using laboratory data, although with similar caveats (Knopf and Alpert, 2013; Murray et al., 2012).

Molecular dynamics (MD) simulations and direct kinetic methods have been used to study ice nucleation (e.g., Taylor and Hale, 1993; Matsumoto et al., 2002; Lupi et al., 2014; Espinosa et al., 2014). However, the classical nucleation theory (CNT) is nearly the only theoretical approach employed to describe immersion freezing in cloud models (e.g., Khvorostyanov and Curry, 2004; Barahona and Nenes, 2008, 2009; Hoose et al., 2010). According to CNT, nucleation is initiated by the growth of a cap-shaped ice germ on the surface of the immersed particle (Pruppacher and Klett, 1997; Kashchiev, 2000). The geometry of the ice germ is defined by a force balance at the particle-ice-liquid interface, and characterized by the contact angle, $\theta$. In this sense, the ice germ is assumed to "wet" the immersed particle in the same way a liquid droplet wets a solid surface (De Gennes, 1985). Low values of $\theta$ indicate a high affinity of the particle for ice and a low energy of formation of the ice germ.

Direct application of CNT in immersion freezing is thwarted by uncertainty in fundamental parameters of the theory, i.e., the ice-liquid interfacial energy, $\sigma_\mathrm{iw}$, and the activation energy. Moreover, using a single $\theta$ to describe the nucleation efficiency of dust and other materials typically leads to large discrepancy between CNT predictions and experimental measurements (e.g., Zobrist et al., 2007; Alpert et al., 2011; Broadley et al., 2012; Rigg et al., 2013). MD simulations show that an ice germ formed near a surface tends to have a complex geometry instead cap-shaped assumption of CNT (e.g. Lupi et al., 2014; Cox et al., 2015; Fitzner et al., 2015). Within a liquid the ice germ may not "wet" the particle but rather exert stress on the substrate (Cahn, 1980; Rusanov, 2005), and it is not clear that this can be described as a simple function of $\theta$ (Cahn, 1980). It has also been shown that $\sigma_\mathrm{iw}$ obtained by fitting CNT to measured nucleation rates tends to be biased high to account for mixing effects neglected in common formulations of the theory (Barahona, 2014).

More fundamentally, CNT neglects important interactions near the immersed particle that may influence the nucleation rate. It is assumed that ice nucleation solely depends on the local geometry of the absorbed molecules on the immersed particle (Kashchiev, 2000). This implies that the particle influences the formation of the ice germ but does not influence the adjacent water. The viscosity and density of water in the vicinity of the particle and in contact with the ice germ are assumed similar to those in the bulk (Kashchiev, 2000; Pruppacher and Klett, 1997). This is at odds with evidence of a strong effect of immersed particles on the vicinal water (Drost-Hansen, 1969; Michot et al., 2002). In fact, such an effect may be responsible for the enhancement of ice nucleation near immersed solids (Anderson, 1967).

## 1.1 Evidence for the formation of ordered structures near the liquid-particle interface

It has been known for some time that water near interfaces displays physicochemical properties different from those of the bulk (e.g., Drost-Hansen, 1969; Michot et al., 2002; Bellissent-Funel, 2002). By examining a wealth of available observations

Drost-Hansen (1969) concluded that vicinal water (i.e., the water immediately adjacent to the particle) may exist in an ordered state near the solid-liquid interface that may propagate over considerable distance, of the order of hundreds to thousands of molecular diameters. More recent experiments showing that hydrophilic surfaces have a long-range impact further support this conclusion (e.g., Zheng et al., 2006). The interaction between the particle and the vicinal water becomes more significant as

the temperature decreases and the viscosity increases (Wolfe et al., 2002). Recent studies have shown the presence of ordered water near the interface of biological (Cooke and Kuntz, 1974; Snyder et al., 2014), metallic (Michot et al., 2002) and clay (Yu et al., 2001; Rinnert et al., 2005) particles, a notion that is also supported by molecular dynamics simulations (Lupi et al., 2014; Cox et al., 2015). In a groundbreaking work, Anderson (1967) found strong evidence of ice formation several molecular diameters away from the clay-water interface. The author concluded that ice formation does not require an ice germ attached

to the substrate, but rather the nascent ice germ is stabilized by ordering in the interfacial zone. To date no quantitative theory has been developed exploiting such a view of ice nucleation.

The description of the properties of the vicinal water is still under investigation. Early studies concluded that ordered water near immersed surfaces does not resemble a caltrate-like orientation of water molecules (Drost-Hansen, 1969). Rather, in the supercooled region the presence of structured low-density regions near solid surfaces (termed "ice-like") has been reported for

different materials (e.g., Etzler, 1983; Yu et al., 2001; Michaelides and Morgenstern, 2007; Feibelman, 2010; Snyder et al., 2014). In this region Etzler (1983) parameterized the density and enthalpy of vicinal water as a mixture of ice-like and bulk-like water. Additional experimental observations show the modification of the mobility of water near interfaces, and a higher viscosity than the bulk (Warne et al., 2000; Yu et al., 2001; Wolfe et al., 2002; Wang et al., 2006). In some cases, clays and biological systems exhibit a viscous layer of water at the particle-liquid interface that remains liquid even if the bulk has

already frozen (Drost-Hansen, 1969). These effects are typically characterized as non-equilibrium, since they affect the flux of molecules to the nascent ice germ rather than the thermodynamics of ice nucleation. Li et al. (2014) found experimentally that the viscosity of interfacial water regulates the ice nucleation activity, giving support to the idea that the work of nucleation and the enhancement of the viscosity in the interfacial region are tightly linked. In fact, increased viscosity may be a necessary condition for ice nucleation since structural ordering is not possible in a fluid with low viscosity (Anderson, 1967).

These considerations are largely missing in the theoretical description of ice nucleation. There is currently no theory that can account for the thermodynamic and kinetic effects of an immersed particle on the surrounding water, hence on ice nucleation. Such a task is undertaken in this work. Section 2 presents the theoretical description of a new approach, accounting for the thermodynamics of vicinal water (Section 2.3) and their relation to the work of nucleation (Section 2.4) and the nucleation rate (Sections 2.5 and 2.6). These new relations are analyzed and applied to specific cases of atmospheric relevance in Section 3.

## 2 Theoretical development

The new approach is developed within the scope of the kinetic treatment of nucleation, when cluster formation is the limiting step to ice formation (Pruppacher and Klett, 1997; Kashchiev, 2000). The central result of this theory is the well-known general expression for the nucleation rate in steady state,(e.g, Kashchiev, 2000, Cf. Eq. 13-33),

$$J = Zf^*C^*,$$ (1)

where $Z$ is the Zeldovich factor, $f^*$ is the attachment frequency (also called the impingement factor) and $C^*$ is the concentration of supercritical clusters. $Z$ corrects for the detachment of monomers from the cluster during nucleation. It can also be interpreted as the probability that a molecule reaches the ice germ following a thermally activated "random walk". Generally,

$$Z = \left[ -\frac{\left(\frac{\partial^2 \Delta G^*}{\partial n^2}\right)_{n=n^*}}{2\pi k_B T} \right]^{1/2}$$ (2)

where $\Delta G^*$ is the work of critical germ formation and $n^*$ is the number of water molecules in the ice germ. If the molecular cluster size distribution can be assumed to be near equilibrium, which is the case for the immersion freezing, then

$$C^* = C_0 \exp\left(-\frac{\Delta G^*}{k_B T}\right)$$ (3)

where $C_0$ is the monomer concentration adjacent to the surface of the growing ice germ, impliying that interface transfer is the dominant mechanism of cluster growth.

These expressions can be applied directly to model ice nucleation as follows. For homogeneous ice nucleation, $\Delta G^* = \Delta G_{\text{hom}}$, $f^* = f^*_{\text{hom}}$, and $C_0 = v_w^{-1}$ being $v_w$ the molecular volume of water (Pruppacher and Klett, 1997),

$$J_{\text{hom}} = \frac{Z f^*_{\text{hom}}}{v_w} \exp\left(-\frac{\Delta G_{\text{hom}}}{k_B T}\right),$$ (4)

and for heterogeneous ice nucleation, $\Delta G^* = \Delta G_{\text{het}}$, $f^* = f^*_{\text{het}}$, respectively, $C_0 = a_0^{-1}$ being $a_0$ the average cross-sectional area of a water molecule, i.e.,

$$J_{\text{het}} = \frac{Z f^*_{\text{het}}}{a_0} \exp\left(-\frac{\Delta G_{\text{het}}}{k_B T}\right).$$ (5)

Using $C_0 = a_0^{-1}$ is advantageous because $J_{\text{het}}$ is typically normalized to the particle surface area (Murray et al., 2012; Hoose and Möhler, 2012). It however involves the assumption that the density of water does not vary within the droplet, remaining constant even at the particle-water interface. In other words, anywhere within the liquid the per-area molecular density should

be the same. This assumption however does not lead to significant error since the effect of the particle on the water density is small (e.g., Etzler, 1983) and $J_{\text{het}}$ is linearly related to $C_0$.

Equation (5) provides the basis for this work. It shows that to predict the effect of the immersed particle on ice formation it is necessary to understand how the presence of the particle affects $\Delta G_{\text{het}}$ and $f_{\text{het}}^*$. Such a task is undertaken in this section. Section 2.1 provides an overview of the main assumptions of CNT, which are then constrasted with the Negentropic Nucleation Framework (NNF) in Section 2.2. Sections 2.3 and 2.4 analyze the thermodynamic aspects of immersion ice nucleation, and formulate a new expression for $\Delta G_{\text{het}}$. Section 2.5 develops an expression for $f_{\text{het}}^*$ accounting for the effect of the particle on the mobility of water molecules. In Section 2.6 a new expression for the nucleation rate is formulated.

## 2.1 Classical Nucleation Theory

Since CNT is the most widely used theoretical approach in atmospheric models we start by highlighting its main characteristic. Common CNT expressions use several assumptions to simplify the description of the interaction between water and the immersed particle (e.g., Khvorostyanov and Curry, 2004; Zobrist et al., 2007; Hoose et al., 2010). Typically the particle is assumed to have a negligible effect on the mobility and the thermodynamics of the vicinal water, i.e., $f_{\text{het}}^* \approx f_{\text{hom}}^*$. The later is calculated assuming that the formation of clusters within the liquid phase mimics a first order reaction in an ideal gas where every molecule that randomly jumps the ice-liquid interface is incorporated within the ice lattice. Thus $f_{\text{hom}}^*$ is the product of the frequency factor (derived from transition state theory) and the monomer concentration at the ice-liquid interface. This leads to (Pruppacher and Klett, 1997; Kashchiev, 2000),

$$f_{\text{het, CNT}}^* = f_{\text{hom, CNT}}^* = \frac{\Omega d_0}{v_w} \frac{k_B T}{h} \exp\left[-\frac{\Delta G_{\text{act}}}{k_B T}\right], \tag{6}$$

where $\Delta G_{\text{act}}$ is the activation energy, i.e., the energy required for a water molecule to leave its equilibrium position in the bulk towards the vicinity of the ice germ (Pruppacher and Klett, 1997; Zobrist et al., 2007), $h$ is Plank's constant, $\Omega$ the surface area of the ice germ and $d_0$ is the molecular diameter of water.

The work of ice nucleation results from the assumption that the ice germ has a hemi-spherical shape. Other assumptions include no surface stress (Cahn, 1980) and negligible mixing effects during the germ formation (Barahona et al., 2014). These considerations lead to the expression (Turnbull and Fisher, 1949),

$$\Delta G_{\text{het,CNT}} = g(\theta)\Delta G_{\text{hom, CNT}}, \tag{7}$$

where $\Delta G_{\text{hom, CNT}}$ is the homogeneous nucleation work, given by

$$\Delta G_{\text{hom,CNT}} = \frac{16\pi\sigma_{\text{iw}}^3 v_w^2}{3\left(k_B T \ln S_i\right)^2}, \tag{8}$$

where $\sigma_{\text{iw}}$ is the ice-water interfacial energy and $S_{\text{i}}$ is the supersaturation. The effect of the immersed particle on $J_{\text{het, CNT}}$ depends on the adsorption of water molecules on individual sites, and is characterized by the contact angle, $\theta$, in the form,

$$g(\theta) = \frac{1}{4}(2 + \cos\theta)(1 - \cos\theta)^2. \tag{9}$$

Equation (9) can be extended to account for line tension, curvature and misfit effects (e.g., Khvorostyanov and Curry, 2004), which however requires the usage of additional unconstrained parameters. Introducing Eqs. (6) and (7) into Eq. (5) we obtain the known expression,

$$J_{\text{het, CNT}} = \frac{Z\Omega d_0}{a_0 v_w} \frac{k_{\text{B}} T}{h} \exp\left[-\frac{\Delta G_{\text{act}} + g(\theta)\Delta G_{\text{hom, CNT}}}{k_{\text{B}} T}\right], \tag{10}$$

where $\Omega = 4\pi r_g^2$ being $r_g = \left(\frac{3n^* v_w}{4\pi}\right)^{1/3}$ the ice germ radius. Other symbols are defined in Table 1.

Due in part to the assumption of a negligible effect of the particle on the adjacent water the CNT framework does not provide a way to link the properties of the vicinal water to the nucleation rate. Another caveat is that fundamental parameters like $\Delta G_{\text{act}}$, $\sigma_{\text{iw}}$ and $\theta$ do not have a clear definition outside of the context of the theory. For example, $\Delta G_{\text{act}}$ is typically assumed the same as in bulk water, hence it represents a barrier to "bulk" diffusion instead of interfacial transfer (Kashchiev, 2000; Barahona, 2015). Similarly $\sigma_{\text{iw}}$ is not well defined for a diffuse interface and it is difficult to measure away from equilibrium. Moreover, $\theta$ relies on a droplet-like picture of the nascent ice germ, which may not be appropiate for a germ forming within a denser phase (Brukhno et al., 2008). Due to this most studies treat $\Delta G_{\text{act}}$, $\sigma_{\text{iw}}$ and $\theta$ as empirical parameters, fitted to match measured nucleation rates. Many times this results in complex functional forms of $T$ and $S_{\text{i}}$ that may not be easily expanded to account for the modified properties of water near the immersed particle.

## 2.2 Negentropic Nucleation Framework

Some of the caveats of CNT are addressed in the negentropic nucleation framework (NNF) (Barahona, 2014, 2015). In NNF simple thermodynamic arguments are used to approximate $\Delta G_{\text{hom}}$ and $f_{\text{hom}}$ in terms of water properties that could in principle be independently estimated. This obviates the need for parameters that should be fitted to measured nucleation rates. At the same time, NNF is a relatively simple framework that can be easily implemented in large scale atmospheric models and that has been shown to reproduce homogeneous freezing temperatures down to 180 K (Barahona, 2015; O and Wood, 2016). This section presents the main results of NNF for homogeneous ice nucleation.

In NNF the energy of formation of the interface, $\Phi_{\text{s}}$, is a explicit function of the water activity and temperature in the form,

$$\Phi_{\text{s}} = \Gamma_{\text{w}} s \left(\Delta h_{\text{f}} - \Gamma_{\text{w}} k_{\text{B}} T \ln a_{\text{w}}\right), \tag{11}$$

where the constants $\Gamma_{\text{w}} = 1.46$ and $s = 1.105$ molec$^{1/3}$ define the coverage of the ice-water interface and the lattice geometry of the ice germ, respectively, and $\Delta h_{\text{f}}$ is the latent heat of fusion of water. Other symbols are defined in Table 1. Equation (11)

results from accounting for the finite character of the ice-liquid interface and from the assumption that in joining the ice lattice the water molecules lose most of their entropy (Barahona et al., 2014). The driving force for ice nucleation $\Delta\mu_i$ is given by,

$$\Delta\mu_i = k_\mathrm{B}T\ln\left(\frac{a_\mathrm{w}^2}{a_\mathrm{w,eq}}\right),$$ (12)

where $a_\mathrm{w,eq}$ is the equilibrium water activity. Equation (12) accounts for the work of "unmixing" affecting the bulk of the liquid when the ice germ is formed, which is proportional to $\ln(a_\mathrm{w})$ (Black, 2007). Using Eqs. (11) and (12), the critical germ size and the nucleation work are obtained from the condition of mechanical equilibrium (Barahona, 2014), i.e.,

$$n_\mathrm{hom} = \left(\frac{2\Phi_\mathrm{s}}{3\Delta\mu_i}\right)^3$$ (13)

and,

$$\Delta G_\mathrm{hom,\ NNF} = \frac{4}{27}\frac{\Phi_\mathrm{s}^3}{\Delta\mu_i^2} = \frac{1}{2}n_\mathrm{hom}\Delta\mu_i.$$ (14)

In more recent work the kinetics of homogeneous ice nucleation have been reexamined in NNF to account for molecular rearrangement during the transfer of water molecules across the ice-liquid interface (Barahona, 2015). Within this approach $f_\mathrm{hom}^*$ is determined by the liquid-ice diffusion coefficient for interfacial transfer, $D$, in the form (Kashchiev, 2000; Barahona, 2015),

$$f^* = \frac{D\Omega}{v_w d_0}$$ (15)

where $\Omega$ is the surface area of the ice germ. $D$ represents contributions from purely diffusive process and from structural transformations required to incorporate water molecules into the ice germ. The latter originates because random jump of water molecules across the interface is not a sufficient condition for ice crystal growth. Neighboring molecules need to be rearranged to accommodate new ones into the ice lattice, generating entropy and dissipating work. Using considerations from non-equilibrium thermodynamics $D$ can be written in the form (Barahona, 2015),

$$D = D_\infty\left[1+\exp\left(\frac{W_\mathrm{d}}{k_\mathrm{B}T}\right)\right]^{-1}$$ (16)

where $D_\infty$ the bulk self-diffusion coefficient of water, and $W_\mathrm{d}$ is the average dissipated work during interface transfer. The latter is proportional to the excess free energy of solidification of water, i.e., $W_\mathrm{d} = -n_t\Delta\mu_\mathrm{s}$, being $n_t = 16$, the number of possible trajectories in which individual water molecules can make four-bonded water. Equation (16) shows explicitly that bulk

diffusion (i.e., $D_\infty$) as well a structural rearrangement are required for ice germ growth. Introducing Eq. (16) into Eq. (15) we obtain,

$$f_{\text{hom}}^* = \frac{D_\infty \Omega}{v_w d_0} \left[1 + \exp\left(-n_t \Delta \mu_s\right)\right]^{-1} \tag{17}$$

Application of Eq. (17) in homogeneous ice nucleation shows agreement of $J_{\text{hom}}$ with experimental data at very low $T$, where kinetic processes dominate the formation ice (Barahona, 2015).

NNF provides explicit dependencies of $D$ and $\Phi_s$ on thermodynamic properties withouth depending on nucleation rate measurements. Thus it provides a suitable basis to study the thermodynamics and kinetics of ice formation in the vicinity of immersed particles. Doing so requires first building a model to describe the thermodynamics of the vicinal water.

## 2.3 Thermodynamics of the liquid-particle interface

The discussion presented in Section 1.1 suggests that the immersed particle enhances order near the particle-liquid interface, lowering the energy required to nucleate ice. To represent this the vicinal layer is described as a solution of hypothetical ice-like (IL) and liquid-like (LL) regions, with Gibbs free energy given by

$$\mu_{\text{vc}} = (1 - \zeta)\hat{\mu}_{\text{LL}} + \zeta \hat{\mu}_{\text{IL}}, \tag{18}$$

where $\hat{\mu}_{\text{LL}}$ and $\hat{\mu}_{\text{LL}}$ are the chemical potentials of the LL and IL species within the solution, respectively, and $\zeta$ is the fraction of IL regions in the layer. Increased order is thus represented by a higher fraction of IL regions, hence higher $\zeta$. Equation (18) can also be written in terms of the chemical potentials of the "pure" LL and IL species, $\mu_{\text{LL}}$ and $\mu_{\text{IL}}$, respectively, in the form,

$$\mu_{\text{vc}} = (1 - \zeta)\mu_{\text{LL}} + \zeta \mu_{\text{IL}} + \Delta G_{\text{mix}} \tag{19}$$

where $\Delta G_{\text{mix}} = (\hat{\mu}_{\text{IL}} - \mu_{\text{IL}})\zeta + (1 - \zeta)(\hat{\mu}_{\text{LL}} - \mu_{\text{LL}})$ is the Gibbs energy of mixing. For a mechanical mixture of pure LL and IL species, $\Delta G_{\text{mix}} = 0$, whereas for an ideal solution $\Delta G_{\text{mix}}$ is determined by the ideal entropy of mixing (Prausnitz et al., 1998). Reorganizing Eq. (19) we obtain,

$$\mu_{\text{vc}} = \mu_{\text{LL}} + \zeta \Delta \mu_{\text{il}} + \Delta G_{\text{mix}} \tag{20}$$

where $\Delta \mu_{\text{il}} = \mu_{\text{IL}} - \mu_{\text{LL}}$. $\Delta \mu_{\text{il}}$ can be approximated using the equilibrium between bulk liquid and ice as reference state (Kashchiev, 2000). Making,

$$\mu_{\text{IL}} = \mu_{\text{eq}} + k_{\text{B}} T \ln(a_{\text{IL}}), \tag{21}$$

and

$$\mu_{\mathrm{LL}} = \mu_{\mathrm{eq}} + k_{\mathrm{B}}T\ln\left(\frac{a_{\mathrm{w,\,eff}}}{a_{\mathrm{w,\,eq}}}\right), \tag{22}$$

where $a_{\mathrm{w,\,eff}}$ is termed the "effective water activity" and it is the value of $a_{\mathrm{w}}$ associated with the LL regions in the vicinal water, and $a_{\mathrm{IL}}$ is the water activity in the IL regions. Assuming that similarly to bulk ice the solute does not significantly partition to
the IL phase, then $a_{\mathrm{IL}} \approx 1$. With this, and combining Eq. (21) and Eq. (22) and rearranging we obtain,

$$\Delta\mu_{\mathrm{il}} = -k_{\mathrm{B}}T\ln\left(\frac{a_{\mathrm{w,\,eff}}}{a_{\mathrm{w,\,eq}}}\right). \tag{23}$$

The central assumption behind Eq. (23) is that $a_{\mathrm{w,\,eq}}$ corresponds to the equilibrium water activity between liquid and ice, or in other words that near equilibrium $\Delta\mu_{\mathrm{il}} \approx \Delta\mu_{\mathrm{s}}$. In reality $\Delta\mu_{\mathrm{s}}$ corresponds to actual liquid and ice instead of the hypothetical LL and IL substances. This difference can be accounted for by selecting a proper functional form for $\Delta G_{\mathrm{mix}}$, for which several
empirical and semiempirical interaction models, with varying degrees of complexity exist (Prausnitz et al., 1998). In this work it is going to be assumed that the vicinal water can be described as a regular solution. This is the simplest model that accounts for the interaction between solvent and solute during mixing and that is flexible enough to include corrections for the difference between $\Delta\mu_{\mathrm{s}}$ and $\Delta\mu_{\mathrm{il}}$. Using this model Holten et al. (2013) were able to approximate the chemical potential of supercooled water. The authors also showed that taking into account clustering of water molecules led to better agreement of the estimated
water properties with MD simulations and experimental results.

According to the regular solution model, modified by clustering (Holten et al., 2013, Cf. Eq. 16),

$$\Delta G_{\mathrm{mix}} = \frac{k_{\mathrm{B}}T}{N}\left[\zeta\ln(\zeta) + (1-\zeta)\ln(1-\zeta)\right] + A_w\zeta(1-\zeta) \tag{24}$$

The first term on the right hand side corresponds to the usual definition of the ideal entropy of mixing, i.e., random ideal mixing and a weak interaction between IL and LL regions, modified to account for clustering in groups of $N$ molecules.
$N = 6$ corresponds to clustering in hexamers and is near the optimum fit between MD simulations and the solution model (Holten et al., 2013). It must be noted that Holten et al. (2013) recommended an alternative model termed "athermal solution", where nonideality is ascribed to entropy changes upon mixing. In vicinal water some evidence points at nonideality originating from enthalpy changes near the particle (Etzler, 1983), hence a regular solution is more appropiate in this case. For $N = 6$ the difference between the two models is negligible (Holten et al., 2013).

The second term on the right hand side of Eq. (24) is an empirical functional form used to approximate the enthalpy of mixing, selected so that $\Delta G_{\mathrm{mix}} = 0$ for both, $\zeta = 0$ and $\zeta = 1$. $A_w$ is a phenomenological interaction parameter here assumed to implicitly correct the approximation $\Delta\mu_{\mathrm{il}} \approx \Delta\mu_{\mathrm{s}}$. Typically $A_w$ must be fitted to experimental observations. In this work $A_w$ is calculated using an alternative approach, as follows.

An important aspect of the regular solution model is that it predicts that $\mu_{\text{vc}}$ has a critical temperature, $T_c$, defined by the conditions,

$$\frac{\partial^2 \mu_{\text{vc}}}{\partial \zeta^2} = 0 \, , \ \frac{\partial^3 \mu_{\text{vc}}}{\partial \zeta^3} = 0. \tag{25}$$

These conditions originate because $\frac{\partial^2 \mu_{\text{vc}}}{\partial \zeta^2} = 0$ represents a stability limit for vicinal water. A solution would split into two phases if by doing so lowers its Gibbs free energy (Prausnitz et al., 1998, Cf. Section 6.12). For a metastable solution $\mu_{\text{vc}}$ must be minimal, hence $\frac{\partial \mu_{\text{vc}}}{\partial \zeta} = 0$. The condition $\frac{\partial^2 \mu_{\text{vc}}}{\partial \zeta^2} < 0$ indicates that any increase in $\zeta$ increases $\mu_{\text{vc}}$ (i.e., the curve $\mu_{\text{vc}}$ vs. $\zeta$ becomes concave downward) such that it is thermodynamically more favorable for the solution to split into distinct phases than to increase its concentration. The last condition, $\frac{\partial^3 \mu_{\text{vc}}}{\partial \zeta^3} = 0$, indicates that the metastable region reduces to a single point. Using Eq. (20) into Eq. (25) we obtain,

$$\frac{\partial^2 \mu_{\text{vc}}}{\partial \zeta^2} = \frac{k_{\text{B}} T}{N} \left( \frac{1}{\zeta(1-\zeta)} \right) - 2 A_w = 0, \tag{26}$$

and

$$\frac{\partial^3 \mu_{\text{vc}}}{\partial \zeta^3} = \frac{k_{\text{B}} T}{N} \left( \frac{\zeta^2 - (1-\zeta)^2}{\zeta^2 (1-\zeta)^2} \right) = 0. \tag{27}$$

The last expression is only valid for $\zeta = 0.5$, indicating that a single critical temperature exist for a regular solution. Using this into Eq. (26) and solving for $A_w$ gives for $T = T_{\text{c}}$,

$$A_w = \frac{2 k_{\text{B}} T_{\text{c}}}{N}. \tag{28}$$

Physically, $T_c$ represents the stability limit of the vicinal water, at which it spontaneously separates into IL and LL regions. For $T < T_{\text{c}}$ the chemical potential of a equimolar solution of IL and LL would be larger than that of a simple mechanical mixture of the two species. Thus it is thermodynamically more favorable for the solution to split into its individual components, i.e., ice and liquid, leading to a stability limit of the system. Equation (28) thus provides an opportunity to theoretically determine $A_w$, since $T_c$ should also correspond to a negligible work of nucleation. This further explained in Section 2.4.2.

Introducing Eqs. (23), (24), and (28), into Eq. (20) we obtain,

$$\mu_{\text{vc}} = \mu_{\text{LL}} - \zeta k_{\text{B}} T \ln \left( \frac{a_{\text{w, eff}}}{a_{\text{w, eq}}} \right) + \frac{k_{\text{B}} T}{N} \left[ \zeta \ln(\zeta) + (1-\zeta) \ln(1-\zeta) \right] + \frac{2 k_{\text{B}} T_{\text{c}}}{N} \zeta(1-\zeta). \tag{29}$$

Making,

$$\Lambda_{\text{mix}} = \frac{1}{N} \left[ \zeta \ln(\zeta) + (1-\zeta) \ln(1-\zeta) \right] + \frac{2}{N} \frac{T_{\text{c}}}{T} \zeta(1-\zeta), \tag{30}$$

Equation (29) can be written in the form,

$$\mu_{\text{vc}} = \mu_{\text{LL}} - \zeta k_{\text{B}} T \ln \left( \frac{a_{\text{w, eff}}}{a_{\text{w, eq}}} \right) + k_{\text{B}} T \Lambda_{\text{mix}} \tag{31}$$

Equation (31) is the equation of state of the vicinal water. It describes the properties of the vicinal water in terms of the material-specific parameter $\zeta$, and the interaction parameters $N$ and $T_{\text{c}}$. MD simulations indicate that $N \sim 6$ (Bullock and Molinero, 2013; Holten et al., 2013). $T_{\text{c}}$ is thus the only remaining unknown in Eq. (31) and it is calculated Section 2.4.2.

## 2.4 Work of germ formation

The equation of state of vicinal water can be used to link $\Delta G_{\text{hom}}$ and $\Delta G_{\text{het}}$ as follows. In immersion freezing the particle remains within the droplet long enough that equilibrium is established. This condition is mathematically expressed by the equality, $\mu_{\text{vc}} = \mu_{\text{w}}$, where $\mu_{\text{w}}$ is the chemical potential of water in the bulk of the liquid, i.e., away from the particle. Using Eq. (31) this implies,

$$\mu_{\text{w}} = \mu_{\text{LL}} - \zeta k_{\text{B}} T \ln \left( \frac{a_{\text{w, eff}}}{a_{\text{w, eq}}} \right) + k_{\text{B}} T \Lambda_{\text{mix}}. \tag{32}$$

This expression indicates that the effect of the particle on its vicinal water can be understood as an enhancement of the chemical potential of the LL regions, a consequence of the tendency of the particle to lower $\mu_{\text{vc}}$. Since $\Delta \mu_{\text{il}} < 0$, $\mu_{\text{LL}}$ must increase to maintain equilibrium. Using again the equilibrium between bulk liquid and ice as reference state so that $\mu_{\text{w}} = \mu_{\text{eq}} + k_{\text{B}} T \left( \frac{a_{\text{w}}}{a_{\text{w, eq}}} \right)$, we obtain after symplifying,

$$\ln(a_{\text{w}}) = \ln(a_{\text{w, eff}}) - \zeta \ln \left( \frac{a_{\text{w, eff}}}{a_{\text{w, eq}}} \right) + \Lambda_{\text{mix}}. \tag{33}$$

or equivalently,

$$a_{\text{w}} = a_{\text{w, eff}} \left( \frac{a_{\text{w, eq}}}{a_{\text{w, eff}}} \right)^{\zeta} \exp(\Lambda_{\text{mix}}). \tag{34}$$

Equation (34) suggests that thermodynamically immersion freezing can be described as homogeneous ice nucleation occurring at an enhanced water activity. This is because it is possible to create a path including homogeneous ice nucleation with the same change in Gibbs free energy than for heterogeneous freezing. Figure 1 shows that for a particle-droplet system in equilibrium, $a_{\text{w, eff}}$ satisfies the condition:

$$\Delta G_{\text{het}}(a_{\text{w}}) = \Delta G_{\text{hom}}(a_{\text{w, eff}}). \tag{35}$$

Equation (35) represents a thermodynamic relation between $\Delta G_{\text{het}}$ and $\Delta G_{\text{hom}}$. It has the advantage that $\Delta G_{\text{het}}$ can be obtained without invoking assumptions on the mechanistic details of the interaction between the particle and the ice germ, which are

parameterized by $\zeta$. Since $a_{\text{w}}$ is typically the controlled variable in ice nucleation, $a_{\text{w, eff}}$ can be readily obtained by solving Eq. (34),

$$a_{\text{w, eff}} = \left( \frac{a_{\text{w}}}{a_{\text{w, eq}}^{\zeta}} \right)^{\frac{1}{1-\zeta}} \exp\left( -\frac{\Lambda_{\text{mix}}}{1-\zeta} \right). \tag{36}$$

Although ascribing ice nucleation to the LL fraction of vicinal water agrees with the decisive role of free water in the formation of ice (Wang et al., 2016), caution must be taken in considering this to be the actual mechanism of ice nucleation, which could be quite complex. Equation (35) however establishes a thermodynamic constrain for $\Delta G_{\text{het}}$ that should be met by any ice nucleation mechanism. It is also important to analyze the behavior of $a_{\text{w, eff}}$ as $\zeta \to 1$. It can be shown that the quotient $\frac{\Lambda_{\text{mix}}}{\zeta-1}$ converges for $\zeta \to 1$ as follows. From Eq. (30) we can write,

$$\frac{\Lambda_{\text{mix}}}{1-\zeta} = \frac{1}{N(1-\zeta)} \left[ \zeta \ln(\zeta) + (1-\zeta)\ln(1-\zeta) \right] + \frac{2}{N} \frac{T_{\text{c}}}{T} \zeta, \tag{37}$$

Using $\ln(x) \to (x-1)$ for $x \to 1$ the last expression can be shown to converge to,

$$\lim_{\zeta \to 1} \frac{\Lambda_{\text{mix}}}{1-\zeta} = -\frac{2\zeta}{N} + \frac{2}{N} \frac{T_{\text{c}}}{T} \zeta = \frac{2}{N}\left( \frac{T_{\text{c}}}{T} - 1 \right). \tag{38}$$

The fact that $\lim_{\zeta \to 1} \exp\left( -\frac{\Lambda_{\text{mix}}}{1-\zeta} \right) \neq 1$ for $T \neq T_{\text{c}}$ stems from the simple interaction model used to define $\Delta G_{\text{mix}}$ (i.e., the regular solution approximation). $T_{\text{c}}$ may have depend on $\zeta$ however the regular solution approximation predicts a unique critical temperature at $\zeta = 0.5$. This however does not lead to uncertainty in $\Delta G_{\text{hom}}$ since for $\zeta \to 1$ the first term on the right hand side of Eq. (36) is almost singular at $a_{\text{w}} = a_{\text{w, eq}}$. Thus if $\lim_{\zeta \to 1} \exp\left( -\frac{\Lambda_{\text{mix}}}{1-\zeta} \right) < 1$ then $a_{\text{w}}$ must be just above $a_{\text{w, eq}}$ to make $a_{\text{w, eff}} = 1$. In other words, for all practical purposes $a_{\text{w, eff}} \to 1$ when the system approaches thermodynamic equilibrium.

### 2.4.1 Extension of the homogeneous model to the spinodal limit

In applying the homogeneous model to the heterogeneous problem in the form detailed in Section 2.4, caution must be taken in describing the limiting condition where the size of the ice germ becomes exceedingly small, i.e., $n_{\text{hom}} \to 1$, representing the vanishing of the energy barrier to ice nucleation. This is possible since as $\zeta \to 1$, $a_{\text{w, eff}}$ becomes large (Eq. 36), and for $\zeta = 1$ it is only defined at thermodynamic equilibrium. Since for $n_{\text{hom}} \to 1$ thermodynamic potentials are not well defined it is necessary to test the validity of NNF at such a limit. Moreover, in its original formulation (Section 2.2) NNF predicts a positive $\Delta G_{\text{hom}}$ for $n_{\text{hom}} = 1$, at odds with the notion that the formation of a monomer-sized germ should carry no work.

At the limiting condition, $n_{\text{hom}} = 1$, the work of nucleation is smaller than the thermal energy of the molecules, and represents the onset of spontaneous phase separation (termed "spinodal decomposition") during nucleation (Vekilov, 2010). Here it is argued that being a far-from-equilibrium process ice nucleation always carries energy dissipation. When accounted for, the apparent inconsistency in NNF at $n_{\text{hom}} = 1$ vanishes, since as shown below such condition is not accessible. This approach

differs from previous treatments where this limit is associated with a negligible driving force for nucleation (Kalikmanov and van Dongen, 1993).

To account for the finite, albeit small, amount of work dissipated from the generation of entropy during spontaneous fluctuation, a simple approach is proposed. It involves writing the work of cluster formation in the form,

$$\Delta G = -n\Delta\mu_i + n^{2/3}\Phi_s + W_{\text{diss}} \tag{39}$$

where $W_{\text{diss}}$ represents work dissipation, assumed independent of the germ size since it results from spontaneous fluctuations occurring in the liquid phase. Equation (39) is the typical expression for $\Delta G$ (Barahona, 2014, Cf. Eq. 15) with an additional term accounting for irreversibility. The nucleation work is defined for $n = n_{\text{hom}}$ in the form,

$$\Delta G_{\text{hom}} = -n_{\text{hom}}\Delta\mu_i + n_{\text{hom}}^{2/3}\Phi_s + W_{\text{diss}} \tag{40}$$

where $n_{\text{hom}}$ is obtained from the mechanical stability condition $\frac{\partial\Delta G}{\partial n} = 0$ and still given by Eq. (13), since $W_{\text{diss}}$ is assumed independent of $n$. $W_{\text{diss}}$ is then obtained from the conditions,

$$\Delta G_{\text{hom}}|_{n_{\text{hom}}=1} = 0, \frac{\partial^2\Delta G_{\text{hom}}}{\partial n_{\text{hom}}^2}|_{n_{\text{hom}}=1} = 0. \tag{41}$$

The first condition expresses the fact that the formation of monomer-sized ice germ carries no work. The second condition establishes that $n_{\text{hom}} = 1$ should correspond to a stability limit of the system where nucleation and spontaneous separation are analogous. This is referred as the spinodal point. From Eq. (40) we obtain,

$$\frac{\partial^2\Delta G_{\text{hom}}}{\partial n_{\text{hom}}^2} = -\frac{2}{9}n_{\text{hom}}^{-4/3}\Phi_s = 0. \tag{42}$$

Since $n_{\text{hom}}$ only attains positive values, then only the trivial solution $\Phi_s = 0$ satisfies Eq. (42), i.e., the energy barrier to the formation of the ice germ vanishes at the spinodal. Using $\Phi_s = 0$ and $\Delta G_{\text{hom}}|_{n_{\text{hom}}=1} = 0$, Eq. (40) can be solved for $W_{\text{diss}}$ in the form,

$$W_{\text{diss}} = \Delta\mu_i. \tag{43}$$

Thus the minimum amount of work dissipated during nucleation corresponds to a fluctuation relaxing $\Delta\mu_i$. Replacing this expression within Eq. (40) we obtain,

$$\Delta G_{\text{hom}} = -\Delta\mu_i(n_{\text{hom}} - 1) + n_{\text{hom}}^{2/3}\Phi_s \tag{44}$$

Using Eq. (13) into Eq. (44) gives after rearranging the work of germ formation by homogeneous ice nucleation,

$$\Delta G_{\text{hom}} = \frac{1}{2}\Delta\mu_i(n_{\text{hom}} + 2). \tag{45}$$

Equation (45) only differs from the NNF expression, Eq. (14), on the right hand side, where it is implied that nucleation in solution requires the coordination of at least two molecules, a condition that has been observed experimentally in the crystallization of proteins (Vekilov, 2010). It also suggests that dissipation effects are negligible for typical homogeneous nucleation conditions, i.e., $\Delta G_{\text{hom}} \approx \Delta G_{\text{hom, NNF}}$, since $n_{\text{hom}} \sim 200$ (Barahona, 2014). Moreover, the fact that $\Delta G_{\text{hom}} > 0$ even when $n_{\text{hom}} \to 0$, implies that ice nucleation always requires some work. Using Eq. (35) the heterogeneous work of nucleation can be readily written as,

$$\Delta G_{\text{het}}(a_{\text{w}}) = \left[\frac{1}{2}\Delta\mu_i(n_{\text{hom}} + 2)\right]_{a_{\text{w, eff}}}. \tag{46}$$

Equation (46) also suggests an operational definition for the critical ice germ in immersion freezing in the form,

$$n_{\text{het}} = (n_{\text{hom}} + 2)_{a_{\text{w, eff}}}. \tag{47}$$

The results of Eqs. (45) and (46) require further explanation since in principle an ice germ with only two molecules does not exist. Thus Eq. (45) must be interpreted in a different way. As $\zeta \to 1$, or in deeply supercooled conditions, the fraction of ice-like regions in the vicinal water becomes large. Under such a scenario the reorientation of only two molecules may be enough to initiate ice nucleation. In other words, beyond the spinodal point ice nucleation is controlled by molecular motion within already formed ice-like regions. For homogeneous ice nucleation this would require extreme supercooling ($T \sim 140$ K, Fig. 2). In immersion ice nucleation it may occur at higher $T$ since the formation of a high fraction of ice-like regions in the vicinal water is facilitated by efficient INP. This is further explored in Section 3.

NNF carries the assumption that thermodynamic potentials can be defined for the ice germ. In other words $n_{\text{hom}}$ should be large enough that it represents a statistical ensemble of molecules. Of course this is not the case for $n_{\text{hom}} = 1$, and it may cast doubt on the application of Eq. (39) to such limits. This possibility is however mitigated in two ways. Unlike CNT which is based on the interfacial tension, the NNF framework is robust for small germs. Size effects impact $\Delta G$ mostly through $\Phi_{\text{s}}$ since $\Delta\mu_i$ does not change substantially with the size of the system. In NNF the product $\Gamma_{\text{w}}s\Delta h_{\text{f}}$ in Eq. (11) remains constant, and $\Phi_{\text{s}}$ is relatively insensitive to $n$. This is because $\Delta h_{\text{f}}$ decreases with $n$ as the total cohesive energy of the germ is inversely proportional to the number of molecules within the ice-liquid interfacial layer (Zhang et al., 1999; Johnston and Molinero, 2012). At the same time, the product $\Gamma_{\text{w}}s$, i.e., the ratio of the number of surface to interior molecules in the germ (Barahona, 2014; Spaepen, 1975) should increase for small ice germs offsetting the decrease in $\Delta h_{\text{f}}$. Such behavior is supported by MD simulations (Johnston and Molinero, 2012). Equation (11) thus remains valid for small germs. A second mitigating factor is discussed in Section 3.1 where it is shown that conditions leading to $n_{\text{het}} \to 1$ are rare in the atmosphere, and, $J_{\text{het}}$ is largely independent of $n_{\text{het}}$ for very small germs.

### 2.4.2 Critical temperature

To complete the thermodynamic description of ice nucleation near the particle-liquid interface it is necessary to specify the critical separation temperature defined in Eq. (28). The criterion used to find $T_c$ is that the reversible work of nucleation (that is, without accounting for the dissipation term, $W_{\text{diss}}$) becomes negligible. $W_{\text{diss}}$ is not included since the definition of $\Delta G_{\text{mix}}$ (Eq. 24), does not account for such effects.

Analysis of Eqs. (44) and (46) reveals that $\Delta G_{\text{hom}}$ (hence $\Delta G_{\text{het}}$) is minimum when the reversible work becomes negligible. As $T$ decreases $\Delta \mu_i$ increases, decreasing $n_{\text{hom}}$ and $\Delta G_{\text{hom}}$. However as $n_{\text{hom}} \to 0$ the tendency is reversed since $\Delta G_{\text{hom}} \sim W_{\text{diss}}$. In this regime dissipative effects dominate and $\Delta G_{\text{hom}}$ and $\Delta G_{\text{het}}$ become proportional to $\Delta \mu_i$ (Fig. 2). Thus the minimum in $\Delta G_{\text{het}}$ signals $n_{\text{hom}} \to 0$ and $n_{\text{het}} \to 2$. If no dissipation or kinetic effects were present (like for example at low supercooling) then phase separation would ensue since the work of nucleation would be smaller than the thermal energy of the water molecules. This limit should also correspond to the stability limit of the vicinal water where IL and LL species separate spontaneously; hence it can be used to find $T_c$. It must be noted that this criterion does not mean that $\Delta G_{\text{het}} = 0$ at $T = T_c$ but rather that such would be the case for a thermodynamically reversible nucleation process.

In the regular solution model the interaction parameter $A_w$ is defined for $T_c$ at $\zeta = 0.5$ (Section 2.3). Thus to find $T_c$ we look for the temperature that would produce a minimum in $\Delta G_{\text{het}}$ at $\zeta = 0.5$. Matematically, this is the temperature that simultaneously satisfies the conditions described in Eqs. (25) and (41). Figure 2 shows that this occurs around $T \sim 211$ K for $\zeta = 0.5$. Since both $\Delta G_{\text{het}}$ and $a_{\text{w, eff}}$ depend on $T_c$ we can iteratively look for the minimun solving Eqs. (36) and (46) to find $T_c = 211.473$ K. Figure 2 also suggests that when $T$ remains constant there is a critical value of $\zeta$ that marks the transition between two thermodynamic regimes. This is analyzed in Section 3.1.

### 2.5 Kinetics of immersion freezing

Almost every theoretical approach to describe the effect of INP on ice formation focuses on the thermodynamics of ice nucleation. However as discussed in Section 1.1, increased molecular ordering increases the viscosity of vicinal water, implying that the immersed particle modifies the flux of water molecules to the nascent ice germ, hence the kinetics of ice nucleation (Etzler, 1983; Feibelman, 2010). Since these structural changes are also related to modifications in the chemical potential of the vicinal water, it is likely that the same mechanism that decreases $\Delta G_{\text{het}}$ also controls the mobility of water molecules in the environment around the particle. Such a connection between the water thermodynamic properties and its molecular mobility is well established (Adam and Gibbs, 1965; Debenedetti and Stillinger, 2001; Scala et al., 2000), but it is generally neglected in nucleation theory (e.g., Pruppacher and Klett, 1997; Ickes et al., 2017). In this section a heuristic model is proposed to account for such effects.

Kinetic effects modify the value of the impingement factor, $f_{\text{het}}^*$, which controls the flux of water molecules to the ice germ. In general the ice germ grows by diffusion and rearrangement of nearby water molecules across the ice-liquid interface,

characterized by the interfacial diffusion coefficient, $D$. Increased ordering is characterized by a higher IL fraction, hence higher $\zeta$. Thus, in immersion freezing $D$ must be a function of $\zeta$. Using Eq. (15) this can be expressed in the form,

$$f_{\text{het}}^* = \frac{\Omega}{v_w d_0} D(\zeta), \tag{48}$$

Assuming that within the vicinal layer the ice germ growth follows a similar mechanism as in the bulk of the liquid, then Eq. (16) can be applied to the heterogeneous process in the form,

$$D(\zeta) = D_\infty(\zeta) \left\{ 1 + \exp\left[ \frac{W_{\text{d}}(\zeta)}{k_{\text{B}}T} \right] \right\}^{-1} \tag{49}$$

The last expression indicates that ice-liquid interfacial transfer requires a diffusional and a rearrangement component. $D_\infty(\zeta)$ characterizes purely diffusional processes occuring within the particle-liquid interface. Molecular rearragement during ice germ growth within the vicinal layer is determined by $W_{\text{d}}(\zeta)$. Since only molecules in the LL fraction of the vicinal water would rearrange to join the ice lattice then the latter is given by,

$$W_{\text{d}}(\zeta) = W_{\text{d}}(1 - \zeta) = -n_t \Delta\mu_{\text{s}}(1 - \zeta) \tag{50}$$

Replacing the last expression into Eq. (49) we obtain,

$$D(\zeta) = D_\infty(\zeta) \left[ 1 + \exp\left( \frac{-n_t \Delta\mu_{\text{s}}(1 - \zeta)}{k_{\text{B}}T} \right) \right]^{-1} \tag{51}$$

This expression is consistent with the thermodynamic model presented in Section 2.3 since as $\zeta \to 1$ the vicinal water would have a larger "ice" character and fewer molecules need to rearrange to be incorporated into the growing ice germ.

### 2.5.1 Diffusion within the particle-liquid interface

The diffusional component of $D$ corresponds to the random jump of water molecules across the ice-liquid interface. For $\zeta \to 0$ there is no interaction between the particle and the adjacent water, hence diffusion must proceed as in the bulk of the supercooled water. At the opposite limit, $\zeta \to 1$, $D_\infty(\zeta) \to 0$, which simply states that interfacial transfer vanishes when no net driving force exists across the ice-liquid interface; i.e., the system is in equilibrium. To model this behavior the well-known relaxation theory proposed by Adam and Gibbs (1965) (hereinafter, AG65) is employed. According to AG65, relaxation and diffusion in supercooled liquids require the formation of cooperative regions (CRs). The average transition probability determining the timescale of diffusion is determined by the size of the smallest CR. Following a statistical mechanics treatment and assuming that each CR interacts weakly with the rest of the system, the authors derived the following expression for the average transition probability,

$$\bar{W} \propto \exp\left( -\frac{A}{TS_c} \right), \tag{52}$$

where $A$ represents the product of the minimum size of a CR in the liquid and the energy required to displace water molecules from their equilibrium position in the bulk, and $S_c$ is the configurational entropy. Since $A$ is approximately constant the mobility of water molecules is controlled by $S_c$, which has been confirmed in molecular dynamics simulations and experimental studies (e.g., Scala et al., 2000; Debenedetti and Stillinger, 2001). The self-diffusivity of water is proportional to the transition probability, and can be expressed in the form $D_\infty \sim D_0 \bar{W}$ where $D_0$ is a constant. Using Eq. (52) this suggests the relationship,

$$\frac{D_\infty(\zeta)}{D_\infty} = \frac{\bar{W}(\zeta)}{\bar{W}(\zeta=0)} = \exp\left[-\frac{A}{TS_{c,0}}\left(\frac{S_{c,0}}{S_c}-1\right)\right], \tag{53}$$

where $D_\infty = D_\infty(\zeta=0)$ and $S_{c,0} = S_c(\zeta=0)$ represent values in the bulk of the liquid. Equation (53) implies that the flux of molecules to the ice germ during immersion freezing is controlled by the configurational entropy of vicinal water. Usage of Eq. (53) thus requires developing an expression for $S_c$, which is approximated in the form,

$$S_c = (1-\zeta)S_{c,LL} + \zeta S_{c,IL}, \tag{54}$$

where $S_{c,LL}$ and $S_{c,IL}$ are the configurational entropies of the IL and LL fractions, respectively. The term $S_{c,LL}$ in Eq. (54) dominates $S_c$ since diffusion is controlled by molecules mobile enough to be incorporated in CRs (Stanley and Teixeira, 1980). On the other hand, $S_{c,IL}$ may determine $S_c$ when $\zeta \to 1$.

The regular model proposed in Section 2.3 suggests that the interaction between IL and LL regions is weak since the $\Delta G_{mix}$ is typically small compared to $\mu_{vc}$. Thus we can approximate $S_{c,LL} \approx S_{c,0}$. Unfortunately equating $S_{c,IL}$ to the configurational entropy of bulk ice (which can be deduced from geometrical arguments Pauling (1935)), would violate the requirement that $D \to 0$ at thermodynamic equilibrium. To estimate $S_{c,IL}$ we assume instead that water molecules in the IL regions should be displaced from their equilibrium position (essentially "diffusing" into the LL regions) to be incorporated into the ice lattice. During this processs they gain an amount of energy equal to $-\Delta\mu_s$ which is returned to the system upon entering the ice-liquid interface. Since this energy exchange results mostly from configurational rearrangement (Spaepen, 1975) then we can approximate, $S_{c,IL} \approx -\Delta\mu_s/T$ (Barahona, 2014; Spaepen, 1975). With this, and using $\Delta\mu_s = -k_B T \ln\left(\frac{a_w}{a_{w,eq}}\right)$, Eq. (54) can be rewritten in the form,

$$S_c = S_{c,0}(1-\zeta) + \zeta k_B \ln\left(\frac{a_w}{a_{w,eq}}\right). \tag{55}$$

Introducing this expression into Eq. (53) and rearranging we obtain,

$$\frac{D_\infty(\zeta)}{D_\infty} = \exp\left[-\frac{A}{TS_{c,0}}\frac{\zeta\sigma_E}{(1-\zeta\sigma_E)}\right], \tag{56}$$

where $\sigma_E = 1 - S_{c,0}^{-1} k_B \ln\left(\frac{a_w}{a_{w,eq}}\right)$. Using $D_\infty = D_0 \bar{W}$ an equivalent expression to Eq. (56) can be written in the form,

$$D_\infty(\zeta) = D_\infty \left(\frac{D_\infty}{D_0}\right)^{\frac{\zeta \sigma_E}{1 - \zeta \sigma_E}},$$

(57)

Equation (57) represents the effect of the immersed particle on the rate of growth of the ice germ. For $\zeta = 0$, the particle does not affect the flux of water molecules to the nascent ice germ and $D_\infty(\zeta) = D_\infty$. However as $\zeta \to 1$, $D_\infty(\zeta) \propto \exp\left(-\frac{1}{1 - \zeta \sigma_E}\right)$ and interface transfer becomes severely limited. This effect is much stronger than the reduction in the dissipated work from an increased $\zeta$ (Section 2.5) and dominates $D$.

Introducing Eqs. (51) and (57) into Eq. (48) and rearranging we obtain,

$$f_{het}^* = \frac{D_\infty \Omega}{v_w d_0} \left(\frac{D_\infty}{D_0}\right)^{\frac{\zeta \sigma_E}{1 - \zeta \sigma_E}} \left[1 + \left(\frac{a_w}{a_{w,eq}}\right)^{n_t(1-\zeta)}\right]^{-1},$$

(58)

where $\Delta \mu_s = -k_B T \ln\left(\frac{a_w}{a_{w,eq}}\right)$ was used.

## 2.6 Nucleation Rate

The results of Sections 2.3 to 2.5 provide the basis to write an expression for the ice nucleation rate of droplets by immersion freezing. Before completing such a description we need to provide an expression for $Z$. Application of Eq. (2) typically leads to the known expression (Pruppacher and Klett, 1997),

$$Z = \left(\frac{\Delta G_{het}}{3\pi k_B T n_{het}^2}\right)^{1/2}.$$

(59)

On the other hand using Eq. (46) into Eq. (2) we obtain,

$$Z_d = \left[\frac{\Delta G_{het}(n_{het} - 2)^{1/3}}{3\pi k_B T n_{het}^{7/3}}\right]^{1/2}$$

(60)

where the subscript "d" indicates that energy dissipation is taken into account. For $n_{het} > 3$ it is easily verifiable that $Z_d \approx Z$. Indeed the discrepancy between $Z_d$ and $Z$ is only 30% at $n_{het} = 3$ and much smaller for larger ice germs. However for $n_{het} = 2$, $Z_d = 0$. This issue is rather fundamental and may represent the breaking of the assumption that each germ grows by addition of a single molecule at a time. Hence Eq. (59) will be used keeping in mind that for very small ice germs it represents only an approximation.

With the above considerations it is now possible to substitute Eqs. (46), (47), (58) and (59) into Eq. (5) to obtain the heterogeneous ice nucleation rate,

$$J_{het} = \frac{2 Z D_\infty \Omega}{3 v_w^2} \left(\frac{D_\infty}{D_0}\right)^{\frac{\zeta \sigma_E}{1 - \zeta \sigma_E}} \left[1 + \left(\frac{a_w}{a_{w,eq}}\right)^{n_t(1-\zeta)}\right]^{-1} \exp\left(-\frac{n_{het} \Delta \mu_i}{2 k_B T}\right),$$

(61)

where $d_0 = (6v_w/\pi)^{1/3}$ and $a_0 = \pi d_0^2/4$ were used, and, $\Omega = \Gamma_w s n_{het}^{2/3} a_0$, is the surface area of the ice germ. Other symbols and values used are listed in Table 1.

## 2.7  The role of active sites

There is evidence that in dust and other INP ice is formed preferentially in the vicinity of surface patches, commonly referred as active sites. The existence of active sites have been established experimentally for deposition ice nucleation (i.e., ice nucleation directly from the vapor phase) (Kiselev et al., 2017), and they may be also important for immersion freezing (e.g., Murray et al., 2012). In the classical view active sites have the property of locally decrease $\Delta G_{het}$ increasing $J_{het}$. In the so-called singular hypothesis each active site has an associated characteristic temperature at which it nucleates ice. Current interpretation assigns $J_{het} \to \infty$ at each active site at its characteristic temperature, with some variability due to "statistical fluctuations" in the germ size (Vali, 2014). Some CNT-based approaches to describe immersion freezing account for the existence of active sites by assuming distributions of contact angles for each particle. Hence each active site is assigned a characteristic contact angle instead of a characteristic temperature (e.g., Zobrist et al., 2007; Ickes et al., 2017).

The view of the role of active sites as capable of locally decreasing $\Delta G_{het}$ relies heavily on an interpretation of immersion freezing that mimics ice nucleation from the vapor phase (Fig. 3a). Such a description is however too limited for ice formation within the liquid phase. For example, it is implicitly assumed that the active site brings molecules together, similar to an adsorption site. However a particle immersed within a liquid is already surrounded by water molecules (Fig. 3b). In fact, nascent ice structures are associated with low density regions within the liquid (Bullock and Molinero, 2013). Thus in the classical view the active site should be able to "pull molecules apart" instead of bringing them together. This creates a conceptual problem. To locally reduce $\Delta G_{het}$ active sites should be able to permanently create low density regions within the liquid, which would require a large amount of energy. In other words, active sites would have the unusual property of creating a thermodynamic barrier maintaining their surrounding water in a non-equilibrium state. Such situation is unlikely in immersion freezing.

The concept of local nucleation rate also presents some difficulties. In the strict sense $J_{het}$ is the velocity with which the size distribution of molecular clusters in an equilibrium population crosses the critical size (Kashchiev, 2000; Seinfeld and Pandis, 1998). In immersion freezing the domain of such a distribution is the whole volume of the droplet. Thus only a single value of $J_{het}$ can be defined for a continuous liquid phase, independently of where the actual nucleation process is occurring, since no permanent spatial gradients of $T$ or concentration exist within equilibrium systems. Having otherwise implies that parts of the system would need to be maintained in a non-equilibrium state, having their own cluster size distribution. This requires the presence of non-permeable barriers within the liquid, a condition not encountered in immersion freezing. Similarly, the characteristic temperature of an active site is an unmeasurable quantity since a system in equilibrium has the same temperature everywhere. Hence it would be impossible to distinguish whether the particle as a whole or only the active site must reach certain temperature before nucleation takes place.

These difficulties can be reconciled if instead of promoting nucleation through a thermodynamic mechanism, active sites provide a kinetic advantage to ice nucleation. A way in which this can be visualized is shown in Fig. 3b. The vicinal water is in equilibrium with the particle, and exhibits a larger degree of ordering near the interface. Since in immersion freezing

the formation of ice in the liquid depends on molecular rearrangement, the active site should produce a transient structural transformation that allows the propagation of ice. These sites would be characterized by defects where templating is not efficient allowing greater molecular movement hence facilitating restructuring. Their presence is guaranteed since particles are never uniform at the molecular scale. In this view active sites create ice by promoting fluctuation instead of by locking

water molecules in strict configurations. It implies that for uniform systems (e.g., a single droplet with a single particle) $\Delta G_{\text{het}}$ depends on the equilibrium between the particle and the vicinal water and active sites enhance fluctuation around specific locations. This obviates the need for the hypothesis of a well-defined characteristic temperature for each active site. It however does not mean that active sites are transient. They are permanent features of the particle and should have a reproducible behavior, inducing ice nucleation around the same place in repeated experiments (e.g., Kiselev et al., 2017).

Within the framework presented above, there can only be one $J_{\text{het}}$ defined in the droplet volume. The presence of active sites introduces variability in $J_0$ instead of $\Delta G_{\text{het}}$. The latter is determined by the thermodynamic equilibrium between the particle and its vicinal water. Although the theory presented here does not account for internal gradients in the droplet-particle system, in practice it is likely that the that the observed $J_{\text{het}}$ corresponds to the site promoting the largest density fluctuations. Variability in $J_{\text{het}}$ would be introduced by fluctuation in the cluster size distribution in the liquid and from multiplicity of active

sites in the particle population. In this sense the proposed view is purely stochastic.

## 3   Discussion

### 3.1   Ice nucleation regimes

A consequence of the linkage between the properties of vicinal water and $\Delta G_{\text{het}}$ is the existence of distinct nucleation regimes. This was mentioned in Section 2.4.1 and here it is explored in detail. Recall from Fig. 2, that for a given temperature $\Delta G_{\text{het}}$

passes by a minimum defined by the condition $\frac{\partial^2 \Delta G_{\text{het}}}{\partial n_{\text{het}}^2} = 0$. Figure 4 (right panel) depicts a similar behavior but maintaining $\zeta$ constant instead of $T$. It shows that there is a temperature, $T_s$, at which $\Delta G_{\text{het}}$ is minimum. For $T > T_s$ $\Delta G_{\text{het}}$ increases with increasing $T$ because $n_{\text{het}}$ increases (Fig. 4, left panels). This is the typical behavior predicted by the classical model (e.g., Khvorostyanov and Curry, 2005) hence such regime will be termed "germ-forming" since $\Delta G_{\text{het}}$ is determined by the formation of the ice-liquid interface.

A different behavior is found for $T < T_s$, where $\Delta G_{\text{het}}$ decreases with increasing $T$. In this regime $n_{\text{het}}$ remains almost constant at very low values, $\Delta G_{\text{het}}$ is small and results mostly from the dissipation of work. Ice nucleation is not limited by the formation of the ice-liquid interface but rather by the propagation of small fluctuations in the vicinity of pre-formed ice-like regions. Therefore it is controlled by diffusion of water molecules to such regions rather than by $\Delta G_{\text{het}}$. This is akin to a spinodal decomposition process (Cahn and Hilliard, 1958) and will be termed "spinodal ice nucleation". It is however not truly

spinodal decomposition since it requires a finite, albeit small, amount of work to occur.

Since for each value of $\zeta$ there is a minimum in $\Delta G_{\text{het}}$ (Fig. 4), then theoretically all INPs are capable of nucleating ice in both regimes. However, in practice spinodal ice nucleation would only occur if $T_s$ lies within the 233 K $< T <$ 273 K range where immersion freezing occurs. For example, for $\zeta = 0.1$, Fig. 4, right panel, shows that the minimum in $\Delta G_{\text{het}}$ occurs

at $T < 220$ K. Since homogeneous ice nucleation should occur above this temperature, INP characterized by $\zeta = 0.1$ will not exhibit spinodal ice nucleation. These particles always nucleate ice in the classical germ-forming regime ($T > T_s$). The situation is however different for $\zeta = 0.9$, since $T_s \approx 270$ K. These INP are capable of nucleating ice in both the spinodal ($T < T_s$) and the germ-forming ($T > T_s$) regimes. For the spinodal regime $\Delta G_{het}$ is very low and decreases slighly with increasing $T$, indicating that the thermodynamic barrier to nucleation is virtually removed. Ice formation is therefore almost entirely controlled by kinetics.

The existence of the spinodal nucleation regime signals the possibility of an interesting behavior in freezing experiments, where the same $\Delta G_{het}$ may correspond to two very different INP. To show this the values of $\Delta G_{het}$ and $n_{het}$ corresponding to $J_{het} = 10^6$ m$^{-2}$ s$^{-1}$ are depicted in Fig. 4, black lines. These lines form semi-closed curves when plotted against temperature indicating that the same $\Delta G_{het}$ may correspond to two different values of $\zeta$. The upper branch (with high $\Delta G_{het}$) corresponds to the germ-forming regime and the lower branch to the spinodal regime. This picture maybe convoluted by the fact that high $\zeta$ also implies strong kinetic limitations during ice nucleation and is further discussed in Section 3.2.

## 3.2 Preexponential Factor

Kinetics effects on ice nucleation are typically analyzed in terms of the preexponential factor, which is proportional to $f_{het}^*$ in the form,

$$J_0 = \frac{Z f_{het}^*}{a_0}. \tag{62}$$

$J_0$ expresses the normalized flux of water molecules to the ice germ, corrected by $Z$. Figure 5, shows $J_0$ calculated using Eqs. (58) and (59). Results from CNT (Eq. 6) are also shown. In general $J_0$ varies with $T$ and $\zeta$. The sensitivity of $J_0$ to $T$ is determined by $D_\infty$ (Barahona, 2015) with $J_0$ increasing with $T$, since water molecules increase their mobility. Also at higher $T$ less work is dissipated during interface transfer. These effects dominate the variation in $J_0$ for $\zeta < 0.5$, suggesting that the particle has a limited effect on the mobility of vicinal water. Ice nucleation around these particles would be reasonably well described by assuming a negligible effect of the particle on $J_0$, as done in CNT. This is evidenced by the CNT-derived values for $\theta = 10°$ and $\theta = 90°$, which represent particles with high and low particle-ice affinity, respectively, and correspond to the range of expected variability in CNT. The $\theta = 90°$ and $\zeta \sim 0$ lines in Fig. 5 are within an order of magnitude of each other, in agreement with homogeneous nucleation results (Barahona, 2015). The $\theta = 10°$ line is also close to the $\zeta \sim 0.5$ curve. In both cases $J_0$ increases by about two orders of magnitude between 220 K and 273 K and decreases by about two orders of magnitude from $\zeta = 0.0$ to $\zeta = 0.5$, or from $\theta = 90°$ to $\theta = 10°$ in CNT. The latter reflects the effect of variation in $Z$ on $J_0$.

The behavior of $J_0$ for $\zeta > 0.5$ dramatically differs from CNT. For $\zeta > 0.5$, and particularly for $\zeta > 0.8$, $J_0$ decreases strongly with increasing $T$. This is because as $\zeta \to 1$ and $T \to 273$ K the driving force for interfacial transfer, i.e., the separation of $\mu_{vc}$ from thermodynamic equilibrium vanishes. As the system moves near these conditions $D$ becomes very small. This is the result of the high IL fraction of the vicinal water limiting the number of configurations available to form cooperative regions, required to induce water mobility (Section 2.5.1). Such a behavior cannot be reproduced by CNT since no explicity dependency of $D$

on the properties of the vicinal layer is accounted for. For $\zeta > 0.99$ $J_0$ decreases by more than 30 orders of magnitude from 220 K to $273K$; molecular transport nearly stops. Ice nucleation may not be possible at such extreme, despite the fact that these particles very efficiently reduce $\Delta G_{\text{het}}$ (Fig. 4); water may remain in the liquid state at very low temperature. Such an effect has been experimentally observed in some biological systems (Wolfe et al., 2002).

## 3.3 Nucleation Rate

The interplay between kinetics and thermodynamics determines the complex behavior of $J_{\text{het}}$ in immersion ice nucleation. Particles highly efficient at decreasing $\Delta G_{\text{het}}$ also decrease the rate of interfacial diffusion to the point where they may effectively prevent ice nucleation. On the other hand, INP with low $\zeta$ do not significantly affect $J_0$ but have a limited effect on $\Delta G_{\text{het}}$. This is confounded with the presence of two thermodynamic nucleation regimes, where $\Delta G_{\text{het}}$ may be large and increases with $T$ ("germ-forming"), and another one where $\Delta G_{\text{het}}$ is very small and decreases as $T$ increases ("spinodal nucleation"). This picture can be simplified since within the range $233 \text{ K} < T < 273 \text{ K}$, where immersion freezing is relevant for atmospheric conditions, INP with $\zeta > 0.7$ are at the same time more likely to nucleate ice in the spinodal regime and to exhibit strong kinetic limitations. Similarly for $\zeta < 0.6$ the transition to spinodal nucleation occurs below 233 K (Fig. 2). These INP tend to nucleate ice in the germ-forming regime without significantly affecting $J_0$. Thus the thermodynamic regimes introduced in Section 3.1 loosely correspond to kinetic regimes. Roughly, ice nucleation in the spinodal regime is controlled by kinetics and in the germ-forming regime it is controlled by thermodynamics. This is a useful approximation but it should be used with caution. Even in the germ-forming regime the particle affects the kinetics of ice-liquid interfacial transfer to some extent. Similarly, in the spinodal regime $\Delta G_{\text{het}}$ is small, but finite.

Figure 6 shows the behavior of $J_{\text{het}}$ as $T$ increases for different values of $\zeta$. $J_{\text{het}}$ in the germ-forming regime resembles the behavior predicted by CNT. $J_{\text{het}}$ increases steeply with decreasing $T$ and increasing $\zeta$. Similarly for CNT, $J_{\text{het}}$ increases for decreasing $T$ and $\theta$. This is characteristic of the thermodyamic control on $J_{\text{het}}$ where $\Delta G_{\text{het}}$ and $\frac{\mathrm{d}\Delta G_{\text{het}}}{\mathrm{d}T}$ are large (Fig. 4), and $J_0$ is relatively unaffected by the particle. In this regime it is always possible to find a contact angle (typically between $10°$ and $100°$) that results in close agreement of $J_{\text{het}}$ between CNT and NNF predictions (Fig. 6), particularly for $J_{\text{het}} < 10^{12}\text{cm}^{-2}\text{s}^{-1}$ which covers most values of atmospheric interest. This is also true for $a_{\text{w}} = 0.9$ (Fig. 6, right panel) although the approximation to the equilibrium temperature signals a steeper behavior in CNT peaking at higher values than NNF. Since $\frac{\mathrm{d}J_{\text{het}}}{\mathrm{d}T}$ is large, $J_{\text{het}}$ may show threshold behavior, characteristic of ice nucleation mediated by some dust species like Chlorite and Montmorrillonite (Atkinson et al., 2013; Murray et al., 2012; Hoose and Möhler, 2012).

There is however no value of $\theta$ that would lead to overlap between CNT and NNF for $\zeta > 0.7$. These conditions corresponds largely to spinodal ice nucleation. $J_{\text{het}}$ is kinetically controlled since $\Delta G_{\text{het}}$ is small and $J_0$ varies widely with $T$ (Fig. 5). As in the germ-forming regime $J_{\text{het}}$ also reaches significant values, but increases more slowly with decreasing $T$ (Fig. 6). Higher $\zeta$ leads to $J_{\text{het}}$ becoming significant at higher $T$. But unlike in the germ-forming case, curves with higher $\zeta$ tend to plateau at progressively lower values of $J_{\text{het}}$ since they become kinetically limited by their approximation to the thermodynamic equilibrium. For $\zeta \sim 0.7$ some of the curves of Fig. 6 also display germ-forming behavior at high $T$, and are characterized by sudden decrease in $-\frac{\mathrm{d}J_{\text{het}}}{\mathrm{d}T}$ as $T$ decreases. The sudden change of slope corresponds to the region around the minimum $\Delta G_{\text{het}}$

(Fig. 4) and signals the transition from germ-forming to spinodal ice nucleation. Such behavior has been observed in some INP of bacterial origin (Murray et al., 2012).

Figure 6 also indicates that nucleation regimes cannot be assigned based on the values of $J_{het}$ or on the observed freezing temperature, $T_f$. In both regimes, $J_{het}$ may reach substantial values, hence $T_f$ may cover the entire range 233 K $< T <$ 273 K. What is striking is that $J_{het}$ curves with $\zeta > 0.7$ tend to cross those with $\zeta < 0.7$. This means that two INP characterized by very different $\zeta$ can have the same freezing temperature. This result thus challenges the common notion that INP with higher freezing temperatures are intrinsically more active at nucleating ice, or in other words, that by measuring $T_f$ alone it is possible to characterize the freezing properties of a given material. In reality to discern whether the observed $T_f$ corresponds to a good (in the thermodynamic sense) INP acting in the spinodal regime or a less active INP acting in the germ-forming regime it is necessary to measure $\frac{dJ_{het}}{dT}$ along with $T_f$.

### 3.4 Application to the water activity-based nucleation rate

If a droplet is in equilibrium with its environment then $a_w$ is a function of the relative humidity. Thus the relationship between $a_w$ and the freezing temperature, $T_f$, conveys important information about the potential of a particle to catalyze the formation of ice, and can be used to generate parameterizations of immersion ice nucleation for cloud models (Kärcher, 2003; Koop and Zobrist, 2009; Barahona and Nenes, 2009). A widely used class of parameterizations is based on the so-called water activity criterion (Koop et al., 2000; Koop and Zobrist, 2009), the condition that for a given material the water activity at which heterogeneous ice nucleation is observed, $a_{w,\,het}$, is related by a constant to $a_{w,\,eq}$ (Koop et al., 2000; Koop and Zobrist, 2009). Here it is shown that the two-state thermodynamic model proposed in Section 2.3 implies the water activity criterion as a purely thermodynamic constraint to freezing.

#### 3.4.1 Water activity shift

By definition the thermodynamic path shown in Fig. 1 operates between two equilibrium states. The relation between $\Delta G_{het}$ and $\Delta G_{hom}$ is therefore independent of the way the system reaches $a_{w,\,eff}$. In absence of any kinetic limitations to the germ growth, Eq. (35) also represents a direct relationship between $J_{hom}$ and $J_{het}$. (Kärcher, 2003; Marcolli et al., 2007; Koop and Zobrist, 2009; Knopf and Alpert, 2013). Thus one can imagine two separate experiments in which the environmental conditions are set to either $a_w$ or $a_{w,\,eff}$, the former resulting in heterogeneous freezing and the latter in homogeneous ice nucleation. Under these conditions Eq. (34) implies that when heterogeneous ice nucleation is observed at $a_{w,\,het} = a_w$ there is a corresponding homogeneous process that would occur at $a_{w,\,hom} = a_{w,\,eff}$. Thus we can write an equivalent expression to Eq. (34), but relating $a_{w,\,het}$ and $a_{w,\,hom}$ in the form,

$$a_{w,\,het} = a_{w,\,hom} \left( \frac{a_{w,\,eq}}{a_{w,hom}} \right)^\zeta \exp(\Lambda_{mix}), \tag{63}$$

Eq. (63) can be rewritten as,

$$\ln(a_{w, het}) = (1 - \zeta) \ln(a_{w, hom}) + \zeta \ln(a_{w, eq}) + \Lambda_{mix}. \tag{64}$$

Substracting $\ln(a_{w, eq})$ from each side of Eq. (64) gives,

$$\ln(a_{w, het}) - \ln(a_{w, eq}) = (1 - \zeta) \left[ \ln(a_{w, hom}) - \ln(a_{w, eq}) \right] + \Lambda_{mix}. \tag{65}$$

Using the approximation $\ln(x) \approx x - 1$ for $x \sim 1$, Eq. (64) can be linearized in the form,

$$\Delta a_{w, het} = \Delta a_{w, hom}(1 - \zeta) + \Lambda_{mix}, \tag{66}$$

where $\Delta a_{w, hom} = a_{w, hom} - a_{w, eq}$ and $\Delta a_{w, het} = a_{w, het} - a_{w, eq}$ are the homogeneous and heterogeneous water activity shifts, respectively. $\Delta a_{w, hom}$ has been found to be approximately constant for a wide range of solutes (Koop et al., 2000); therefore Eq. (66) suggests that $\Delta a_{w, het}$ should be approximately constant since $\Lambda_{mix} \sim 0.02$ and only depends on $T$. Thus, the two-state model presented in Section 2.3 implies the so-called water activity criterion (Koop et al., 2000) for heterogeneous ice nucleation, giving support to the hypothesis that increasing order near the particle surface drives ice nucleation.

Equations (63) to (66) are fundamental thermodynamic relationships of the system and can be used to analyze the effect of the immersed particle on ice formation independently of kinetic effects. To do so $a_{w, hom}$ must be determined entirely by thermodynamics. This is because if $a_{w, hom}$ is defined at some $J_{hom}$ threshold then it (and by extension $a_{w, het}$) would also depend on the freezing kinetics. A thermodynamic definition of $a_{w, hom}$ has been achieved by Baker and Baker (2004). The authors showed that on average freezing occurs below the temperature at which the compressibility of water reaches a maximum. A this point density fluctuations are wide enough to allow structural tranformations that facilitate the formation of ice-like regions within the droplet volume (Bullock and Molinero (2013) also derived a pure thermodynamic criterion for $a_{w, hom}$ using the equilibrium between low density regions and the bulk solution). Such a criterion does not depend on measured freezing rates and can be extended to the freezing of water solutions, coinciding with the the results of Koop et al. (2000). Within this frameworks $a_{w, hom}$ can be defined without reference to a $J_{hom}$ threshold. By extension, Eq. (64) guarantees that $a_{w, het}$ can be determined entirely by the thermodynamic properties of the system.

Equation (64) also implies that for a given $a_{w, hom}$ there is a temperature for which $a_w = a_{w, het}$, referred as the "Thermodynamic Freezing Temperature", $T_{ft}$. Formally, $T_{ft}$ represents the solution of

$$\ln(a_w) - (1 - \zeta) \ln \left[ a_{w,eq}(T_{ft}) + \Delta a_{w, hom} \right] - \zeta \ln \left[ a_{w,eq}(T_{ft}) \right] - \Lambda_{mix}(T_{ft}) = 0, \tag{67}$$

or in the linearized form,

$$a_w - a_{w,eq}(T_{ft}) - \Delta a_{w, hom}(1 - \zeta) - \Lambda_{mix}(T_{ft}) = 0. \tag{68}$$

Since $\Delta a_{\text{w, hom}}$ is considered a thermodynamic property of the system (Baker and Baker, 2004) then $T_{\text{ft}}$ does not depend on the freezing kinetics. Thus $T_{\text{ft}}$ can be interpreted as the highest temperature where it is most likely to observe ice nucleation for a given thermodynamic state (determined by $a_{\text{w}}$, $\zeta$, and the system pressure).

Figure 7 shows the $T_{\text{ft}} - a_{\text{w}}$ relationship defined by Eq. (67), calculated using $\Delta a_{\text{w, hom}} = 0.304$ (Koop et al., 2000; Barahona, 2014; Baker and Baker, 2004). As expected, the figure resembles experimental results found by several authors (e.g., Koop and Zobrist, 2009; Zuberi et al., 2002; Zobrist et al., 2008; Alpert et al., 2011; Knopf and Alpert, 2013) where curves for $\zeta > 0$ align with constant water activity shifts to $a_{\text{w, eq}}$. To make this evident, lines were drawn using constant values of $\Delta a_{\text{w, het}} = 0.05, 0.15$, and $0.20$ which coincide with lines corresponding to $\zeta = 0.2, 0.3$ and $0.7$, respectively. This shows that Eq. (66) is a good approximation to Eq. (63), and constitutes a theroretical derivation of the water activity criterion. The fact that such behavior can be reproduced by Eq. (63) validates the regular solution approximation used in Section 2.3 and supports the idea that the effect of the immersed particle on ice nucleation can be explained as a relative increase in the ice-like character of the vicinal water.

It must be emphasized that $T_{\text{ft}}$ only establishes the potential of an INP to induce freezing at $a_{\text{w}} = a_{\text{w, het}}$, regardless of whether a measurable $J_{\text{het}}$ can be experimentally realised. Physically, it is plausible that as the particle increases the ice-like character of the vicinal water it also increases the probability of wide density fluctuations. As a result low density regions, wide enough to accommodate the ice gem, exist at higher $T$ than in homogeneous ice nucleation. Following the argument of Baker and Baker (2004) this would also imply that the compressibility of water near the particle reaches a maximum at higher $T$ than in the bulk. More research however is needed to elucidate this point. The presence of a spinodal regime would also mean that the observed freezing temperature may differ from $T_{\text{ft}}$ since at such a limit nucleation is no longer controlled by thermodynamics. This is illustrated in the next section.

### 3.4.2 Freezing by Humic-Like INP

$\Delta a_{\text{w, het}}$ has been determined in several studies and used to predict and parameterize $J_{\text{het}}$ in atmospheric models (e.g., Zobrist et al., 2008; Knopf and Alpert, 2013). Thus it is useful to analyze under what conditions $\zeta$ (hence $J_{\text{het}}$) can be estimated using measured $\Delta a_{\text{w, het}}$ values. Rearranging Eq. (66) we obtain,

$$\Delta a_{\text{w, het}} - \Delta a_{\text{w, hom}}(1 - \zeta) - \Lambda_{\text{mix}} = 0. \tag{69}$$

If $\Delta a_{\text{w, hom}}$ and $\Delta a_{\text{w, het}}$ are known, $\zeta$ can be estimated iteratively solving Eq. (69). Note that $\Lambda_{\text{mix}}$ is temperature dependent (Eq. 34) implying a slight dependency of $\zeta$ on $T$ when $\Delta a_{\text{w, het}}$ is constant. However since $\Lambda_{\text{mix}}$ is also typically small $\zeta$ is almost proportional to $1 - \frac{\Delta a_{\text{w, het}}}{\Delta a_{\text{w, hom}}}$.

To test Eq. (69) the data for Leonardite (LEO) and Pawokee Peat (PP) particles (humic-like substances) obtained by Rigg et al. (2013) are used. The authors reported $\Delta a_{\text{w, het}} = 0.2703$ for LEO and $\Delta a_{\text{w, het}} = 0.2466$ for PP. These values are assumed to be independent of $a_{\text{w}}$ and $T$ with an experimental error in $\Delta a_{\text{w, het}}$ of $0.025$. Average $J_{\text{het}}$ obtained from different samples and from repeated freezing and melting experiments for both materials is depicted in Fig. 8. Applying Eq. (69) over the $T = 210$

K$-250$ K range and using $\Delta a_{\text{w, hom}} = 0.304$ results in $\zeta = 0.049 - 0.058$ for LEO and $\zeta = 0.096 - 0.121$ for PP. Within this temperature range these values correspond to the germ-forming regime, hence $J_{\text{het}}$ is thermodynamically-controlled. Comparison against the experimentally determined $J_{\text{het}}$ for three different values of $a_{\text{w}}$ is shown in Fig. 8. Within the margin of error there is a reasonable agreement between the modeled and the experimental $J_{\text{het}}$.

Figure 8, top panels, however reveals that even if $J_{\text{het}}$ becomes significant around the values predicted by Eq. (69), $-\frac{d \ln J_{\text{het}}}{dT}$ is overestimated, particularly for PP. This may indicate that that these INP nucleate ice in the spinodal regime. To test this hypothesis $J_{\text{het}}$ was fitted to the reported measurements by varying $\zeta$ within the range where spinodal nucleation would be dominant. To avoid agreement by design a single $\zeta$ was used for all experiments for each species resulting in $\zeta = 0.949$ for PP and $\zeta = 0.952$ for LEO (Fig. 8, bottom panels). For PP $J_{\text{het}}$ and $-\frac{d \ln J_{\text{het}}}{dT}$ agree better with the experimental values, whereas

for LEO the agreement improves at high $T$ but worsens at low $T$. In this regime $J_{\text{het}}$ seems to be slightly overestimated by the theory at the lowest $a_w$ tested. This may be due to small uncertainties in $a_{\text{w}}$ that play a large role in $J_{\text{het}}$ (as for example the assumption of a $T$-independent $a_{\text{w}}$, Alpert et al. (2011)). There is the possibility that the humic acid present in PP may slightly dissolve during the experiments (D. Knopf, personal communication), which would impact not only $a_{\text{w}}$ but also may modify the composition of the particles, hence $\zeta$.

The exercise above suggests that ice nucleation in PP may follow a spinodal mechanism. Using a single value of $\Delta a_{\text{w, het}}$ to predict $\zeta$, as expressed mathematically by Eq. (69), seems to work for LEO. Since Eq. (69) represents a thermodynamic relation between $\Delta a_{\text{w, hom}}$ and $\Delta a_{\text{w, het}}$, it is expected to work well when nucleation is thermodynamically-controlled, i.e, the germ-forming regime. However it may fail for spinodal ice nucleation since it does not consider the effect of the particle on $J_0$. $\Delta a_{\text{w, het}}$ however carries important information about $J_{\text{het}}$ (Knopf and Alpert, 2013) but for spinodal ice nucleation

the relationship between $\Delta a_{\text{w, het}}$ and $\zeta$ must be more complex than predicted by Eq. (69) since kinetic limitations play a significant role. Figure 8 also shows that similar $T_{\text{f}}$ can be obtained by either high or low $\zeta$. The particular regime in which an INP nucleates ice determines $-\frac{d \ln J_{\text{het}}}{dT}$, hence the sensitivity of the droplet freezing rate to the particle size and to the cooling rate.

### 3.5  Limitations

It is important to analyze the effect of several assumptions introduced in Section 2 on the analysis presented here. One of the limitations of the approach used in deriving Eq. (61) is that it employs macroscale thermodynamics in the formulation of the work of nucleation. The effect of this assumption is however minimized in several ways. First, unlike frameworks based on the interfacial tension, NNF is much more robust to changes in ice germ size since the product $\Gamma_{\text{w}} s \Delta h_{\text{f}}$ remains constant (Section 2.4). Second, in the spinodal regime $\Delta G_{\text{het}}$ is independent of $n_{\text{het}}$ and only for $T > 268$ K and in the germ-forming regime, the

approach presented here may lead to uncertainty (Section 3.1). Thus Eq. (61) remains valid for most atmospheric conditions, although caution must be taken when $T_{\text{f}} > 268$ K. Alternatively the framework presented here could be extended to account explicitly for the effect of size on $\Delta h_{\text{f}}$ and $\Gamma_{\text{w}}$ (e.g., Zhang et al., 1999).

Further improvement could be achieved by implementing a more sophisticated equation of state for the vicinal water. Here a two-state assumption has been used, such that $\mu_{\text{vc}}$ is a linear combination of ice-like and liquid-like fractions. Such approxima-

tion has been used with success before (Etzler, 1983; Holten et al., 2013). However it is known that the structure of supercooled water represents an average of several distinct configurations (Stanley and Teixeira, 1980). These are in principle accounted for in the proposed approach since $\zeta$ represents a relative, not an absolute increase in the IL fraction. However there is no guarantee that such increase can be linearly mapped in the way described in Section 2. Fortunately this would only mean in practice that

the value of $\zeta$ for a given material is linked to the particular form of the equation of state used to describe the vicinal water.

Equation (61) is also blind to the surface properties of the immersed particle. The implicit assumption is that the effect of surface composition, charge, hydrophilicity and roughness on $J_{\text{het}}$ can be parameterized as a function of $\zeta$. The example shown in Section 3.4 suggests this is indeed the case. Making such relations explicit must however lie at the center of future development of the proposed approach. Similarly a heuristic approach was used to study the effect of irreversibility on the

nucleation work. This can be improved substantially by making use of a generalized Gibbs approach (Schmelzer et al., 2006), which unfortunately may also increase the number of free parameters in the model. None of these limitations is expected to change the conclusions of this study, however they may affect the values of $\zeta$ fitted when analyzing experimental data. The approach proposed here however has the advantage of being a simple, one parameter approximation that can be easily implemented in cloud models.

**4   Summary and Conclusions**

Immersion freezing is a fundamental cloud process and its correct representation in atmospheric models is critical for accurate climate and weather predictions. Current theories rely on a view that mimics ice formation from the vapor, neglecting several interactions unique to the liquid. This work develops for the first time a comprehensive approach to account for such interactions. The ice nucleation activity of immersed particles is linked to their effect on the vicinal water. It is shown that the same

mechanism that lowers the thermodynamic barrier for ice nucleation also tends to decrease the mobility of water molecules, hence limiting interfacial transfer and ice germ growth. The role of the immersed particle in ice nucleation can be understood as increasing order in the adjacent water facilitating the formation of ice-like structures. Thus, instead of being purely driven by thermodynamics, heterogeneous ice nucleation in the liquid phase is a process determined by the competition between thermodynamic and kinetic constraints to the formation and propagation of ice.

In the new approach the properties of vicinal water are approximated using a regular solution between high and low density regions, with composition defined by an aerosol specific parameter, $\zeta$, which acts as a "templating factor" for ice nucleation. This results on an identity between the homogeneous and the heterogeneous work of nucleation (Eq. 35) implying that by knowing an expression for $\Delta G_{\text{hom}}$, $\Delta G_{\text{het}}$ can be readily written. This is advantageous as homogeneous ice nucleation is far better understood than immersion ice nucleation, and, because it avoids a mechanistic description of the complex interaction

between the particle, the ice and the liquid. To describe $\Delta G_{\text{hom}}$ the NNF framework (Barahona, 2014) was employed. This approach was extended to include non-equilibrium dissipation effects.

A model to describe the effect of the immersed particle on the mobility of water molecules, hence on the kinetics of immersion freezing, was also developed. This model builds upon an expression for the interfacial diffusion flux that accounts for

the work required for water molecules to accommodate in an ice-like manner during interface transfer. Here this expression is extended to account for the effect of the particle on the molecular flux to the ice germ. It was shown that $J_0$ stronlgy decreases as the system moves towards thermodynamic equilibrium.

The model presented here suggests the existence of a spinodal regime in ice nucleation where a pair of molecules with orientation similar to that of bulk ice may be enough to trigger freezing. Ice nucleation in the spinodal regime requires a highly efficient templating effect by the particle, however also tends to be strongly limited by the kinetics of the ice-liquid interfacial transfer. Compared to the classical germ-forming regime, nucleation by a spinodal mechanism is much more limited by diffusion and exhibits a more moderate increase in $J_{\text{het}}$ as temperature decreases. The existence of two nucleation regimes and the strong kinetic limitations occurring in efficient INP imply that the freezing temperature is an ambiguous measure of ice nucleation activity. This is because for a given $T$ two INP characterized by different $\zeta$ may have the same $J_{\text{het}}$, although with very different sensitivity to surface area and cooling rate.

The relationship between the measured shift in water activity $\Delta a_{\text{w, het}}$ and $\zeta$ was analyzed. It was shown that the proposed model leads directly to the derivation of the so-called water activity criterion for heterogeneous ice nucleation. The concept of "thermodynamic freezing temperature", $T_{\text{ft}}$ was introduced and defined as the highest temperature where it is likely to observe ice nucleation for a given thermodynamic state. $T_{\text{ft}}$ is useful in analyzing how changes in the thermodynamic environment around the droplet affect ice nucleation, independently of the freezing kinetics.

The theory presented here was tested using data for humic-like substances. It was found that assuming a fixed water activity shift to predict $J_{\text{het}}$ could be appropriate for low $\zeta$ as found in Leonardite (the germ-forming regime), however may lead to overprediction of $-\frac{d \ln J_{\text{het}}}{dT}$ for the high $\zeta$ characterizing Pawokee Peat INP. This is because the water activity criterion represents a thermodynamic relation between $a_{\text{w}}$ and $T_{\text{f}}$ but does not account for kinetic limitations which may be significant in spinodal ice nucleation.

Immersion freezing research has seen a resurgence during the last decade (DeMott et al., 2011). A wealth of data is now available to test theories and new approaches to describe ice formation in atmospheric models. To effectively doing so it is necessary to develop models that realistically capture the complexities of the liquid phase. Further development of the approach presented here will look to better describe the non-reversible aspects of nucleation as well as to establish a more complete description of the properties of the vicinal water. Application to the freezing of atmospheric aerosol requires the definition of the ice nucleation spectrum, which will be pursued in a future work. Nevertheless, the present study constitutes for first the time an approximation to the modeling of ice nucleation that links the modifications of the properties of vicinal water by immersed particles with their ice nucleation ability. The approach presented here may help expanding our understanding of immersion ice nucleation and facilitate the interpretation of experimental data in situations where current models fall short. Application of these ideas in cloud models will allow elucidating under what conditions different nucleation regimes occur in the atmosphere.

*Acknowledgements.* Donifan Barahona was supported by the NASA Modeling and Analysis Program, grant: 16-MAP16-0085.

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

**Table 1.** List of symbols.

| | |
|---|---|
| $a_0$ | Cross-sectional area of a water molecule, $\pi d_0^2/4$, m$^2$ |
| $A_w$ | Phenomenological interaction parameter |
| $a_w$ | Activity of water |
| $a_{w,\,eff}$ | Effective water activity |
| $a_{w,eq}$ | Equilibrium $a_w$ between bulk liquid and ice (Koop and Zobrist, 2009) |
| $a_{w,\,het}$ | Thermodynamic freezing threshold for heterogeneous ice nucleation |
| $a_{w,\,hom}$ | Thermodynamic freezing threshold for homogeneous ice nucleation |
| $C_0$ | Monomer concentration, m$^{-2}$ |
| $E, T_0$ | Parameters of the VFT equation defining $D_\infty$, 892 and 118 K, respectively (Smith and Kay, 1999) |
| $D$ | Diffusion coefficient for interface transfer, m$^2$ s$^{-1}$ |
| $D_\infty$ | Self-diffusion coefficient of bulk water (Smith and Kay, 1999), m$^2$ s$^{-1}$ |
| $D_0$ | Fitting parameter, $3.06 \times 10^{-9}$ m$^2$ s$^{-1}$ (Smith and Kay, 1999) |
| $d_0$ | Molecular diameter of water, $(6v_w/\pi)^{1/3}$, m |
| $f_{het}^*$ | Impingement factor for heterogeneous ice nucleation, s$^{-1}$ |
| $f_{hom}^*$ | Impingement factor for homogeneous ice nucleation, s$^{-1}$ |
| $G$ | Gibbs free energy, J |
| $h$ | Planck's constant, Js |
| $J_0$ | Preexponential factor m$^{-2}$ s$^{-1}$ |
| $J_{het}$ | Heterogeneous nucleation rate, m$^{-2}$ s$^{-1}$ |
| $k_B$ | Boltzmann constant, J K$^{-1}$ |
| $N$ | Number of clustering molecules in LL and IL regions, 6 (Holten et al., 2013) |
| $n$ | Number of molecules in a ice cluster |
| $n^*$ | Critical germ size |
| $n_{het}$ | Critical germ size for heterogeneous ice nucleation |
| $n_{hom}$ | Critical germ size for homogeneous ice nucleation |
| $n_t$ | Number of formation paths of the transient state, 16 (Barahona, 2015) |
| $p_{s,w}, p_{s,i}$ | Liquid water and ice saturation vapor pressure, respectively, Pa (Murphy and Koop, 2005) |
| $s$ | Geometric constant of the ice lattice, $1.105$ molec$^{1/3}$ (Barahona, 2014) |
| $S_i$ | Saturation ratio with respect to ice |
| $S_{c,0}$ | Configuration entropy of water* |
| $S_c$ | Configuration entropy of vicinal water |
| $T$ | Temperature, K |
| $T_c$ | Critical separation temperature, 211.473 K |

| | |
|---|---|
| $v_{\mathrm{w}}$ | Molecular volume of water in ice (Zobrist et al., 2007), $\mathrm{m}^{-3}$ |
| $v_{\mathrm{w},0}$ | Molecular volume of water at 273.15 K |
| $\bar{W}$ | Average transition probability in water |
| $W_{\mathrm{diss}}$ | Work dissipated during cluster formation, J |
| $W_{\mathrm{d}}$ | Work dissipated during interface transfer, J |
| $Z$ | Zeldovich factor |
| $\Delta a_{\mathrm{w,\,het}}$ | $a_{\mathrm{w,\,het}} - a_{\mathrm{w,eq}}$ |
| $\Delta a_{\mathrm{w,\,hom}}$ | $a_{\mathrm{w,\,hom}} - a_{\mathrm{w,eq}}$, 0.304 (Koop et al., 2000; Barahona, 2014) |
| $\Delta G$ | Work of cluster formation, J |
| $\Delta G_{\mathrm{act}}$ | Activation energy for ice nucleation, J |
| $\Delta G_{\mathrm{hom}}$ | Nucleation work for homogeneous ice nucleation, J |
| $\Delta G_{\mathrm{het}}$ | Nucleation work for heterogeneous ice nucleation, J |
| $\Delta h_{\mathrm{f}}$ | Heat of solidification of water, $\mathrm{J\,mol}^{-1}$ (Barahona et al., 2014; Johari et al., 1994) |
| $\Delta \mu_{\mathrm{s}}$ | Excess free energy of solidification of water, J |
| $\Delta \mu_{\mathrm{i}}$ | Driving force for ice nucleation, J |
| $\Lambda_{\mathrm{mix}}$ | Dimensionless mixing parameter, defined in Eq. (30) |
| $\Phi$ | Energy of formation of the ice-liquid interface, $\mathrm{molec}^{1/3}$ J |
| $\Gamma_{\mathrm{w}}$ | Molecular surface excess of at the interface, 1.46 (Barahona et al., 2014; Spaepen, 1975) |
| $\mu_{\mathrm{w}}, \mu_{\mathrm{s}}, \mu_{\mathrm{vc}}$ | Chemical potential of water, ice and vicinal water, respectively J |
| $\rho_{\mathrm{w}}, \rho_{\mathrm{i}}$ | Bulk density of liquid water and ice, respectively, $\mathrm{Kg\,m}^{-3}$ (Pruppacher and Klett, 1997) |
| $\sigma_{\mathrm{E}}$ | Dimensionless residual entropy |
| $\sigma_{\mathrm{iw}}$ | Ice–liquid interfacial energy $\mathrm{J\,m}^{-2}$ (Barahona et al., 2014) |
| $\theta$ | Contact angle |
| $\zeta$ | Templating factor |
| $\Omega_{\mathrm{g}}$ | Ice germ surface area, $\mathrm{m}^{-2}$ |

* From the data of Scala et al. (2000) the following fit was obtained:

$S_{\mathrm{c},0} = k_{\mathrm{B}} v_{\mathrm{w}}/v_{\mathrm{w},0}(-7.7481 \times 10^{-5} T^2 + 5.5160 \times 10^{-2} T - 6.6716)$ $(\mathrm{J\,K}^{-1})$ for $T$ between 180 K and 273 K.

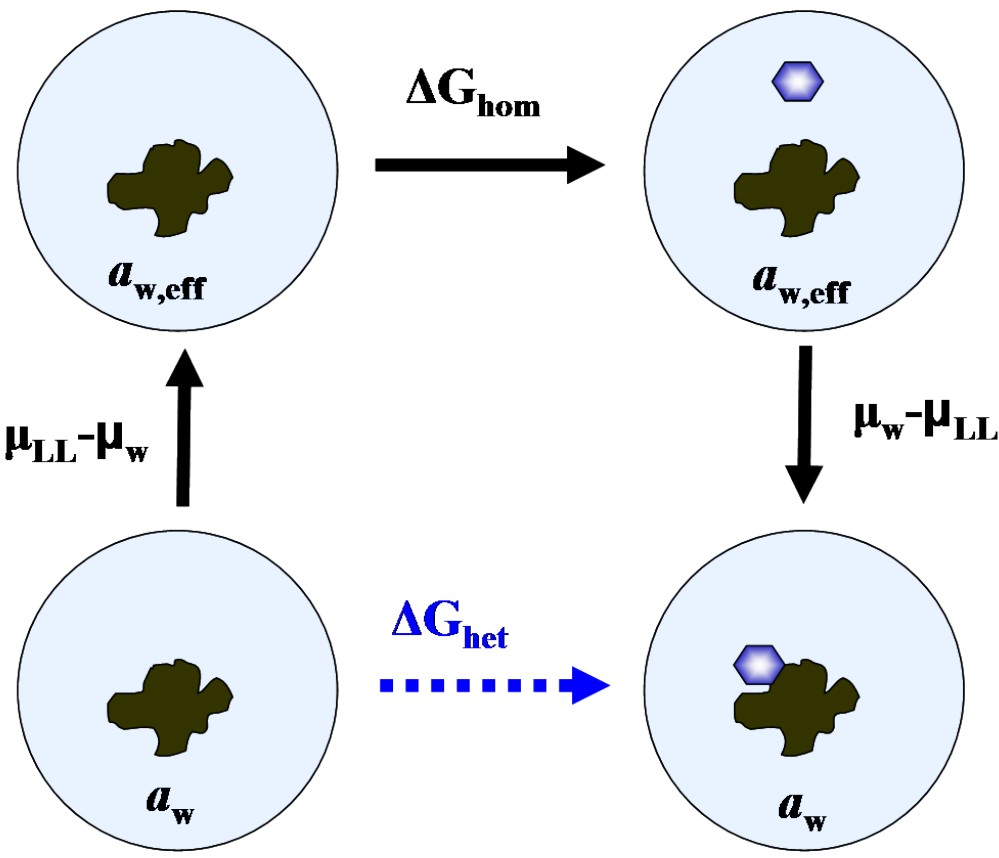

**Figure 1.** Diagram representing a thermodynamic path including homogeneous ice nucleation with the same work as heterogeneous freezing.

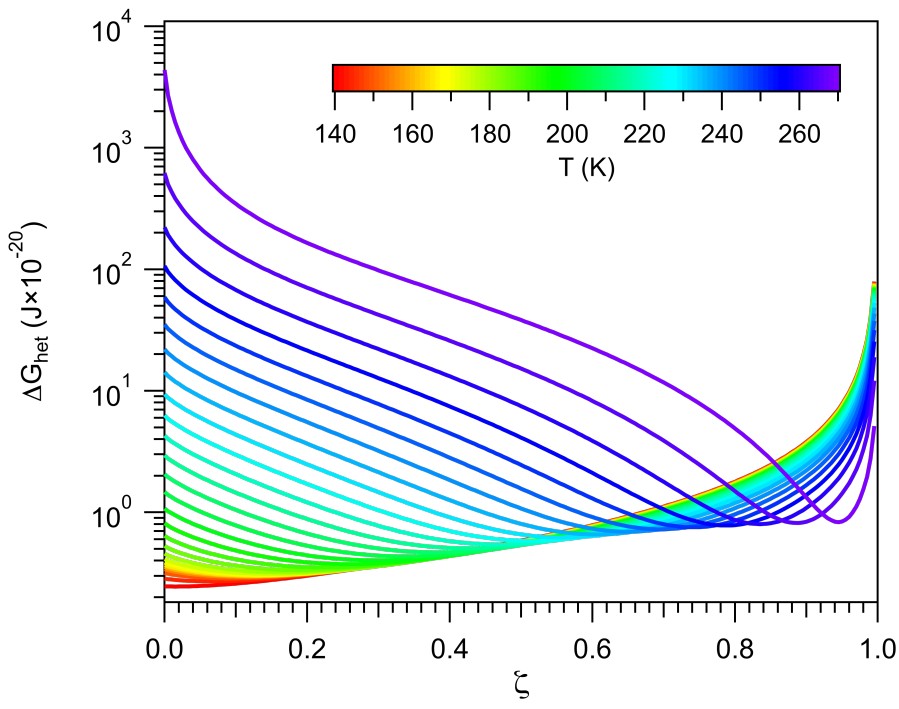

**Figure 2.** Work of heterogeneous ice nucleation. Color indicates different temperatures.

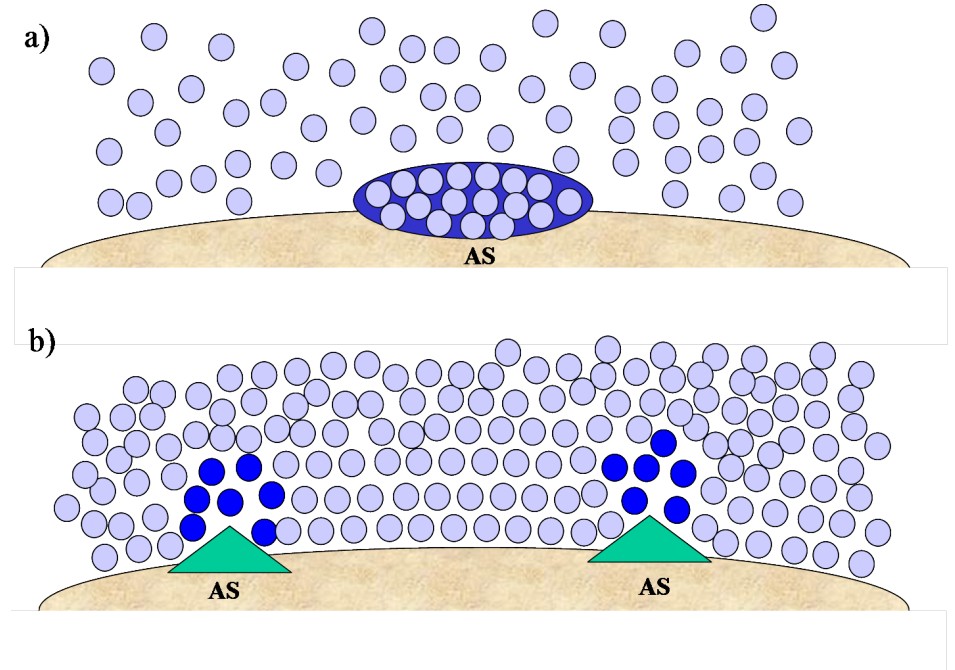

**Figure 3.** Different representations of immersion freezing. (a) An ice germ (dark blue) forming on an active site (AS) by random collision of water molecules (light blue). (b) Low density regions (dark blue) forming in the vicinity of active sites within a dense liquid phase (light blue).

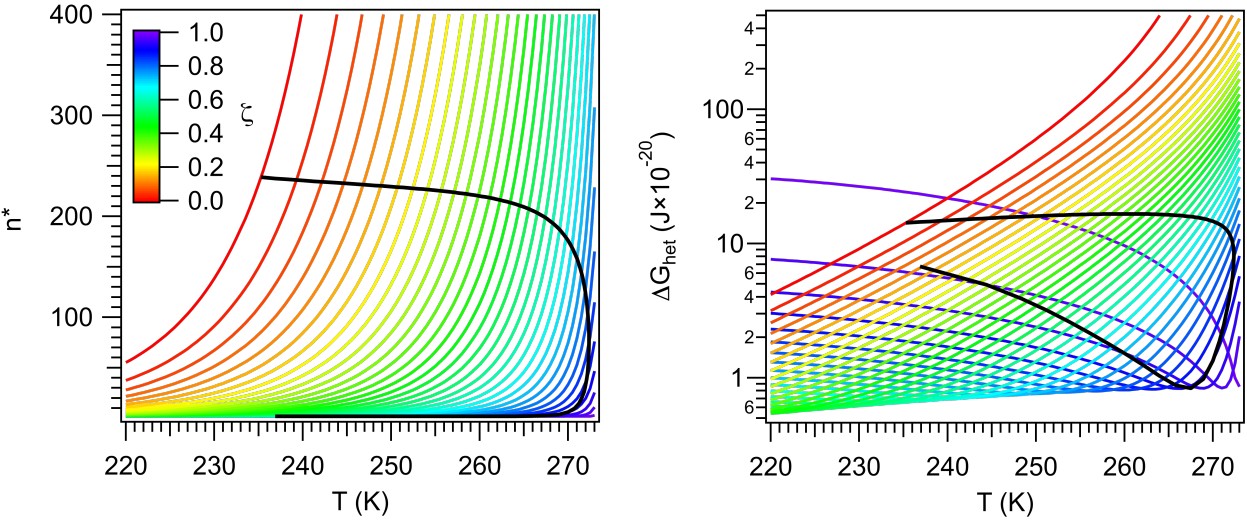

**Figure 4.** Critical germ size (left panel) and work of heterogeneous ice nucleation (right panels) for different values of $\zeta$ (color). Black lines correspond to constant $J_{\text{het}} = 10^6 \ \text{m}^{-2} \ \text{s}^{-1}$.

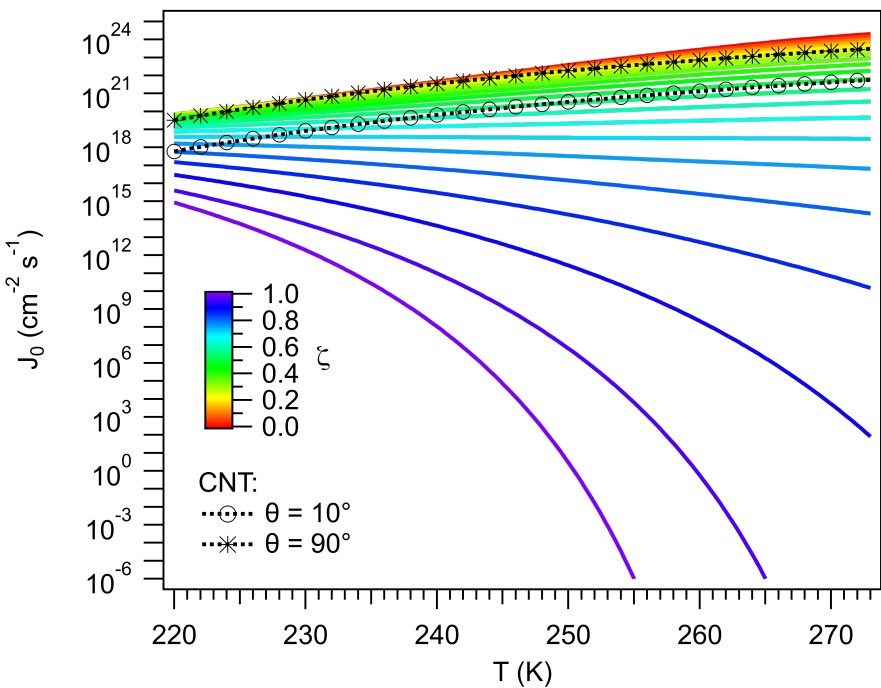

**Figure 5.** Preexponential factor. Colored lines indicates different values of $\zeta$. Black lines correspond to results calculated using CNT for different values of the contact angle, $\theta$.

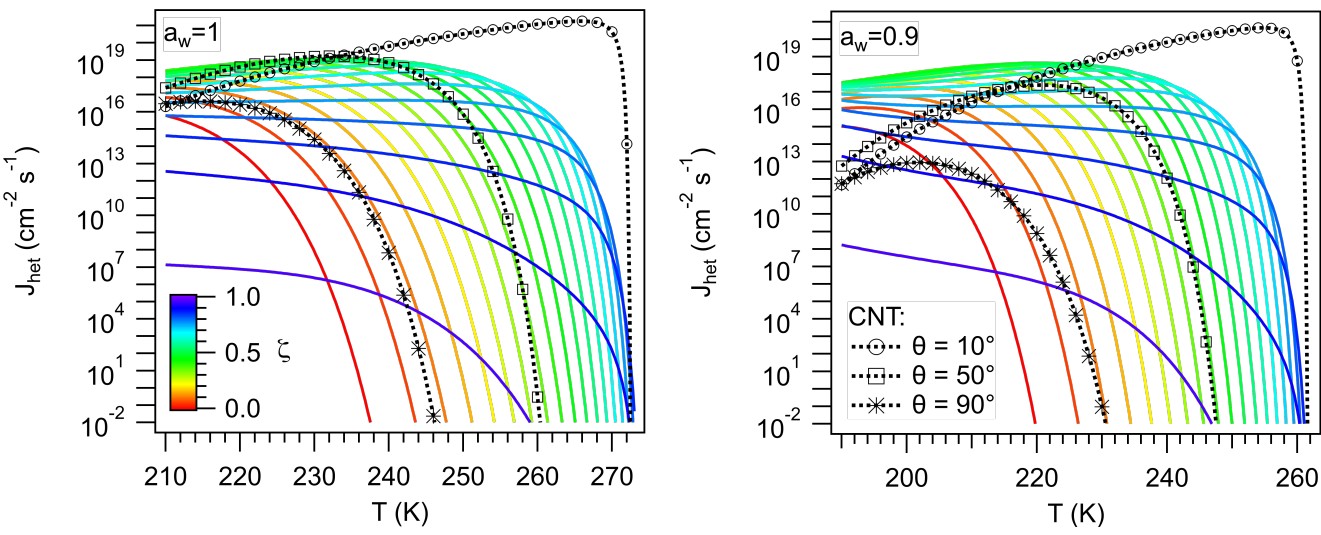

**Figure 6.** Ice nucleation rate calculated using Eq. (61) for different values of $\zeta$ (color). Black lines were calculated using CNT for different values of the contact angle, $\theta$.

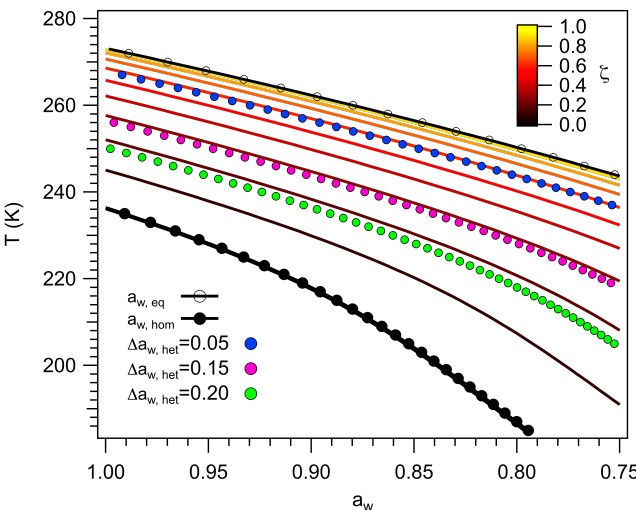

**Figure 7.** Thermodynamic freezing temperature as a function of water activity. Colored lines correspond to $T_{ft}(a_w = a_{w,het})$ for different values of $\zeta$. Also shown are the water activities at equilibrium and at the homogeneous freezing threshold, $a_{w,eq}$ and $a_{w,hom}$, respectively, and lines drawn applying constant water ativity shifts, $\Delta a_{w,het}$, of $0.05$, $0.15$ and $0.20$.

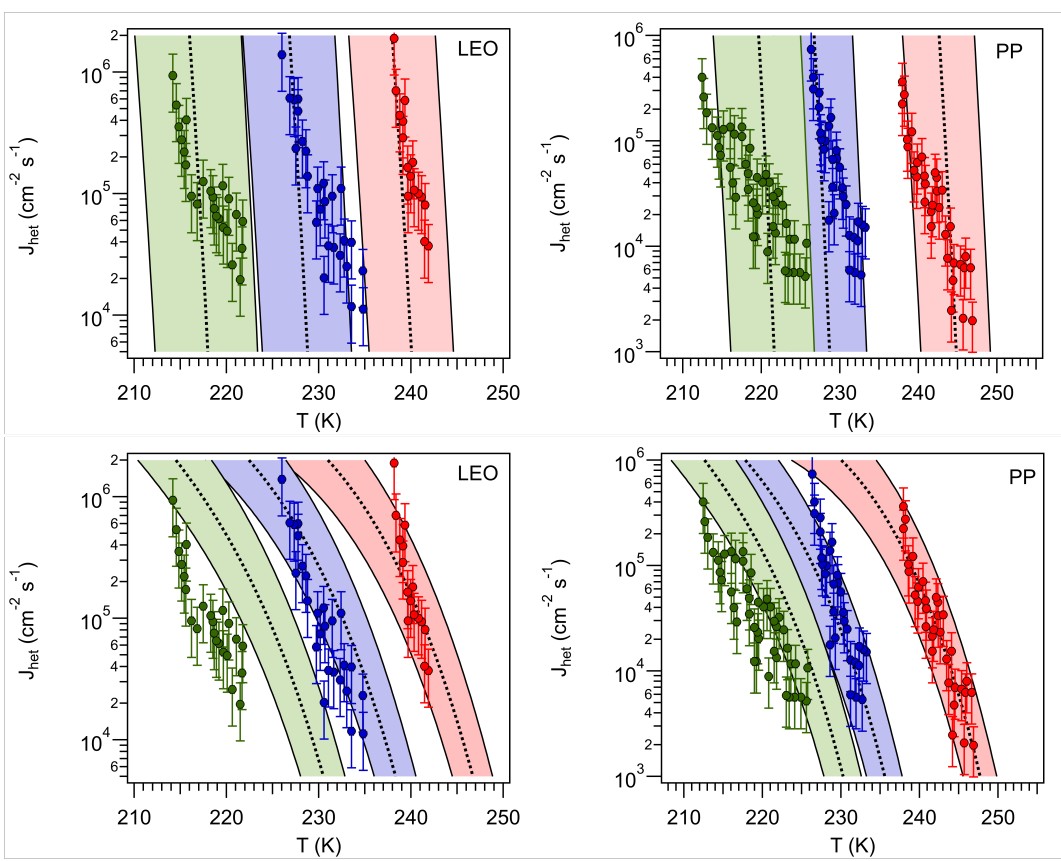

**Figure 8.** Top panels: Heterogeneous ice nucleation rate calculated using a constant shift in $a_{\mathrm{w}}$ (black, dotted, lines) for Leonardite (LEO $\Delta a_{\mathrm{w, het}} = 0.2703$) and Pawokee Peat (PP, $\Delta a_{\mathrm{w, het}} = 0.2466$) (top panels). Red, blue and green colors correspond to $a_{\mathrm{w}}$ equal to 1.0, 0.931 and 0.872, respectively, for LEO and 1.0, 0.901 and 0.862 for PP. Shaded area corresponds to $\Delta a_{\mathrm{w, het}} \pm 0.025$. Markers correspond to experimental measurements reported by Rigg et al. (2013); error bars represent an order of magnitude deviation from the reported value. Bottom panels: $J_{\mathrm{het}}$ calculated for constant $\zeta = 0.949$ for LEO and $\zeta = 0.952$ for PP. The shaded area corresponds to $a_{\mathrm{w}} \pm 0.01$ and $\zeta \pm 0.0015$.