# Peer review of "On the Thermodynamic and Kinetic Aspects of Immersion Ice Nucleation"

_Atmospheric Chemistry and Physics, 2017_

## Referee Comment (RC1) · Anonymous Referee #3 · 23 Mar 2018

This paper proposes a new theoretical model for immersion nucleation, by investigating the thermodynamic and kinetic impact of the solid particle on near-by water molecules and its consequences for ice nucleation within the liquid droplet.

Although immersion freezing is one of the main pathways of ice formation in the atmosphere, it is still poorly understood and the topic addressed in the paper is of great relevance for cloud physics. Furthermore, the paper puts together an important number of previous works in an attempt to make progress on our understanding of immersion nucleation. It is overall rather clearly written and the reasoning is supported by high quality figures and schematics. This paper could hence be an appropriate contribution to ACP. However, I believe there are shortcomings in the theoretical derivation and its presentation that should be resolved before the paper can be considered for publication. I therefore recommend major revisions of the current manuscript. In the following I will explain my concerns in more detail.

Major points :

1) *Presentation of the theoretical development*: I am not a specialist of ice nucleation and the related thermodynamics and kinetics. However, this will be the case for other ACP readers who would like to use the results presented in the paper. Since the theoretical derivation mainly consists in chemical physics, one possibility would be that the author submits this study to another journal, such as "The Journal of Chemical Physics". If the author chooses to present this work in ACP, I think some significant efforts should be spent in order to make the paper more accessible to the bulk of ACP readers. In particular, I think the organization of the derivation could be improved in that regard.

Indeed, most ACP readers will be interested in the derived nucleation rate for immersion nucleation. Thus, I would start the theoretical section with the general expression for the nucleation rate, i.e. the product of the concentration of critical clusters $c_g = C_0 \exp(-\Delta G/k_B T)$, corrected by the Zeldovich factor times the flux of water molecules towards those clusters $F_w$:

$$J_{het} = Z c_g F_w$$

Thermodynamic effects of the particle on vicinal water affect $c_g$ and $Z$ (through $\Delta G$, the nucleation barrier for critical germ size) while kinetic effects affect $F_w$ (the flux of water molecules towards the ice germs). After stating this, I would then elaborate on how expressions for the different factors are obtained in the new theory. This is mainly a change in presentation: most of the content is already present in the paper, but it should be made clearer where the derivation is going, e.g. when reading section 2.3 the reader sometimes misses the goal of the development which is only made clear in section 2.4.
2) *Comparison with the classical theory of nucleation*: The main point of the paper is to take into account the change in the thermodynamic and "dynamic" properties of vicinal water near the immersed solid particle and the impact on ice nucleation. In that sense, it differs from the classical nucleation theory (CNT) which rather considers the influence of the solid particle-liquid water interface directly. Although the CNT expression for the nucleation rate is recalled in section 2.2, it is not really contrasted with the new theory. I miss a more thorough discussion comparing the different expressions and hypotheses between the theory introduced here and CNT. In particular, a table comparing the CNT and new theory expressions for the different factors in $J_{het}$ would be useful. I would suggest to add a dedicated section on that point in the discussion (and remove section 2.2).

3) *Contents*: This is another reason for my reservations. On several instances, I have noticed algebra mistakes which are repeated in several formulas. This casts some doubts on the whole theoretical derivation and it is unclear without repeating all the work whether the related figures are correct or not. Since the theoretical derivation is central to the paper, it is essential that the author makes sure all the formulas are correct (and convinces the reviewer). References to previous studies should also be made as explicit as possible, to make the argument easier to follow. I list below the main two mistakes I have noticed:

- page 6, eq (7), (8) and (10): if the $g^E$ term represents an excess energy imposing a penalty to mixing (and representing the tendency of IL and LL regions to cluster), it should be positive: $g^E = +A_w\zeta(1-\zeta)$ with $A_w = 2k_BT_c/N$. In the current formulation the first part of Eq. (9), i.e. $\frac{\partial^2\mu_{vc}}{\partial\zeta^2} = 0$ does not hold at $\zeta = 0.5$. Some of the following equations build on this result (among which Eqs (12), (13), ...). In Eq. (17) $\Lambda_E$ should be $\Lambda_E = \frac{2}{N}\frac{T_c}{T}$. Because of this error, the current Eq. (12) disagrees with Eq. (8) in Holten et al. (2013).

- page 7, Eq. (17): I have: $a_w = (a_{w,eff})^{\frac{1}{1+\zeta}}(a_{w,eq})^{\frac{\zeta}{1+\zeta}}\exp(\Lambda_E\zeta\frac{1-\zeta}{1+\zeta})$ instead of the

formula of the author. This formula is used in many instances, for example in Eqs. (19), (20), (21), (22), and (45)

**Specific comments:**

1. suggest to change the title from "On the Thermodynamic and Dynamic Aspects of Immersion Ice Nucleation" to "On the Thermodynamic and Kinetic Aspects of Immersion Ice Nucleation"; the "dynamic" aspects that the author refers to are related to the diffusion of water molecules in the fluid and in that sense could be referred to as "kinetic" (dynamic brings fluid dynamics to mind)

2. p 4, Eq (1) differs from the common expression in Prupaccher and Klett, which also includes $N_c \Omega$, the number of water molecules in contact with the cluster

3. p 5, Eqs (4), (5) and (6): it is more common to define an increasing entropy upon mixing. Thus, in both equations (4) and (5) it might be clearer to add a minus sign in front of $T\Delta S_{mix}$ in Eq. (4) and (5) and after "=" in Eq (6). This has no impact on the subsequent equations, but would be more consistent with the usual conventions

4. p 10, Eq (24): I am not convinced that with that definition, $\Delta\mu_i$ should be referred to as supersaturation

5. p 10, line 9: unit of $s$ should be $molec^{\frac{1}{3}}$ so that the units match in Eqs (23) and (25)

6. p 10, Eq 27: should be $n^{-\frac{4}{3}}$ rather than $n^{-\frac{1}{3}}$

7. p 11, line 2: specify here again the condition for mechanical equilibrium

8. p 12, Eq (22) & l 25: here $C_0$ seems to be the monomer concentration per surface unit of the particle (and not in a volume of fluid), but this is only mentioned after Eq (44) where it is specified that $C_0 = a_0^{-1}$ where $a_0$ is the cross-sectional area of a water molecule. This should already be written line 25. The numerical value of $a_0$ (or a formula) should be mentioned.

   Furthermore, it is surprising that the author takes $C_0 = a_0^{-1}$. This implies that only the molecules in direct contact with the particle are considered as vicinal water susceptible to grow into ice germs. This contradicts the motivation for the development expressed, e.g. p3, l13-14: "In a groundbreaking work, Anderson (1967) found strong evidence of ice formation several molecular diameters away from the clay-water interface." The author should at least comment on that.

9. p 12, Eq (33): It is not clear to me how the author comes up with that expression for the Zeldovich factor in this case, especially with $n^* = n_{hom} + 2$. I would rather obtain:

$$Z = \left[ -\frac{\frac{\partial^2 \Delta G}{\partial n^2}\big|_{n=n^*}}{2\pi k_B T} \right]^{\frac{1}{2}} = \left[ \frac{\Delta G_{het}}{3\pi k_B T n^*(n^* + 2)} \right]^{\frac{1}{2}} \tag{1}$$

   with $n^* = [n^*_{hom}]_{a_{w,eff}}$. The derivation should be briefly explained.

10. Section 2.4: please be more specific in this section regarding which assumptions have been previously made in the literature and which are introduced in this paper. Beyond the suggestions above for Sect.2, the presentation of this subsection on kinetics could be improved; e.g. I would put the text from l 24 p13 to l 1 p14 before Eq (35) since it provides some justification for the linear scaling introduced in Eq. (35)

11. p 16, l 28-32: Section 3.1 Please give a mathematical definition of the freezing temperature . The current definition is not very clear, the term "equilibrium temperature" suggests thermodynamic equilibrium between ice and liquid water,

whereas nucleation is a kinetic process. I am not convinced, given the informa-
tion in the next paragraph, that this $T_{f,eq}$ can be referred to as an equilibrium
temperature. In the legend and ylabel of Fig. 4, please add the symbol $T_{f,eq}$.

12. p 21 l 22: "regular solution" -> mixture ?

13. Table 1: when relevant, the numerical values (or the expressions) of the quantities
corresponding to the symbols should be added there, and the books/papers from
which the estimates are taken should be referenced. For instance, the value of
$a_0$ is not given. The units should always be specified (e.g. the cooling rate has
no units). Also note that the unit "$mol$" is different from molecule and one should
rather write "$molec$"

**References**

Holten, V., Limmer, D. T., Molinero, V., and Anisimov, M. A.: Nature of the anomalies in the
supercooled liquid state of the mW model of water, The Journal of Chemical Physics, 138,
174 501, doi:10.1063/1.4802992, https://doi.org/10.1063/1.4802992, 2013.

---

## Referee Comment (RC2) · Anonymous Referee #1 · 25 Mar 2018

Review of
**On the Thermodynamic and Dynamic Aspects of Immersion Ice Nucleation**
by D. Barahona

**General comment:**
In this manuscript the role of different ordered structures of water close and far from an immersed particle is investigated. A theory of immersion freezing based on these different states is derived. The theoretical investigations are compared to real measurements of heterogeneous nucleation rates in different experiments. Since ice nucleation in general, and especially heterogeneous nucleation is not well understood and the theoretical investigations are not convincing at the moment, a theory based on thermodynamics of water is a very interesting step for improving our knowledge of heterogeneous ice nucleation. Thus, in general this is a valid contribution for Atmospheric Chemistry and Physics.

However, before the manuscript can be accepted, some issues has to be clarified. Therefore I recommend major revisions of the manuscript. In the following I will explain my concerns in details:

**Major points**

1. Representation of the theory:
   The topic of ice nucleation is quite complicated and usually only classical nucleation theory or some additional topics are well known in the ice cloud community, whereas the more detailed thermodynamic basis is usually hidden in many discussions. In this study, the author has to present details for the development of the theory but also has to make sure that the reader can follow his line of arguments. It would be very helpful if the author would present a kind of roadmap at the very beginning to describe what he wants to derive finally and which steps will be necessary in order to do so. Otherwise the reader is really lost in details, which stem either from standard thermodynamic arguments or are of phenomenological type.

2. Derivation of equation (17)
   I could not reproduce the central equation (17) in the form the author did, I ended with the expression

$$a_w = a_{w,eff} \left( \frac{a_{w,eq}}{a_{w,eff}} \right)^{\frac{\zeta}{1+\zeta}} \exp \left( \Lambda_E \zeta \frac{1-\zeta}{1+\zeta} \right) \qquad (1)$$

   This is crucial, since the equation is often used in the following derivation. For instance, I have several reservations about equation (19), since the limit $\zeta \to 1$ is not well defined. The author has to check his derivation of equation (17) and, if necessary also the derivation of the subsequent theory. In section 2.3.3 the model is extended to the spinodal limit and the limit $\zeta \to 1$ is investigated, which is unbounded in the current representation, but probably not for the derivation I have found. Thus, it is not clear to me if the discussion in this section still holds.

**Minor points:**

1. Could you explain the sign of the excess energy $g^E = -A_w \zeta (1 - \zeta)$? What is the thermodynamic reason for this choice?

2. What are the thermodynamic conditions for the derivation of the critical temperature, i.e. where do the conditions $\frac{\partial^2 \mu_{vc}}{\partial \zeta^2} = 0$ and $\frac{\partial^3 \mu_{vc}}{\partial \zeta^3} = 0$ come from? Please explain this shortly in the text.

3. In section 2.3.2 the water activity shift for heterogeneous nucleation is derived from the theory. Could you compare this results also numerically with the use of a constant shift in actual parameterisations and comment this? How large is $\zeta$ for the usual parameterisations?

4. In figure 4 different curves of water activity are shown. As far as I understand, the colors (dark red to yellow) indicate different versions of the new theory $(a_{w,het})$. Thus, the label $a_{w,het}$ as red in the diagram is misleading.

---

## Author Comment (AC1) · 13 May 2018

**1   Response to Referee 1**

**Reviewer:** *In this manuscript the role of different ordered structures of water close and far from an immersed particle is investigated. A theory of immersion freezing based on these different states is derived. The theoretical investigations are compared to real measurements of heterogeneous nucleation rates in different experiments. Since ice nucleation in general, and especially heterogeneous nucleation is not well understood and the theoretical investigations are not convincing at the moment, a theory based on thermodynamics of water is a very interesting step for improving our knowledge of het-erogeneous ice nucleation. Thus, in general this is a valid contribution for Atmospheric*

*Chemistry and Physics. However, before the manuscript can be accepted, some is-sues has to be clarified. Therefore I recommend major revisions of the manuscript. In the following I will explain my concerns in details.*

**Response:** I thank the reviewer for the comments on the manuscript. They are ad-dressed in detail below.

**Major Points**

**Reviewer:** *1. Representation of the theory: The topic of ice nucleation is quite com-plicated and usually only classical nucleation theory or some additional topics are well known in the ice cloud community, whereas the more detailed thermodynamic basis is usually hidden in many discussions. In this study, the author has to present details for the development of the theory but also has to make sure that the reader can follow his line of arguments. It would be very helpful if the author would present a kind of roadmap at the very beginning to describe what he wants to derive finally and which steps will be necessary in order to do so. Otherwise the reader is really lost in details, which stem either from standard thermodynamic arguments or are of phenomenological type.*

**Response:** The revisited paper has been reorganized to clarify the approach. Specif-ically, the calculation of the nucleation rate is now the central theme of the theoretical derivation. Additional explanation has also been included contrasting the classical ap-proach and the proposed theory. The derivation of the equation of state of vicinal water was reworked to make it clearer and correct errors/typos. Finally, the Section on kinet-ics has been reorganized introducing general concepts earlier in the text. This has made the revisited paper much more readable.

**Reviewer:** *Derivation of equation (17). I could not reproduce the central equation (17)*

*in the form the author did, I ended with the expression*

$$a_w = a_{\text{w, eff}}^{\frac{1}{1+\zeta}} a_{w,eq}^{\frac{\zeta}{1+\zeta}} \exp\left(\Lambda_E \frac{1-\zeta}{1+\zeta}\right) \qquad (1)$$

*This is crucial, since the equation is often used in the following derivation.*

**Response:** The wrong expression for $\Delta\mu_{\text{S}}$ was written in the text. $\Delta\mu_{\text{S}}$ must actually be calculated at $a_{\text{w, eff}}$, hence Eq. (14) of should read:

$$\Delta\mu_{\text{S}} = -k_{\text{B}}T\ln\left(\frac{a_{\text{w, eff}}}{a_{\text{w, eq}}}\right), \qquad (2)$$

After introducing this equation into Eq.(13) it can be readily seen that Eq. (17) of the original paper is correct. Equation (14) was also used to simplify Eq. (40); this has been corrected as well. The derivation of Eq. (2) is shown at the end of this document (Eq. 10).

**Reviewer:** *For instance, I have several reservations about equation (19), since the limit $\zeta \to 1$ is not well defined. The author has to check his derivation of equation (17) and, if necessary also the derivation of the subsequent theory. In section 2.3.3 the model is extended to the spinodal limit and the limit $\zeta \to 1$ is investigated, which is unbounded in the current representation, but probably not for the derivation I have found. Thus, it is not clear to me if the discussion in this section still holds.*

**Response:** The new derivation at the end of this document shows more clearly that $a_{\text{w, eff}}$ is in fact bounded for $\zeta \to 1$. From Eq.(18),

$$a_{\text{w}} = a_{\text{w, eff}}\left(\frac{a_{\text{w, eq}}}{a_{\text{w, eff}}}\right)^{\zeta}\exp(\Lambda_{\text{mix}}) \qquad (3)$$

and from Eq.(19)

$$a_{\text{w, eff}} = \left( \frac{a_{\text{w}}}{a_{\text{w, eq}}^{\zeta}} \right)^{\frac{1}{1-\zeta}} \exp\left( -\frac{\Lambda_{\text{mix}}}{\zeta - 1} \right). \tag{4}$$

Since $\Lambda_{\text{mix}} = 0$ for $\zeta = 1$, then Eq.(3) implies that $a_{\text{w}} = a_{\text{w, eq}}$ for $\zeta = 1$. Hence from Eq.(4) $a_{\text{w}} = a_{\text{w, eq}} = a_{\text{w, eff}} = 1$, indicating that $\zeta = 1$ corresponds to thermodynamic equilibrium.

**Minor Points**

**Reviewer:** *Could you explain the sign of the excess energy $g^E = -A_w \zeta(1 - \zeta)$? What is the thermodynamic reason for this choice?*

**Response:** The choice was made simply to obtain a positive $T_c$. However as pointed out by other reviewers there is an error in the derivation. Motivated by this I have re-worked the derivation correcting errors and making it more readable. Since the excess term plays a minor role, the correction only resulted in up to two orders of magnitude difference in $J_{\text{het}}$, but following essentially the same behavior. The new derivation is presented at the end of this document and it is now included in the revised paper.

**Reviewer:** *What are the thermodynamic conditions for the derivation of the critical temperature, i.e. where do the conditions $\frac{\partial^2 \mu_{\text{VC}}}{\partial \zeta^2} = 0$, $\frac{\partial^3 \mu_{\text{VC}}}{\partial \zeta^3} = 0$ come from? Please explain this shortly in the text.*

**Response:** A solution would split into two phases if by doing so lowers its Gibbs free energy (Prausnitz et al. (1998), c.f. Section 6.12). For a metastable solution $\mu_{\text{VC}}$ must be locally minimal, hence $\frac{\partial \mu_{\text{VC}}}{\partial \zeta} = 0$. The condition $\frac{\partial^2 \mu_{\text{VC}}}{\partial \zeta^2} < 0$ indicates that any

increase in $\zeta$ increases $\mu_{\mathrm{VC}}$ (i.e., the curve $\mu_{\mathrm{VC}}$ vs. $\zeta$ becomes concave downward) such that it is thermodynamically more favorable for the solution to split into distinct phases than to increase its concentration; $\frac{\partial^2 \mu_{\mathrm{VC}}}{\partial \zeta^2} = 0$ thus limits the metastable region. The last condition, $\frac{\partial^3 \mu_{\mathrm{VC}}}{\partial \zeta^3} = 0$, indicates that the metastable region reduces to a single point and that there is a single critical temperature $T_c$ for a regular solution.

The explanation above has been introduced in the text.

**Reviewer:** *In section 2.3.2 the water activity shift for heterogeneous nucleation is derived from the theory. Could you compare this results also numerically with the use of a constant shift in actual parameterisations and comment this? How large is $\zeta$ for the usual parameterisations?*

**Response:** Since the mixing term is typically small, $\zeta \approx 1 - \frac{\Delta a_{\mathrm{w, het}}}{\Delta a_{\mathrm{w, hom}}}$, hence $0 < \zeta <\sim$ $1$ (the upper limit is somewhere around $.96$ due to mixing effects). This relationship is true only in the germ-forming regime where $J_{\mathrm{het}}$ is mainly dictated by thermodynamics. Kärcher and Lohmann (2003) suggested the approximation $J_{\mathrm{het}} \approx J_{\mathrm{hom}}[f(\Delta a_{\mathrm{w, het}})]$. This of course resembles the definition of the nucleation work derived in Section 2.3.1. The revisited paper discusses this in further detail. As expected the two approaches considerable differ for the spinodal regime. It must be noted that Knopf and Alpert (2013) also parameterized $J_{\mathrm{het}}$ as a function of $\Delta a_{\mathrm{w, het}}$. However their expressions are material-specific and comparison against their work is left for future works.

**Reviewer:** *In figure 4 different curves of water activity are shown. As far as I understand, the colors (dark red to yellow) indicate different versions of the new theory (aw,het ). Thus, the label aw,het as red in the diagram is misleading*

**Response:** Corrected.

**2 Corrected derivation of the equation of state of vicinal water**

The vicinal layer is defined as a solution of hypothetical ice-like (IL) and liquid-like (LL) regions, with Gibbs free energy given by

$$\mu_{\mathrm{vc}} = (1-\zeta)\hat{\mu}_{\mathrm{LL}} + \zeta\hat{\mu}_{\mathrm{IL}}, \tag{5}$$

where $\hat{\mu}_{\mathrm{LL}}$ and $\hat{\mu}_{\mathrm{LL}}$ are the chemical potentials of the LL and IL species within the solution, respectively, and $\zeta$ is the fraction of IL regions in the layer. Equation (5) can also be written in terms of the chemical potentials of the "pure" LL and IL species, $\mu_{\mathrm{LL}}$ and $\mu_{\mathrm{IL}}$, respectively, in the form,

$$\mu_{\mathrm{vc}} = (1-\zeta)\mu_{\mathrm{LL}} + \zeta\mu_{\mathrm{IL}} + \Delta G_{\mathrm{mix}} \tag{6}$$

where $\Delta G_{\mathrm{mix}} = (\hat{\mu}_{\mathrm{IL}} - \mu_{\mathrm{IL}})\zeta + (1-\zeta)(\hat{\mu}_{\mathrm{LL}} - \mu_{\mathrm{LL}})$ is the Gibbs energy of mixing. For a mechanical mixture of pure LL and IL species, $\Delta G_{\mathrm{mix}} = 0$, whereas for an ideal solution $\Delta G_{\mathrm{mix}}$ is determined by the ideal entropy of mixing (Prausnitz et al., 1998). Reorganizing Eq. (6) we obtain,

$$\mu_{\mathrm{vc}} = \mu_{\mathrm{LL}} + \zeta\Delta\mu_{\mathrm{il}} + \Delta G_{\mathrm{mix}} \tag{7}$$

where $\Delta\mu_{\mathrm{il}} = \mu_{\mathrm{IL}} - \mu_{\mathrm{LL}}$. $\Delta\mu_{\mathrm{il}}$ can be approximated by using the equilibrium between bulk liquid and ice as reference state so that (Kashchiev, 2000),

$$\mu_{\mathrm{IL}} = \mu_{\mathrm{eq}} + k_{\mathrm{B}}T\ln(a_{\mathrm{IL}}), \tag{8}$$

and
$$\mu_{LL} = \mu_{eq} + k_B T \ln \left( \frac{a_{w,\,eff}}{a_{w,\,eq}} \right), \tag{9}$$

where $a_{w,\,eff}$ is termed the "effective water activity" and it is the value of $a_w$ associated with the LL regions in the vicinal water, and $a_{IL}$ is the water activity in the IL regions. Assuming that similarly to bulk ice the solute does not significantly partition to the IL phase, then $a_{IL} \approx 1$. With this, and combining Eqs.(8) and (9), and rearranging we obtain,

$$\Delta\mu_{il} = -k_B T \ln \left( \frac{a_{w,\,eff}}{a_{w,\,eq}} \right), \tag{10}$$

The central assumption behind Eq. (10) is that $a_{w,\,eq}$ corresponds to the equilibrium water activity between liquid and ice, or in other words that near equilibrium $\Delta\mu_{il} \approx \Delta\mu_s$, being $\Delta\mu_s$ the excess free energy of solidification of water.

In reality $\Delta\mu_s$ corresponds to actual liquid and ice instead of the hypothetical LL and IL substances. This difference can be accounted for by selecting a proper functional form for $\Delta G_{mix}$, for which several empirical and semiempirical interaction models with varying degrees of complexity exist (Prausnitz et al., 1998). In this work it is going to be assumed that the vicinal water can be described as a regular solution. This is the simplest model that accounts for the interaction between solvent and solute during mixing and that is flexible enough to include corrections for the difference between $\Delta\mu_s$ and $\Delta\mu_{il}$. Holten et al. (2013) have shown that a regular solution can reasonably approximate the chemical potential of supercooled water. Moreover, the authors also showed that taking into account clustering of water molecules upon mixing leads to better agreement with MD simulations and experimental results.

According to the regular solution model, modified by clustering (Holten et al., 2013, c.f. Eq. 16),

$$\Delta G_{\text{mix}} = \frac{k_{\mathbf{B}}T}{N}\left[\zeta\ln(\zeta) + (1-\zeta)\ln(1-\zeta)\right] + A_w\zeta(1-\zeta) \tag{11}$$

The first term on the right hand side corresponds to the usual definitioin of the ideal entropy of mixing, i.e., random ideal mixing and a weak interaction between IL and LL regions, modified to account for clustering in groups of $N$ molecules. $N = 6$ corresponds to clustering in hexamers and is near the optimum fit between MD simulations and the solution model (Holten et al., 2013). It must be noted that Holten et al. (2013) recommended an alternative model termed "athermal solution", where nonideality is ascribed to entropy changes upon mixing. In vicinal water some evidence points at nonideality originating from enthalpy changes near the particle (Etzler, 1983), hence a regular solution is more appropriate in this case. For $N = 6$ the difference between the two models is negligible (Holten et al., 2013).

The second term on the right hand side of Eq. (11) is an empirical functional form used to approximate the enthalpy of mixing selected so that $\Delta G_{\text{mix}} = 0$ for $\zeta = 0$ and $\zeta = 1$. $A_w$ is a phenomenological interaction parameter and typically must be fitted to experimental observations. Here it is assumed $A_w$ also implicitly corrects the approximation $\Delta\mu_{\mathrm{il}} \approx \Delta\mu_{\mathrm{s}}$.

An important aspect of the regular solution model is that it predicts that $\Delta G_{\text{mix}}$ (hence $\mu_{\mathrm{vc}}$) has a critical temperature, $T_c$, at $\zeta = 0.5$, defined by the conditions,

$$\frac{\partial^2 \Delta G_{\text{mix}}}{\partial \zeta^2} = 0 \,, \; \frac{\partial^3 \Delta G_{\text{mix}}}{\partial \zeta^3} = 0. \tag{12}$$

Using Eq. (11) into Eq. (12) and solving for $A_w$ gives for $T = T_c$,

$$A_w = \frac{2k_{\mathbf{B}}T_{\mathbf{c}}}{N}. \tag{13}$$

Physically, $T_c$ represents the stability limit of the vicinal water, at which it spontaneously separates into IL and LL regions. Equation (13) thus provides an opportunity to determine $A_w$, since $T_c$ should also correspond to the temperature at which the work of nucleation becomes negligible. This is explored in Section 3.2.

Combining Eqs. (10), (11), and (13), into Eq. (7) we obtain,

$$\mu_{vc} = \mu_{LL} - \zeta k_B T \ln\left(\frac{a_{w,\,eff}}{a_{w,\,eq}}\right) + \frac{k_B T}{N}\left[\zeta \ln(\zeta) + (1-\zeta)\ln(1-\zeta)\right] + \frac{2k_B T_c}{N}\zeta(1-\zeta). \tag{14}$$

Making,

$$\Lambda_{mix} = \frac{1}{N}\left[\zeta \ln(\zeta) + (1-\zeta)\ln(1-\zeta)\right] + \frac{2}{N}\frac{T_c}{T}\zeta(1-\zeta), \tag{15}$$

Equation (14) can be written in the form,

$$\mu_{vc} = \mu_{LL} - \zeta k_B T \ln\left(\frac{a_{w,\,eff}}{a_{w,\,eq}}\right) + k_B T \Lambda_{mix} \tag{16}$$

Equation (16) is the equation of state of vicinal water. It describes the properties of vicinal water in terms of the material-specific parameter $\zeta$, and the interaction parameters $N$ and $T_c$. MD simulations indicate that $N \sim 6$ (Bullock and Molinero, 2013; Holten et al., 2013). $T_c$ is thus the only remaining unknown in Eq. (16) and it is calculated in Section 3.3.

In immersion freezing the particle remains within the droplet long enough that equilibrium is established. This condition is mathematically expressed by the equality, $\mu_{vc} = \mu_w$, where $\mu_w$ is the chemical potential of water in the bulk of the liquid, i.e., away from the particle. Using Eq. (16) this implies,

$$\mu_{\text{w}} = \mu_{\text{LL}} - \zeta k_{\text{B}} T \ln\left(\frac{a_{\text{w, eff}}}{a_{\text{w, eq}}}\right) + k_{\text{B}} T \Lambda_{\text{mix}}. \tag{17}$$

Using again the equilibrium between bulk liquid and ice as reference state, so that $\mu_{\text{w}} = \mu_{\text{eq}} + k_{\text{B}} T \ln(a_{\text{w}})$, and using Eq. (8), Eq. (17) can be written in terms of the water activity in the form,

$$a_{\text{w}} = a_{\text{w, eff}} \left(\frac{a_{\text{w, eq}}}{a_{\text{w, eff}}}\right)^{\zeta} \exp(\Lambda_{\text{mix}}). \tag{18}$$

From Eq. (18) $a_{\text{w, eff}}$ can be readily obtained in the form,

$$a_{\text{w, eff}} = \left(\frac{a_{\text{w}}}{a_{\text{w, eq}}^{\zeta}}\right)^{\frac{1}{1-\zeta}} \exp\left(-\frac{\Lambda_{\text{mix}}}{\zeta - 1}\right). \tag{19}$$

**References**

Barahona, D.: Thermodynamic derivation of the activation energy for ice nucleation, Atm. Chem. Phys., 15, 13 819–13 831, https://doi.org/10.5194/acp-15-13819-2015, http://www.atmos-chem-phys.net/15/13819/2015/, 2015.

Barahona, D.: Analysis of the effect of water activity on ice formation using a new thermodynamic framework, Atm.Chem. Phys., 14, 7665–7680, https://doi.org/10.5194/acp-14-7665-2014, http://www.atmos-chem-phys.net/14/7665/2014/, 2014.

Bullock, G. and Molinero, V.: Low-density liquid water is the mother of ice: on the relation between mesostructure, thermodynamics and ice crystallization in solutions., Faraday Discuss., https://doi.org/10.1039/C3FD00085K, 2013.

Etzler, F. M.: A statistical thermodynamic model for water near solid interfaces, J. Coll. Interf. Sci., 92, 43–56, 1983.

Holten, V. and Anisimov, M.: Entropy-driven liquid–liquid separation in supercooled water, Scientific reports, 2, 713, https://doi.org/10.1038/srep00713, 2012.

Holten, V., Limmer, D. T., Molinero, V., and Anisimov, M. A.: Nature of the anomalies in the supercooled liquid state of the mW model of water, J. Chem. Phys., 138, 174 501, 2013.

Kärcher, B. and Lohmann, U.: A parameterization of cirrus cloud formation: Heterogeneous freezing, J. Geophys. Res., 108, 4402,https://doi.org/10.1029/2002JD003220, 2003.

Kashchiev, D.: Nucleation: basic theory with applications, Butterworth Heinemann, 2000.

Knopf, D. A. and Alpert, P. A.: A water activity based model of heterogeneous ice nucleation kinetics for freezing of water and aqueous solution droplets, Faraday disc., 165, 513–534, 2013.

Koop, T., Luo, B., Tslas, A., and Peter, T.: Water activity as the determinant for homogeneous ice nucleation in aqueous solutions, Nature, 406, 611–614, 2000.

Marcolli, C., Gedamke, S., Peter, T., and Zobrist, B.: Efficiency of immersion mode ice nucleation on surrogates of mineral dust, Atmos. Chem. Phys., 7, 5081–5091, 2007.

Prausnitz, J. M., Lichtenthaler, R. N., and de Azevedo, E. G.: Molecular thermodynamics of fluid-phase equilibria, Prentice Hall, Upper Saddle River, NJ, USA, 3rd edn., 1998.

---

## Author Comment (AC2) · 13 May 2018

**1   Response to Referee 3**

**Reviewer:** *This paper proposes a new theoretical model for immersion nucleation, by investigating the thermodynamic and kinetic impact of the solid particle on nearby water molecules and its consequences for ice nucleation within the liquid droplet. Although immersion freezing is one of the main pathways of ice formation in the atmosphere, it is still poorly understood and the topic addressed in the paper is of great relevance for cloud physics. Furthermore, the paper puts together an important number of previous works in an attempt to make progress on our understanding of immersion nucleation.  It is overall rather clearly written and the reasoning is supported by high*

*quality figures and schematics. This paper could hence be an appropriate contribution to ACP. However, I believe there are shortcomings in the theoretical derivation and its presentation that should be resolved before the paper can be considered for publication. I therefore recommend major revisions of the current manuscript. In the following I will explain my concerns in more detail.*

**Response:**The comments by the reviewer are greatly appreciated. Please find detailed responses below.

**Major Points**

**Reviewer:** *1) Presentation of the theoretical development: I am not a specialist of ice nucleation and the related thermodynamics and kinetics. However, this will be the case for other ACP readers who would like to use the results presented in the paper. Since the theoretical derivation mainly consists in chemical physics, one possibility would be that the author submits this study to another journal, such as "The Journal of Chemical Physics". If the author chooses to present this work in ACP, I think some significant efforts should be spent in order to make the paper more accessible to the bulk of ACP readers. In particular, I think the organization of the derivation could be improved in that regard:.*

**Response:** Investigations on subject of ice nucleation, either from the experimental or the theoretical point of view are within the scope of ACP. Many haven been published in the journal during the last decade. Understandably most studies are experimental. Theoretical investigations are however of great importance to the atmospheric community, particularly as many authors may not regularly consult more fundamental journals like JCP. Most of the concepts discussed in this work are basic thermodynamics and physical chemistry, and therefore within the grasp of the broad atmospheric science community. I agree that the organization could be improved and the revisited paper

has been reorganized to make it more readable.

**Reviewer:** *Indeed, most ACP readers will be interested in the derived nucleation rate for immer- sion nucleation. Thus, I would start the theoretical section with the general expres- sion for the nucleation rate, i.e. the product of the concentration of critical clusters cg = C0 exp(−△G/kB T ), corrected by the Zeldovich factor times the flux of water molecules towards those clusters Fw : Jhet = Zcg Fw Thermodynamic effects of the particle on vicinal water affect cg and Z (through △G, the nucleation barrier for critical germ size) while kinetic effects affect Fw (the flux of water molecules towards the ice germs). After stating this, I would then elaborate on how expressions for the different factors are obtained in the new theory. This is mainly a change in presentation: most of the content is already present in the paper, but it should be made clearer where the derivation is going, e.g. when reading section 2.3 the reader sometimes misses the goal of the development which is only made clear in section 2.4.*

**Response:** This is an excellent suggestion. In the revisited paper I have made the calculation of the nucleation rate the central theme of the paper, starting as the reviewer suggests with a broad definition of $J_{het}$ then followed by Eq. (32). The distinction paragraphs of sections of 2.3.3, 2.4 have now been moved to a new broad introduction before Section 2.1. It must be noted however that the distinction between "kinetic" and "thermodynamic" effects is not clear cut in the proposed model since the flux of water molecules to the nascent ice germ is controlled by the thermodynamic driving force (Barahona, 2015). This is also clarified.

**Reviewer:** *2) Comparison with the classical theory of nucleation: The main point of the paper is to take into account the change in the thermodynamic and "dynamic" properties of vicinal water near the immersed solid particle and the impact on ice nu- cleation. In that sense, it differs from the classical nucleation theory (CNT) which rather considers the influence of the solid particle-liquid water interface directly. Although the*

*CNT expression for the nucleation rate is recalled in section 2.2, it is not really contrasted with the new theory. I miss a more thorough discussion comparing the different expressions and hypotheses between the theory introduced here and CNT. In particular, a table comparing the CNT and new theory expressions for the different factors in Jhet would be useful. I would suggest to add a dedicated section on that point in the discussion (and remove section 2.2).*

**Response:** This becomes much clearer with the reorganization of the paper. Since now a broad formulation of $J_{het}$ is introduced earlier in the work it is easier to distinguish how each theory defines the relevant terms (nucleation work, molecular flux). The suggested table may be confusing since the equations involved are quite long. It is worth mentioning that the theory presented here builds upon previous work (?Barahona, 2015) and therefore does not only differs from CNT on the effect of the particle on the vicinal water but also on how other terms are defined. This has been made clearer in the revised work. The comparison against CNT was partially addressed in Figure 7.

Following the reviewer's suggestion a new, separate section has been introduced to clarify this.

**Reviewer:** *3) Contents: This is another reason for my reservations. On several instances, I have noticed algebra mistakes which are repeated in several formulas. This casts some doubts on the whole theoretical derivation and it is unclear without repeating all the work whether the related figures are correct or not. Since the theoretical derivation is central to the paper, it is essential that the author makes sure all the formulas are correct (and convinces the reviewer). References to previous studies should also be made as explicit as possible, to make the argument easier to follow. I list below the main two mistakes I have noticed:*

**Response:** The reviewer rightly points out an error in the derivation of the theory, as

well as a number of typos. As shown below the effect of this error is limited and does not change the conclusions of the study. The corrected derivation of the involved equations is shown at the end of this document and it is now included in the the revisited paper. All the Figures are corrected in the revisited paper as well.

**Reviewer:** *page 7, Eq. (17): I have: $a_w = a_{w,\text{ eff}}^{\frac{1}{1+\zeta}} a_{w,eq}^{\frac{\zeta}{1+\zeta}} \exp\left(\Lambda_E \frac{1-\zeta}{1+\zeta}\right)$ instead of the formula of the author. This formula is used in many instances, for example in Eqs. (19), (20), (21), (22), and (45)*

**Response:** The wrong expression for $\Delta\mu_{\text{S}}$ was written in the text. $\Delta\mu_{\text{S}}$ must actually be calculated at $a_{\text{w, eff}}$, hence Eq. (14) of should read:

$$\Delta\mu_{\text{S}} = -k_{\text{B}}T\ln\left(\frac{a_{\text{w, eff}}}{a_{\text{w, eq}}}\right), \tag{1}$$

After introducing this equation into Eq.(13) it can be readily seen that Eq. (17) of the original paper is correct. Equation (14) was also used to simplify Eq. (40); this has been corrected as well. The derivation of Eq. (1) is shown at the end of this document (Eq. 12).

**Reviewer:** *page 6, eq (7), (8) and (10): if the $g^E$ term represents an excess energy imposing a penalty to mixing (and representing the tendency of IL and LL regions to clus ter), it should be positive: $g^E = +A_w\zeta(1-\zeta)$ with $A_w = \frac{2k_{\text{B}}T_C}{N}$. In the current formulation the first part of Eq. (9), i.e. $\frac{\partial^2 \mu_{YC}}{\partial\zeta^2} = 0$ does not hold at $\zeta = 0.5$. Some of the following equations build on this result (among which Eqs (12), (13), ...). 2 In Eq. (17) $\Lambda_E$ should be $\Lambda_E = -\frac{2}{N}\frac{T_c}{T}$ . Because of this error, the current Eq. (12) disagrees with Eq. (8) in Holten et al. (2013).*

**Response:** This is indeed an error. Since $A_w$ is a phenomenological parameter it can

**Fig. 1.** Comparison between using the original(dashed lines) and the corrected (solid lines) expressions for $a_{\text{w, eff}}$.

in principle have any value and sign. However this conflicts with the notion of $A_w$ as a function of the critical temperature, $T_c$, and it is a mistake. In the appendix of this document the full derivation the equation of state of vicinal water has been reworked to (i) correct errors and typos, and (ii) to make it more readable stating clearly all the assumptions involved. The new expression is very close to the original expression. Both lead to the same general form for $a_{\text{w, eff}}$, i.e.,

$$a_{\text{w, eff}} = \left(\frac{a_{\text{w}}}{a_{\text{w, eq}}^{\zeta}}\right)^{\frac{1}{1-\zeta}} \exp\left(-\frac{\Lambda_{\text{mix}}}{\zeta - 1}\right). \tag{2}$$

In the original version (Eq. 19, with slightly different nomenclature) :

$$\Lambda_{\text{mix}} = -\frac{2}{N}\frac{T_c}{T}\zeta(1 - \zeta) \tag{3}$$

In the corrected version (Eq. 20 of this document):

$$\Lambda_{\text{mix}} = \frac{1}{N}\left[\zeta\ln(\zeta) + (1 - \zeta)\ln(1 - \zeta)\right] + \frac{2}{N}\frac{T_c}{T}\zeta(1 - \zeta) \tag{4}$$

The mixing term is only significant when $\zeta \sim 0.5$, since the energy of mixing vanishes for pure components. In Figure 1 (see supplement file) the effective water activity, the work of nucleation and the nucleation rate are drawn for $a_w = 1$ at different temperatures and values of $\zeta$, and for the original (dashed) and corrected (solid) expressions for $a_{\text{w,eff}}$. For $\zeta = 0.5$ there is about a factor of two difference in $\Delta G_{\text{het}}$ leading to about

two orders of magnitude difference in $J_{\text{het}}$. Although these are significant differences, $a_{\text{w,eff}}$, $\Delta G_{\text{het}}$ and $J_{\text{het}}$, are otherwise very similar. This shows that $\Lambda_{\text{mix}}$ only plays a secondary role, and the main conclusions of the study remain valid.

**Specific Comments**

**Reviewer:** *1. suggest to change the title from "On the Thermodynamic and Dynamic Aspects of Immersion Ice Nucleation" to "On the Thermodynamic and Kinetic Aspects of Immersion Ice Nucleation"; the "dynamic" aspects that the author refers to are related to the diffusion of water molecules in the fluid and in that sense could be referred to as "kinetic" (dynamic brings fluid dynamics to mind).*

**Response:** Dynamic was used as "kinetics" and "thermodynamics" blend in the new theory. However I agree that it may be confusing. The title has been changed.

**Reviewer:** *2. p 4, Eq (1) differs from the common expression in Prupaccher and Klett, which also includes $N_c$, $\Omega$, the number of water molecules in contact with the cluster*

**Response:** The form of Eq.(1) has been used by several authors (e.g., Marcolli et al., 2007), and is shown Eq. 9-37 of Pruppacher adn Klett (1997), although the correct equation does not include $Z$. This has been corrected in the revisited paper.

**Reviewer:** *3. p 5, Eqs (4), (5) and (6): it is more common to define an increasing entropy upon mixing. Thus, in both equations (4) and (5) it might be clearer to add a minus sign in front of $T\Delta S_{mix}$ in Eq. (4) and (5) and after "=" in Eq (6). This has no impact on the subsequent equations, but would be more consistent with the usual conventions.*

**Response:** In the revisited paper $\Delta G_{mix}$ is written directly from Eq. 16 of Holten et al. (2013). The expression is then explained as a combination of an ideal entropy of mixing

and an empirical form for the enthalpy of mixing. The derivation is also found in several textbooks (e.g., Prausnitz et al., 1998)). This limits the number of definitions introduced in the text.

**Reviewer:** *4. p 10, Eq (24): I am not convinced that with that definition, $\Delta\mu_i$ should be referred to as supersaturation.*

**Response:** The sentence now reads "where $\Delta\mu_i$ represents the driving force for nucleation".

**Reviewer:** *5. p 10, line 9: unit of s should be $molec^{1/3}$ so that the units match in Eqs (23) and (25). 6. p 10, Eq 27: should be $n^{-4/3}$ rather than $n^{-1/3}$. 7. p 11, line 2: specify here again the condition for mechanical equilibrium.*

**Response:** Corrected.

**Reviewer:** *8. p 12, Eq (22) and l 25: here $C_0$ seems to be the monomer concentration per surface unit of the particle (and not in a volume of fluid), but this is only mentioned after Eq (44) where it is specified that $C_0 = 1/a_0$ is the cross-sectional area of of a water molecule. This should already be written line 25. The numerical value of $a_0$ should be mentioned.*

**Response:** As the reviewer points out $C_0$ is the concentration of molecules susceptible to grow into ice germs (i.e., the monomer concentration). Defining it as $1/a_0$ early in the text would unnecessarily limit the scope of the equation. For calculations we use $a_0 = \pi d_0^2$ with $d_0 = \left(\frac{6 * v_w}{pi}\right)^{1/3}$ being $v_w$ the molecular volume of water. This has been added to Table 1.

To address the reviewer's concern the following line was also added to the Section: "$C_0$ could be defined either per-area or per-volume basis".

**Reviewer:***Furthermore, it is surprising that the author takes $C_0 = 1/a_0$. This implies that only the molecules in direct contact with the particle are considered as vicinal water susceptible to grow into ice germs. This contradicts the motivation for the development expressed, e.g. p3, l13-14: "In a groundbreaking work, Anderson (1967) found strong evidence of ice formation several molecular diameters away from the clay-water interface." The author should at least comment on that.*

**Response:** It is more appropriate to write $C_0$ as proportional to the volume of the vicinal layer. Unfortunately this would bring confusion since historically heterogeneous nucleation rates are normalized to the particle surface area. $C_0 = 1/a_0$ is thus consistent with current literature and avoids a formal definition of the volume of the vicinal layer which could be quite challenging.

However I disagree that this means that only the molecules in direct contact with the particle are considered vicinal water. The only assumption involved is that the density of water does not vary within the droplet, and remains constant even within the vicinal water. Thus, anywhere in the liquid the per-area molecular density should be the same as in the vicinal layer, and $J_{\text{het}}$ can be scaled with respect to the immersed particle surface area. This only an approximation since under the proposed model low density regions are precursors to ice. But the density difference between ice and liquid is relatively small and such discrepancy should play a minor role.

The explanation above has been added after Eq.(44).

**Reviewer:***9. p 12, Eq (33): It is not clear to me how the author comes up with that expression for the Zeldovich factor in this case, especially with $n^* = n_{hom} + 2$. I would rather obtain:*

$$Z = \left[ -\frac{\left( \frac{\partial^2 \Delta G}{\partial n^2} \right)_{n=n^*}}{2\pi k_{\mathrm{B}} T} \right]^{1/2} = \left[ \frac{\Delta G_{\mathsf{het}}}{3\pi k_{\mathrm{B}} T n_{hom}(n_{hom} + 2)} \right]^{1/2} \tag{5}$$

*The derivation should be briefly explained.*

**Response:** It seems that in his/her derivation the reviewer calculated $\left( \frac{\partial^2 \Delta G}{\partial n^2} \right)_{n=n_{hom}}$ instead of $\left( \frac{\partial^2 \Delta G}{\partial n^2} \right)_{n=n_{hom}+2}$. The original expression, Eq. (33) was obtained assuming that the form of the Zeldovich factor without dissipation holds in the new theory. This is exact for the germ forming regime, but it is only an approximation for the spinodal regime. The correct expression for $Z$ is

$$Z = \left[ \frac{\Delta G_{\mathsf{het}} n_{\mathsf{hom}}^{1/3}}{3\pi k_{\mathrm{B}} T (n_{\mathsf{hom}} + 2)^{7/3}} \right]^{1/2} \tag{6}$$

for $n^* > 3$ , $\frac{n_{\mathsf{hom}}^{1/3}}{(n_{\mathsf{hom}}+2)^{7/3}} \approx \frac{1}{(n^*)^2}$, with $n^* = n_{\mathsf{hom}} + 2$. Indeed the discrepancy is only $30\%$ at $n^* = 3$ and much smaller for larger clusters. Therefore the assumption made in Eq.(33) is largely valid.

However for $n^* = 2$ the exact expression cannot be used since $Z = 0$; for this regime Eq.(33) is only an approximation. The issue is rather fundamental. As explained by Kashchiev (2000), Chapter 13, the Zeldovich method consists in approximating $\Delta G(n)$ with a second order Taylor expansion around $n^*$, which then is used to simplify the cluster population balance. As least two assumptions are involved (i) the cluster size distribution is at equilibrium and (ii) each germ grows by addition of a single molecule at a time. Both assumptions break in the spinodal regime. Unfortunately solving this issue requires a complete shift in the way cluster growth is modelled and it is beyond

the scope of this work.

This caveat is now acknowledged and the explanation above has been added to the revisited paper.

**Reviewer:** *10. Section 2.4: please be more specific in this section regarding which assumptions have been previously made in the literature and which are introduced in this paper. Beyond the suggestions above for Sect.2, the presentation of this subsection on kinetics could be improved; e.g. I would put the text from l 24 p13 to l 1 p14 before Eq (35) since it provides some justification for the linear scaling introduced in Eq. (35)*

**Response:** The revisited paper expands this Section (and Section 2) to clarify the approach. Much of the justification to this Section is explained in Barahona (2015) and it is now is briefly summarized in this work. The Section has also been reorganized following the reviewer's suggestion.

**Reviewer:** *11. p 16, l 28-32: Section 3.1 Please give a mathematical definition of the freezing temperature . The current definition is not very clear, the term "equilibrium temperature" suggests thermodynamic equilibrium between ice and liquid water. whereas nucleation is a kinetic process. I am not convinced, given the information in the next paragraph, that this $T_{\mathrm{f,eq}}$ can be referred to as an equilibrium temperature. In the legend and ylabel of Fig. 4, please add the symbol $T_{\mathrm{f,eq}}$.*

**Response:** Thanks for pointing this out. The concept of thermodynamic freezing temperature refers to the pseudoequilibrium temperature between liquid and ice, for a given $a_w$. Qualitatively it must be the temperature at which freezing would be observed if no kinetic limitations existed, so that freezing is only dictated by thermodynamics (hence it must be the highest observable freezing temperature).

In the framework proposed in this work, this can be understood as the value of $T$ for which $a_{\mathrm{w,het}} = a_{\mathrm{w}}$ in Eq. (20) which does not depend on the kinetics of the system.

This definition however depends on selecting a value for $a_{\mathrm{w,hom}}$. A more fundamental definition may be achieved by using the spinodal separation of pure water instead of $a_{\mathrm{w,hom}}$ (Holten and Anisimov, 2012) as reference.

The explanation above has been added to the Section. The symbol has been changed to $T_{\mathrm{ft}}$ and added to the caption of Figure 4.

**Reviewer:** *12. p 21 l 22: "regular solution" -> mixture ?.*

**Response:** It is a solution. This is now more clear in the new derivation.

**Reviewer:** *Table 1: when relevant, the numerical values (or the expressions) of the quantities corresponding to the symbols should be added there, and the books/papers from which the estimates are taken should be referenced. For instance, the value of a0 is not given. The units should always be specified (e.g. the cooling rate has no units). Also note that the unit "mol" is different from molecule and one should rather write "molec"*

**Response:** Corrected.

**2   Corrected derivation of the equation of state of vicinal water**

The vicinal layer is defined as a solution of hypothetical ice-like (IL) and liquid-like (LL) regions, with Gibbs free energy given by

$$\mu_{\mathrm{vc}} = (1 - \zeta)\hat{\mu}_{\mathrm{LL}} + \zeta\hat{\mu}_{\mathrm{IL}}, \tag{7}$$

where $\hat{\mu}_{LL}$ and $\hat{\mu}_{LL}$ are the chemical potentials of the LL and IL species within the solution, respectively, and $\zeta$ is the fraction of IL regions in the layer. Equation (7) can also be written in terms of the chemical potentials of the "pure" LL and IL species, $\mu_{LL}$ and $\mu_{IL}$, respectively, in the form,

$$\mu_{vc} = (1 - \zeta)\mu_{LL} + \zeta\mu_{IL} + \Delta G_{mix} \qquad (8)$$

where $\Delta G_{mix} = (\hat{\mu}_{IL} - \mu_{IL})\zeta + (1 - \zeta)(\hat{\mu}_{LL} - \mu_{LL})$ is the Gibbs energy of mixing. For a mechanical mixture of pure LL and IL species, $\Delta G_{mix} = 0$, whereas for an ideal solution $\Delta G_{mix}$ is determined by the ideal entropy of mixing (Prausnitz et al., 1998). Reorganizing Eq. (8) we obtain,

$$\mu_{vc} = \mu_{LL} + \zeta\Delta\mu_{il} + \Delta G_{mix} \qquad (9)$$

where $\Delta\mu_{il} = \mu_{IL} - \mu_{LL}$. $\Delta\mu_{il}$ can be approximated by using the equilibrium between bulk liquid and ice as reference state so that (Kashchiev, 2000),

$$\mu_{IL} = \mu_{eq} + k_B T \ln(a_{IL}), \qquad (10)$$

and

$$\mu_{LL} = \mu_{eq} + k_B T \ln\left(\frac{a_{w,\,eff}}{a_{w,\,eq}}\right), \qquad (11)$$

where $a_{w,\,eff}$ is termed the "effective water activity" and it is the value of $a_w$ associated with the LL regions in the vicinal water, and $a_{IL}$ is the water activity in the IL regions. Assuming that similarly to bulk ice the solute does not significantly partition to the IL phase, then $a_{IL} \approx 1$. With this, and combining Eqs.(10) and (11), and rearranging we obtain,
$$\Delta\mu_{\mathrm{il}} = -k_{\mathbf{B}}T \ln\left(\frac{a_{\mathrm{w,\,eff}}}{a_{\mathrm{w,\,eq}}}\right), \qquad (12)$$

The central assumption behind Eq. (12) is that $a_{\mathrm{w,\,eq}}$ corresponds to the equilibrium water activity between liquid and ice, or in other words that near equilibrium $\Delta\mu_{\mathrm{il}} \approx \Delta\mu_{\mathrm{S}}$, being $\Delta\mu_{\mathrm{S}}$ the excess free energy of solidification of water.

In reality $\Delta\mu_{\mathrm{S}}$ corresponds to actual liquid and ice instead of the hypothetical LL and IL substances. This difference can be accounted for by selecting a proper functional form for $\Delta G_{\mathrm{mix}}$, for which several empirical and semi-empirical interaction models with varying degrees of complexity exist (Prausnitz et al., 1998). In this work it is going to be assumed that the vicinal water can be described as a regular solution. This is the simplest model that accounts for the interaction between solvent and solute during mixing and that is flexible enough to include corrections for the difference between $\Delta\mu_{\mathrm{S}}$ and $\Delta\mu_{\mathrm{il}}$. Holten et al. (2013) have shown that a regular solution can reasonably approximate the chemical potential of supercooled water. Moreover, the authors also showed that taking into account clustering of water molecules upon mixing leads to better agreement with MD simulations and experimental results.

According to the regular solution model, modified by clustering (Holten et al., 2013, c.f. Eq. 16),

$$\Delta G_{\mathrm{mix}} = \frac{k_{\mathbf{B}}T}{N}\left[\zeta\ln(\zeta) + (1-\zeta)\ln(1-\zeta)\right] + A_w\zeta(1-\zeta) \qquad (13)$$

The first term on the right hand side corresponds to the usual definitioin of the ideal entropy of mixing, i.e., random ideal mixing and a weak interaction between IL and LL regions, modified to account for clustering in groups of $N$ molecules. $N = 6$ corresponds to clustering in hexamers and is near the optimum fit between MD simulations and the solution model (Holten et al., 2013). It must be noted that Holten et al. (2013)

recommended an alternative model termed "athermal solution", where nonideality is ascribed to entropy changes upon mixing. In vicinal water some evidence points at nonideality originating from enthalpy changes near the particle (Etzler, 1983), hence a regular solution is more appropriate in this case. For $N = 6$ the difference between the two models is negligible (Holten et al., 2013).

The second term on the right hand side of Eq. (13) is an empirical functional form used to approximate the enthalpy of mixing selected so that $\Delta G_{\text{mix}} = 0$ for $\zeta = 0$ and $\zeta = 1$. $A_w$ is a phenomenological interaction parameter and typically must be fitted to experimental observations. Here it is assumed $A_w$ also implicitly corrects the approximation $\Delta \mu_{\text{il}} \approx \Delta \mu_{\text{s}}$.

An important aspect of the regular solution model is that it predicts that $\Delta G_{\text{mix}}$ (hence $\mu_{\text{vc}}$) has a critical temperature, $T_c$, at $\zeta = 0.5$, defined by the conditions,

$$\frac{\partial^2 \Delta G_{\text{mix}}}{\partial \zeta^2} = 0 \,, \; \frac{\partial^3 \Delta G_{\text{mix}}}{\partial \zeta^3} = 0. \tag{14}$$

[revised manuscript text omitted]

**Supplement:**

---

## Author Comment (AC3) · 17 May 2018

The comment was uploaded in the form of a supplement:
https://www.atmos-chem-phys-discuss.net/acp-2017-1019/acp-2017-1019-AC3-supplement.pdf

---

## Author Response (AR1)

**Response to Comments**

**1   Referee 3**

**Reviewer:** *This paper proposes a new theoretical model for immersion nucleation, by investigating the thermodynamic and kinetic impact of the solid particle on near-by water molecules and its consequences for ice nucleation within the liquid droplet. Although immersion freezing is one of the main pathways of ice formation in the atmosphere, it is still poorly understood and the topic addressed in the paper is of great relevance for cloud physics. Furthermore, the paper puts together an important number of previous works in an attempt to make progress on our understanding of immersion nucleation. It is overall rather clearly written and the reasoning is supported by high quality figures and schematics. This paper could hence be an appropriate contribution to ACP. However, I believe there are shortcomings in the theoretical derivation and its presentation that should be resolved before the paper can be considered for publication. I therefore recommend major revisions of the current manuscript. In the following I will explain my concerns in more detail.*

**Response:** The comments by the reviewer are greatly appreciated. Please find detailed responses below.

**Major Points**

**Reviewer:**   *1) Presentation of the theoretical development: I am not a specialist of ice nucleation and the related thermodynamics and kinetics. However, this will be the case for other ACP readers who would like to use the results presented in the paper. Since the theoretical derivation mainly consists in chemical physics, one possibility would be that the author submits this study to another journal, such as "The Journal of Chemical Physics". If the author chooses to present this work in ACP, I think some significant efforts should be spent in order to make the paper more accessible to the bulk of ACP readers. In particular, I think the organization of the derivation could be improved in that regard:.*

**Response:** Investigations on the subject of ice nucleation, either from the experimental or the theoretical point of view are within the scope of ACP. Many havenn been published in the journal during the last decade. Understandably most studies are experimental. Theoretical investigations are however of great importance to the atmospheric community, particulalry as many authors many not often consult fundamental physics journals like JCP. Most concepts discussed in this work are also basic physical chemistry and therefore within the grasp of the broad atmospheric science community. I agree that the organization could be improved and the revisited paper has been reorganized to make

it more readable.

**Reviewer:** *Indeed, most ACP readers will be interested in the derived nucleation rate for immer- sion nucleation. Thus, I would start the theoretical section with the general expres- sion for the nucleation rate, i.e. the product of the concentration of critical clusters cg = C0 exp($-\Delta G/k_B$ T ), corrected by the Zeldovich factor times the flux of water molecules towards those clusters Fw : Jhet = Zcg Fw Thermodynamic effects of the particle on vicinal water affect cg and Z (through $\Delta G$, the nucleation barrier for critical germ size) while kinetic effects affect Fw (the flux of water molecules towards the ice germs). After stating this, I would then elaborate on how expressions for the different factors are obtained in the new theory. This is mainly a change in presentation: most of the content is already present in the paper, but it should be made clearer where the derivation is going, e.g. when reading section 2.3 the reader sometimes misses the goal of the development which is only made clear in section 2.4.*

**Response:** This is an excellent suggestion. In the revisited paper I have made the calculation of the nucleation rate the central theme of the paper, starting as teh reviewer suggest with the broad definition of $J_{\text{het}}$ as suggested then followed by Eq. (32). The firts paragrpahs of sections of 2.3.3, 2.4 have now been moved to a new broad introduction before Section 2.1. It mut be noted however that the distiction between "kinetic" and "thermodhyamic" effects is not as clear in the proposed model since the flux of water molecules to the nascent ice germ is controlled by the thermodynamic driving force Barahona (2015).

**Reviewer:** *2) Comparison with the classical theory of nucleation: The main point of the paper is to take into account the change in the thermodynamic and "dynamic" properties of vicinal water near the immersed solid particle and the impact on ice nucleation. In that sense, it differs from the classical nucle- ation theory (CNT) which rather considers the influence of the solid particle- liquid water interface directly. Although the CNT expression for the nucleation rate is recalled in section 2.2, it is not really contrasted with the new theory. I miss a more thorough discussion comparing the different expressions and hy- potheses between the theory introduced here and CNT. In particular, a table comparing the CNT and new theory expressions for the different factors in Jhet would be useful. I would suggest to add a dedicated section on that point in the discussion (and remove section 2.2).*

**Response:** This becomes much clearer with the reorganizatin of the paper. Since now a broad formulation of $J_{\text{het}}$ is introduced earlier in the work it is easier to refer to how the different theories define each of the relevant terms (nucleation work, molecular flux). The suggested table although seeminlgy a good idea is actually more confusing since the equations involved are quite long. It is worth mentioning that the theory presented here builds upon previous work Barahona (2014, 2015) and therefore does not only differs on the effect of the particle on the vicinal water but also on how other terms are defined. This has been made clearer in the revisted work. The comparison against CNT was partially addressed in Figure 7. Following the reviewer's suggestion a new, separate Section has been introduced.

[Figure]

Figure 1: Comparison between the effective water activity, the nucleation work, and the nucleation rate between the original (dashed) and the corrected (solid) formulations of $a_{\mathrm{w,eff}}$.

**Reviewer:** *3) Contents: This is another reason for my reservations. On several instances, I have noticed algebra mistakes which are repeated in several formulas. This casts some doubts on the whole theoretical derivation and it is unclear without repeating all the work whether the related figures are correct or not. Since the theoretical derivation is central to the paper, it is essential that the author makes sure all the formulas are correct (and convinces the reviewer). References to previous studies should also be made as explicit as possible, to make the argument easier to follow. I list below the main two mistakes I have noticed:*

**Response:** The reviewer rightly points out an error in the derivation of the theory, as well as a number of typos. As shown below the effect of this error is limited and does not change the conclusion of the study. The corrected derivation is shown at the end of this document and it is now included in the the revisited paper. All the Figures are corrected in the revisited paper as well.

**Reviewer:** *page 7, Eq. (17): I have: $a_w = a_{w,\ eff}^{\frac{1}{1+\zeta}} a_{w,eq}^{\frac{\zeta}{1+\zeta}} \exp\left(\Lambda_E \frac{1-\zeta}{1+\zeta}\right)$ instead of the formula of the author. This formula is used in many instances, for example in Eqs. (19), (20), (21), (22), and (45)*

**Response:** This is a typo. $\Delta\mu_{\mathrm{S}}$ should be calculated at $a_{\mathrm{w,\ eff}}$, hence Eq. (14) of the original paper should read:

$$\Delta\mu_{\mathrm{S}} = -k_{\mathrm{B}}T \ln\left(\frac{a_{\mathrm{w,\ eff}}}{a_{\mathrm{w,\ eq}}}\right), \tag{1}$$

After introducing this equation into Eq.(13) it can be readily seen that Eq. (17) of the original paper is correct. The derivation of Eq. (8) is shown at the end of this document (Eq. 16).

**Reviewer:** *page 6, eq (7), (8) and (10): if the $g^E$ term represents an excess energy imposing a penalty to mixing (and representing the tendency of IL and LL regions to clus ter), it should be positive: $g^E = +A_w\zeta(1-\zeta)$ with $A_w = \frac{2k_BTc}{N}$.*

*In the current formulation the first part of Eq. (9), i.e. $\frac{\partial^2 \mu_{vc}}{\partial \zeta^2} = 0$ does not hold at $\zeta = 0.5$. Some of the following equations build on this result (among which Eqs (12), (13), ...). 2 In Eq. (17) $\Lambda_E$ should be $\Lambda_E = -\frac{2}{N}\frac{T_c}{T}$ . Because of this error, the current Eq. (12) disagrees with Eq. (8) in Holten et al. (2013).*

**Response:** This is indeed an error. Since $A_w$ is a phenomenological parameter it can in principle have any value and sign. However this conflicts with the notion of $A_w$ as a function of the critical temperature, $T_c$, and it is a mistake. In the appendix of this document the full derivation the equation of state of vicinal water has been reworked to (i) correct errors and typos, and (ii) to make it more readable stating clearly all the assumptions involved. The new expression is very close to the original expression. Both lead to the same general form for $a_{\text{w, eff}}$, i.e.,

$$a_{\text{w, eff}} = \left(\frac{a_{\text{w}}}{a_{\text{w, eq}}^\zeta}\right)^{\frac{1}{1-\zeta}} \exp\left(-\frac{\Lambda_{\text{mix}}}{\zeta - 1}\right). \tag{2}$$

In the original version (Eq. 19 of the original paper, with slightly different nomenclature) :

$$\Lambda_{\text{mix}} = -\frac{2}{N}\frac{T_c}{T}\zeta(1-\zeta) \tag{3}$$

In the corrected version (Eq. 24 of this document):

$$\Lambda_{\text{mix}} = \frac{1}{N}\left[\zeta \ln(\zeta) + (1-\zeta)\ln(1-\zeta)\right] + \frac{2}{N}\frac{T_c}{T}\zeta(1-\zeta) \tag{4}$$

The mixing term is only significant when $\zeta \sim 0.5$, since the energy of mixing vanishes for pure components. In the Figure 1 the effective water activity, the work of nucleation and the nucleation rate are drawn for $a_w = 1$ for different temperatures and $\zeta$. For $\zeta = 0.5$ there is about a factor of two difference in $\Delta G_{\text{het}}$ leading to about two orders of magnitude difference in $J_{\text{het}}$. Although these are significant differences, $a_{\text{w,eff}}$, $\Delta G_{\text{het}}$ and $J_{\text{het}}$, are otherwise very similar. This shows that $\Lambda_{\text{mix}}$ only plays a secondary role, and the main conclusions of the study remain valid.

**Specific Comments**

**Reviewer:** *1. suggest to change the title from "On the Thermodynamic and Dynamic Aspects of Immersion Ice Nucleation" to "On the Thermodynamic and Kinetic Aspects of Immersion Ice Nucleation"; the "dynamic" aspects that the author refers to are related to the diffusion of water molecules in the fluid and in that sense could be referred to as "kinetic" (dynamic brings fluid dynamics to mind).*

**Response:** Dynamic was used as "kinetics" and "thermodynamics" start to blend in the new theory. However I agree that it may be confusing. The title has been changed.

**Reviewer:** *2. p 4, Eq (1) differs from the common expression in Prupaccher and Klett, which also includes $N_c$, $\Omega$, the number of water molecules in contact with the cluster*

**Response:** The form of Eq.(1) has been used by several authors e.g., Marcolli et al. (2007), and is shown Eq. 9-37 of Pruppacher adn Klett (1997), although the correct equation does not include $Z$. This has been corrected in the revisited paper.

**Reviewer:***3. p 5, Eqs (4), (5) and (6): it is more common to define an increasing entropy upon mixing. Thus, in both equations (4) and (5) it might be clearer to add a minus sign in front of $T\Delta S_{mix}$ in Eq. (4) and (5) and after "=" in Eq (6). This has no impact on the subsequent equations, but would be more consistent with the usual conventions.*

**Response:** In the revisited paper $\Delta G_{mix}$ is written directly from Eq. 16 of Holten et al. (2013). The expression is then explained as a combination of an ideal entropy of mixing and an empirical form for the enthalpy of mixing. This allows the readers to follow directly Holten et al. (2013) derivation. The derivation is also found in several textbooks (e.g., Prausnitz et al. (1998)). The goal is to limit the number of definitions in the derivation.

**Reviewer:***4. p 10, Eq (24): I am not convinced that with that definition, $\Delta \mu_i$ should be referred to as supersaturation.*

**Response:** The sentence now reads "where $\Delta \mu_I$ represents the driving force for nucleation".

**Reviewer:***5. p 10, line 9: unit of s should be $molec^{1/3}$ so that the units match in Eqs (23) and (25). 6. p 10, Eq 27: should be $n^{-4/3}$ rather than $n^{-1/3}$. 7. p 11, line 2: specify here again the condition for mechanical equilibrium.*

**Response:** Corrected.

**Reviewer:***8. p 12, Eq (22) and l 25: here $C_0$ seems to be the monomer concentration per surface unit of the particle (and not in a volume of fluid), but this is only mentioned after Eq (44) where it is specified that $C_0 = 1/a_0$ is the cross-sectional area of of a water molecule. This should already be written line 25. The numerical value of $a_0$ should be mentioned.*

**Response:** As the reviewer points out $C_0$ is the concentration of molecules suceptible to grow into ice germs (i.e., the monomer concentration). Defining it as $1/a_0$ would limit the scope of the equation. To address the reviewer's concern the following line was added: "$C_0$ is defined either per-area or per-volume basis". For teh calculations $a_0 = \pi d_0^2$ with $d_0 = \left(\frac{6 * v_w}{pi}\right)^{1/3}$ being $v_w$ the molecular volume of water (See table 1). This has been added to Table 1.

**Reviewer:***Furthermore, it is surprising that the author takes $C_0 = 1/a_0$. This implies that only the molecules in direct contact with the particle are considered as vicinal water susceptible to grow into ice germs. This contradicts the motivation for the development expressed, e.g. p3, l13-14: "In a groundbreaking work, Anderson (1967) found strong evidence of ice formation several molecular diameters away from the clay-water interface." The author should at least comment on that.*

**Response:** It is more appropiate to write as $C_0$ proportional to the volume of the vicinal layer. Unfortunately this would bring confusion since historically heterogeneous nucleation rates are defined using the particle surface area as

basis. Thus $C_0 = 1/a_0$ is more consistent with current literature and avoids a formal definitin of the volume of the vicinal layer which could be quite challenging.

However this does not mean that only the molecules in direct contact with the particle are considered vicinal water. Rather that the density of the water does not vary within the droplet, and remains constant even within the vicinal water. Thus, anywhere in the liquid the per-area molecular density should be the same as in the vicinal layer, and $J_{\text{het}}$ can be scaled with respect to the immersed particle surface area. This only an approximation since under the proposed model low density regions are precursors to ice. But the density difference between ice and liquid is relatively small and such discrepancy should play a minor role.

The explanation above has been added after Eq.(44).

**Reviewer:**9. *p 12, Eq (33): It is not clear to me how the author comes up with that expression for the Zeldovich factor in this case, especially with* $n^* = n_{hom} + 2$. *I would rather obtain:*

$$Z = \left[ -\frac{\left( \frac{\partial^2 \Delta G}{\partial n^2} \right)_{n=n^*}}{2\pi k_{\text{B}} T} \right]^{1/2} = \left[ \frac{\Delta G_{\text{het}}}{3\pi k_{\text{B}} T n_{hom}(n_{hom} + 2)} \right]^{1/2} \tag{5}$$

*The derivation should be briefly explained.*

**Response:** It seems that in his/her derivation the reviewer calculated $\left( \frac{\partial^2 \Delta G}{\partial n^2} \right)_{n=n_{hom}}$ instead of $\left( \frac{\partial^2 \Delta G}{\partial n^2} \right)_{n=n_{hom}+2}$. The original expression, Eq. (33) was obtained assuming that the form of Zeldovich factor from homogeneous ice nucleation holds for the heterogeneous case. This is exact for the germ forming regime, but only an approximation for the spinodal regime. The correct expression for $Z$ is

$$Z = \left[ \frac{\Delta G_{\text{het}} n_{\text{hom}}^{1/3}}{3\pi k_{\text{B}} T (n_{\text{hom}} + 2)^{7/3}} \right]^{1/2} \tag{6}$$

for $n^* > 3$ , $\frac{n_{\text{hom}}^{1/3}}{(n_{\text{hom}}+2)^{7/3}} \approx \frac{1}{(n^*)^2}$, with $n^* = n_{\text{hom}} + 2$. Indeed the discrepancy is only 30% at $n^* = 3$ and much smaller for larger clusters. Therefore the assumption made in Eq.(33) is largely valid.

However for $n^* = 2$ the exact expression cannot be used since $Z = 0$; for this regime Eq.(33) is only an approximation. The issue is rather fundamental. As explained by Kashchiev (2000), Chapter 13, the Zeldovich method consists in approximating $\Delta G(n)$ with a second order taylor expansion around $n^*$, which then is used to simplify the cluster population balance. As least two assumptions are involved (i) the cluster size distribution is at equilibrium and (ii) each germ grows by addition of a single molecule at a time. Both assumptions break in the spinodal regime. Unfortunately solving this issue requires a complete shift in the way cluster growth is modelled and it is beyond the scope of this work.

This caveat is now acknowledged and the explanation above has been added to the revisited paper.

**Reviewer:***10. Section 2.4: please be more specific in this section regarding which assumptions have been previously made in the literature and which are introduced in this paper. Beyond the suggestions above for Sect.2, the presentation of this subsection on kinetics could be improved; e.g. I would put the text from l 24 p13 to l 1 p14 before Eq (35) since it provides some justification for the linear scaling introduced in Eq. (35)*

**Response:** The revisited paper expands this Section (and Section 2) to clarify the approach. Much of the justification to this Section is explained in Barahona (2015) and it is now is briefly summarized in this work. The Section has also been reorgnized following the reviewer's sugestion.

**Reviewer:** 11. p 16, l 28-32: Section 3.1 Please give a mathematical definition of the freezing temperature . The current definition is not very clear, the term "equilibrium temperature" suggests thermodynamic equilibrium between ice and liquid water. whereas nucleation is a kinetic process. I am not convinced, given the information in the next paragraph, that this $T_{\mathrm{f,eq}}$ can be referred to as an equilibrium temperature. In the legend and ylabel of Fig. 4, please add the symbol $T_{\mathrm{f,eq}}$.

**Response:** Thanks for pointing this out. The concept of thermodynamic freezing temperature refers to the pesudo equilibrium (i.e., metastable) temperarure between liquid and ice, for a given $a_w$. Qualitatively it must be the temperature at which freezing is observed if no kinetic limitations exist, hence freezing is only dictated by thermodynamics (hence it must be tha highest observable freezing temperature).

In the framework proposed in this work, this can be understood as the value of $T$ for which $a_{\mathrm{w,het}} = a_{\mathrm{w}}$ in Eq. (20) which does not depend on the kinetics of the system. This definition however depends on selecting a value for $a_{\mathrm{w,hom}}$. A more fundamental definition may be achieved by using the spinodal separation of pure water instead of $a_{\mathrm{w,hom}}$ **?** as reference.

The explanation above has been added to the Section. The symbol has been changed to $T_{\mathrm{ft}}$ and added to the caption of Figure 4.

**Reviewer:** 12. p 21 l 22: "regular solution" -¿ mixture ?.
**Response:** It is a solution. This is now more clear in the new derivation.

**Reviewer:** Table 1: when relevant, the numerical values (or the expressions) of the quantities corresponding to the symbols should be added there, and the books/papers from which the estimates are taken should be referenced. For instance, the value of a0 is not given. The units should always be specified (e.g. the cooling rate has no units). Also note that the unit "mol" is different from molecule and one should rather write "molec"
**Response:** Corrected.

**2   Referee 1**

**Reviewer:** *In this manuscript the role of different ordered structures of water close and far from an immersed particle is investigated. A theory of immersion*

*freezing based on these different states is derived. The theoretical investigations are compared to real measurements of heterogeneous nucleation rates in different experiments. Since ice nucleation in general, and especially heterogeneous nucleation is not well understood and the theoretical investigations are not convincing at the moment, a theory based on thermodynamics of water is a very interesting step for improving our knowledge of heterogeneous ice nucleation. Thus, in general this is a valid contribution for Atmospheric Chemistry and Physics. However, before the manuscript can be accepted, some issues has to be clarified. Therefore I recommend major revisions of the manuscript. In the following I will explain my concerns in details.*

**Response:** I thank the reviewer for the comments on the manuscript. They are addressed in detail below.

**Major Points**

**Reviewer:** *1. Representation of the theory: The topic of ice nucleation is quite complicated and usually only classical nucleation theory or some additional topics are well known in the ice cloud community, whereas the more detailed thermodynamic basis is usually hidden in many discussions. In this study, the author has to present details for the development of the theory but also has to make sure that the reader can follow his line of arguments. It would be very helpful if the author would present a kind of roadmap at the very beginning to describe what he wants to derive finally and which steps will be necessary in order to do so. Otherwise the reader is really lost in details, which stem either from standard thermodynamic arguments or are of phenomenological type.*

**Response:** The revisited paper has been reorganized to clarify the approach. More specifically the calculation of the nucleation rate has been made the central topic of the derivation introducing a general definition firts and the constrasting how each term is aborded in the classical approach and what is new in the proposed theory. The derivation of the equation of state of vicinal water has also been reworked to make it clearer and correct errors/typos. Finally the Section on kinetics has been reorganized contrasting introducing general concepts first. This has made the revisited paper much more readable.

**Reviewer:** *Derivation of equation (17). I could not reproduce the central equation (17) in the form the author did, I ended with the expression*

$$a_w = a_{w,\text{ eff}}^{\frac{1}{1+\zeta}} a_{w,eq}^{\frac{\zeta}{1+\zeta}} \exp\left(\Lambda_E \frac{1-\zeta}{1+\zeta}\right) \tag{7}$$

*This is crucial, since the equation is often used in the following derivation.*

**Response:** I thank the reviewer for pointing this out. . This is a typo. $\Delta\mu_\text{S}$ should be calculated at $a_{w,\text{ eff}}$, hence Eq. (14) of the original paper should read:

$$\Delta\mu_\text{S} = -k_\text{B}T \ln\left(\frac{a_{w,\text{ eff}}}{a_{w,\text{ eq}}}\right), \tag{8}$$

After introducing this equation into Eq.(13) it can be readily seen that Eq. (17) of the original paper is correct. The derivation of Eq. (8) is shown at the end of this document (Eq. 16).

**Reviewer:** *For instance, I have several reservations about equation (19), since the limit $\zeta \to 1$ is not well defined. The author has to check his derivation of equation (17) and, if necessary also the derivation of the subsequent theory. In section 2.3.3 the model is extended to the spinodal limit and the limit $\zeta \to 1$ is investigated, which is unbounded in the current representation, but probably not for the derivation I have found. Thus, it is not clear to me if the discussion in this section still holds.*

**Response:** The new derivation at the end of this document shows more clearly that $a_{\text{w, eff}}$ is in fact bounded for $\zeta \to 1$. From Eq.(24),

$$a_{\text{W}} = a_{\text{W, eff}} \left( \frac{a_{\text{w, eq}}}{a_{\text{w, eff}}} \right)^{\zeta} \exp(\Lambda_{\text{mix}}) \tag{9}$$

and from Eq.(**??**)

$$a_{\text{w, eff}} = \left( \frac{a_{\text{w}}}{a_{\text{w, eq}}^{\zeta}} \right)^{\frac{1}{1-\zeta}} \exp \left( -\frac{\Lambda_{\text{mix}}}{\zeta - 1} \right). \tag{10}$$

Since $\Lambda_{\text{mix}} = 0$ for $\zeta = 1$, then Eq.(9) implies that $a_{\text{W}} = a_{\text{W, eq}}$. Hence for $\zeta = 1$, $a_{\text{W}} = a_{\text{W, eq}} = a_{\text{w, eff}} = 1$, indicating thermodynamic equilibrium.

**Minor Points**

**Reviewer:** *Could you explain the sign of the excess energy $g^E = -A_w \zeta(1 - \zeta)$? What is the thermodynamic reason for this choice?*

**Response:** The choice was made simply to obtain a positive $T_c$. However as pointed out by other reviewers there is an error in the derivation of the mixing/excess term. Motivated by this I have reworked the derivation correcting errors and making it more readable. Since the excess term only plays a minor role, the correction only resulted in about up one order of magnitude difference in $J_{\text{het}}$. The new derivation is presented at the end of this document and it is now included in the revised paper.

**Reviewer:** *What are the thermodynamic conditions for the derivation of the critical temperature, i.e. where do the conditions $\frac{\partial^2 \mu_{\text{VC}}}{\partial \zeta^2} = 0$ , $\frac{\partial^3 \mu_{\text{VC}}}{\partial \zeta^3} = 0$ come from? Please explain this shortly in the text.*

**Response:** A solution would split into two phases if by doing so lowers its Gibbs free energy (Prausnitz et al. (1998), c.f. Section 6.12). For a metastable solution $\mu_{\text{VC}}$ must be minimal, hence $\frac{\partial \mu_{\text{VC}}}{\partial \zeta} = 0$. The condition $\frac{\partial^2 \mu_{\text{VC}}}{\partial \zeta^2} < 0$ indicates that any increase in $\zeta$ increases $\mu_{\text{VC}}$ (i.e., the curve $\mu_{\text{VC}}$ vs. $\zeta$ becomes concave downward) such that it is thermodynamically more favorable for the solution to split into disticnt phases than to increase its concentration; $\frac{\partial^2 \mu_{\text{VC}}}{\partial \zeta^2} = 0$ thus limits the metastable region. The last condition, $\frac{\partial^3 \mu_{\text{VC}}}{\partial \zeta^3} = 0$, indicates that the metastable region reduces to a single point and there is a single critical temperature $T_c$ for a regular solution.

The explanation above has been introduced in the text.

**Reviewer:** *In section 2.3.2 the water activity shift for heterogeneous nucleation is derived from the theory. Could you compare this results also numerically with the use of a constant shift in actual parameterisations and comment this? How large is $\zeta$ for the usual parameterisations?*

**Response:** Since the mixing term is typically small, $\zeta \sim 1 - \frac{\Delta a_{\text{w, het}}}{\Delta a_{\text{w, hom}}}$, hence $0 < \zeta <\sim 1$ (the upper limit is somwhere around .96 due to mixing effects). This relationship is true only in the germ-forming regime where $J_{\text{het}}$ is mainly dictaed by thermodynamics. Kärcher (2003) suggested teh approximation $J_{\text{het}} \approx J_{\text{hom}}(\Delta a_{\text{w, het}})$. This of course resembles the definition of the nucleation work derived in Section 2.3.1. The revisited paper discusses this in further detail. As expected the two approaches considerable differ for the spinodal regime. It must be noted Knopf and Alpert (2013) parameterized $J_{\text{het}}$ as a function of $\Delta a_{\text{w, het}}$ however since their coefficients are material specific comparison against their work is left for future works.

**Reviewer:** *In figure 4 different curves of water activity are shown. As far as I understand, the colors (dark red to yellow) indicate different versions of the new theory (aw,het ). Thus, the label aw,het as red in the diagram is misleading*

**Response:** Corrected.

**3 Corrected derivation of the equation of state of vicinal water**

[revised manuscript text omitted]

$$\mu_{vc} = (1-\zeta)\mu_{LL} + \zeta(\mu_{IL}-\mu_{LL}) + T\Delta S_{mix} + g^E . G_{mix} \tag{19}$$

 where $\Delta G_{mix} = (\hat{\mu}_{IL} - \mu_{IL})\zeta + ($ is the Gibbs energy of mixing. For a mechanical mixture of pure LL and IL species, $\Delta G_{mix} = 0$, whereas for an ideal solution $\Delta G_{mix}$ is determined by the ideal entropy of mixing (?). Reorganizing Eq. (??) we obtain,

$$\Delta S_{mix}\mu_{vc} = \frac{k_B}{N}\zeta\ln(\zeta)\mu_{LL} + (1-\zeta)\ln(1-\zeta).\Delta\mu_{il} + \Delta G_{mix} \tag{20}$$

~~This approximation results from assuming random mixing and a weak interaction between IL and LL regions, which tend to cluster in groups of $N$ molecules. Deviations from this behavior are in principle corrected by the excess energy, $g^E$, provided a suitable expression is available. Since $g^E$ must be negligible for $\zeta = 0$ and $\zeta = 1$ the simplest model describing this behavior takes the form,~~ where $\Delta\mu_{il} = \mu_{IL} - \mu_{LL}$. $\Delta\mu_{il}$ can be approximated using the equilibrium between bulk liquid and ice as reference state (?). Making,

$$\mu_{IL} = \mu_{eq} + k_B T \ln(a_{IL}), \tag{21}$$

and

$$g^E\mu_{LL} = -A_w\zeta(1-\zeta)\mu_{eq} + k_B T \ln\left(\frac{a_{w,\,eff}}{a_{w,\,eq}}\right), \tag{22}$$

where $A_w$ is a phenomenological interaction parameter, and the negative sign accounts for the tendency of IL and LL regions to cluster (?). Using Eq. (?? $a_{w,eff}$ is termed the "effective water activity" and it is the value of $a_w$ associated with the LL regions in the vicinal water, and $a_{IL}$ is the water activity in the IL regions. It is assumed that similarly to bulk ice the solute does not significanlty partition to the IL phase, so that $a_{IL} \approx 1$. With this, and combining Eq.(??) and Eq. (??) within Eq. (??) (??) and rearranging we obtain,

$$\Delta\mu_{\text{vcil}} = \mu_{LL} + \zeta(\mu_{IL} - \mu_{LL}) + \frac{k_B T}{N}\zeta - k_B T \ln(\zeta) + (1-\zeta)\ln(1-\zeta) - A_w \zeta(1-\zeta), \left(\frac{a_{w,\text{ eff}}}{a_{w,\text{ eq}}}\right). \tag{23}$$

This expression corresponds to a regular solution approximation to the properties of vicinal water. It has been previously used with success in describing the The central assumption behind Eq. (??) is that $a_{w,eq}$ corresponds to the equilibrium water activity between liquid and ice, or in other words that near equilibrium $\Delta\mu_{il} \approx \Delta\mu_s$. In reality $\Delta\mu_s$ corresponds to actual liquid and ice instead of the hypothetical LL and IL substances. This difference can be accounted for by selecting a proper functional form for $\Delta G_{mix}$, for which several empirical and semiempirical interaction models, with varying degrees of complexity exist (?). In this work it is going to be assumed that the vicinal water can be described as a regular solution. This is the simplest model that accounts for the interaction between solvent and solute during mixing and that is flexible enough to include corrections for the difference between $\Delta\mu_s$ and $\Delta\mu_{il}$. Using this model ? were able to approximate the chemical potential of supercooled water(?). To use . The authors also showed that taking into account clustering of water molecules led to better agreement of the estimated water properties with MD simulations and experimental results.

According to the regular solution model, modified by clustering (?, Cf. Eq. 16),

$$\Delta G_{mix} = \frac{k_B T}{N}\left[\zeta\ln(\zeta) + (1-\zeta)\ln(1-\zeta)\right] + A_w\zeta(1-\zeta) \tag{24}$$

The first term on the right hand side corresponds to the usual definition of the ideal entropy of mixing, i.e., random ideal mixing and a weak interaction between IL and LL regions, modified to account for clustering in groups of $N$ molecules. $N = 6$ corresponds to clustering in hexamers and is near the optimum fit between MD simulations and the solution model (?). It must be noted that ? recommended an alternative model termed "athermal solution", where nonideality is ascribed to entropy changes upon mixing. In vicinal water some evidence points at nonideality originating from enthalpy changes near the particle (?), hence a regular solution is more appropiate in this case. For $N = 6$ the difference between the two models is negligible (?).

The second term on the right hand side of Eq. (??) , it must be casted in terms of measurable variables. For this we notice that ??) is an empirical functional form used to approximate the enthalpy of mixing, selected so that $\Delta G_{mix} = 0$ for both, $\zeta = 0$ and $\zeta = 1$. $A_w$ is a phenomenological interaction parameter here assumed to implicitly correct the approximation $\Delta\mu_{il} \approx \Delta\mu_s$. Typically $A_w$ must be fitted to experimental observations. In this work $A_w$ is calculated using an alternative approach, as follows.

An important aspect of the regular solution model is that it predicts that $\mu_{\text{vc}}$ has a critical temperature, $T_c$, at $\zeta = 0.5$, defined by the conditions(?),

$$\frac{\partial^2 \mu_{\text{vc}}}{\partial \zeta^2} = 0 \, , \, \frac{\partial^3 \mu_{\text{vc}}}{\partial \zeta^3} = 0. \tag{25}$$

These conditions originate because $\frac{\partial^2 \mu_{\text{vc}}}{\partial \zeta^2} = 0$ represents a stability limit for vicinal water. A solution would split into two phases if by doing so lowers its Gibbs free energy (?, Cf. Section 6.12). For a metastable solution $\mu_{\text{vc}}$ must be minimal, hence $\frac{\partial \mu_{\text{vc}}}{\partial \zeta} = 0$. The condition $\frac{\partial^2 \mu_{\text{vc}}}{\partial \zeta^2} < 0$ indicates that any increase in $\zeta$ increases $\mu_{\text{vc}}$ (i.e., the curve $\mu_{\text{vc}}$ vs. $\zeta$ becomes concave downward) such that it is thermodynamically more favorable for the solution to split into distinct phases than to increase its concentration. The last condition, $\frac{\partial^3 \mu_{\text{vc}}}{\partial \zeta^3} = 0$, indicates that the metastable region reduces to a single point and there is a single critical temperature, $T_c$, for a regular solution. Using Eq. (????) into Eq. (????) and solving for $A_w$ gives for $T = T_c$,

$$A_w = \frac{2k_{\text{B}}T_{\text{c}}}{N}. \tag{26}$$

Physically, $T_c$ represents the stability limit of the vicinal water, at which it spontaneously separates into IL and LL regions.

 Equation  ??) thus provides an opportunity to theoretically determine $A_w$ , since $T_c$ should also correspond to the temperature at which the work of nucleation becomes negligible. This is explored in Section ??.

Introducing Eqs.  (????), (??), and (??), into Eq.  ??) we obtain,

$$\mu_{\text{vc}} = \mu_{\text{LL}} - \zeta k_{\text{B}}T \ln\left(\frac{a_{\text{w, eff}}}{a_{\text{w, eq}}}\right) + \frac{k_{\text{B}}T}{N}\left[\zeta\Delta\mu_s - \frac{2k_{\text{B}}T_c}{N}\ln(\zeta) + (1-\zeta)\ln(1-\zeta)\right] + \frac{2k_{\text{B}}T_c}{N}\zeta(1-\zeta). \tag{27}$$

Making,

$$\Lambda_{\text{mix}} = \frac{1}{N}\left[\zeta\ln(\zeta) + (1-\zeta)\ln(1-\zeta)\right] + \frac{2}{N}\frac{T_c}{T}\zeta(1-\zeta), \tag{28}$$

Equation (**??**) can be written in the form,

$$\mu_{\text{vc}} = \mu_{\text{LL}} - \zeta k_{\text{B}} T \ln\left(\frac{a_{\text{w, eff}}}{a_{\text{w, eq}}}\right) + k_{\text{B}} T \Lambda_{\text{mix}} \tag{29}$$

Equation (**??**) is the equation of state of the vicinal water. It describes the properties of the vicinal water in terms of the material-specific parameter $\zeta$, and the interaction parameters $N$ and $\tilde{T_c}T_{\zeta}$. MD simulations indicate that $N \sim 6$ (**??**). $\tilde{T_c}T_{\zeta}$ 
[revised manuscript text omitted]

| $a_\text{w, eff}$ | Effective water activity |
| $a_\text{w,eq}$ | Equilibrium $a_\text{w}$ between bulk liquid and ice (**?**) |
| $a_\text{w, het}$ | Thermodynamic freezing threshold for heterogeneous ic |
| $a_\text{w, hom}$ | Thermodynamic freezing threshold for homogeneous ice |
| $C_0$ | Monomer concentration, m$^{-2}$ |
| $E, T_0$ | Parameters of the VFT equation defining $D_\infty$, 892 and 1 |
| $D$ | Diffusion coefficient for interface transfer, m$^2$ s$^{-1}$ |
| $D_\infty$ | Self-diffusion coefficient of bulk water (**?**), m$^2$ s$^{-1}$ |
| $D_0$ | Fitting parameter, $3.06 \times 10^{-9}$ m$^2$ s$^{-1}$ (**?**) |
| $d_0$ | Molecular diameter of water, $(6v_w/\pi)^{1/3}$, m |
| $f_\text{het}^*$ | Impingement factor for heterogeneous ice nucleation, s$^-$ |
| $f_\text{hom}^*$ | Impingement factor for homogeneous ice nucleation, s$^-$ |
|  $G$ | Gibbs free energy, J |
|  $h$ | Planck's constant, Js |
| $J_0$ | Preexponential factor m$^{-2}$ s$^{-1}$ |
| $J_\text{het}$ | Heterogeneous nucleation rate, m$^{-2}$ s$^{-1}$ |
|  $k_\text{B}$ | Boltzmann constant, J K$^{-1}$ |
| $N$ | Number of  |
|  $n$ | Number of molecules in a ice cluster |
| $n^*$ | Critical germ size |
| $n_\text{hom}^*$ | Critical germ size for homogeneous ice nucleation |
| $n_\text{t}$ | Number of formation paths of the transient state, 16 (**?**) |
| $p_\text{s,w}, p_\text{s,i}$ | Liquid water and ice saturation vapor pressure, respectiv |
|  $s$ | Geometric constant of the ice lattice, 1.105 molec$^{1/3}$ |
| $S_\text{i}$ | Saturation ratio with respect to ice |
| $S_\text{c,0}$ | Configuration entropy of water$^*$ |
| $S_\text{c}$ | Configuration entropy of vicinal water |
| $T$ | Temperature, K |
| $T_c$ | Critical separation temperature,  211.473 K |

| | |
|---|---|
| $v_{\text{w}}$ | Molecular volume of water in ice (**?**), $\text{m}^{-3}$ |
| $v_{\text{w,0}}$ | Molecular volume of water at 273.15 K |
| $\bar{W}, \bar{W}_{\text{vc}}$ $\bar{W}$ | Average transition probability |
| $W_{\text{d}}$ $W_{\text{diss}}$ | Work dissipated during cluster formation, J |
| $W_{\text{d}}$ | Work dissipated during interface transfer, J |
| $Z$ | Zeldovich factor |
| $\Delta a_{\text{w, het}}$ | $a_{\text{w, het}} - a_{\text{w,eq}}$ |
| $\Delta a_{\text{w, hom}}$ | $a_{\text{w, hom}} - a_{\text{w,eq}}$, 0.304 (**??**) |
| $\Delta G$ | Work of cluster formation, J |
| $\Delta G_{\text{act}}$ | Activation energy for ice nucleation, J |
| $\Delta G_{\text{hom}}$ | Nucleation work for homogeneous ice nucleation, J |
| $\Delta G_{\text{het}}$ | Nucleation work for heterogeneous ice nucleation, J |
|  $\Delta h_{\text{f}}$ | Heat of solidification of water, $\text{J mol}^{-1}$ (**??**) |
|  $\Delta \mu_{\text{s}}$ | Excess free energy of solidification of water, J |
| $\Delta \mu_{\text{i}}$ | Driving force for ice nucleation, J |
|  $\Lambda_{\text{mix}}$ | Dimensionless  $-\frac{2}{N}\frac{T_c}{T}$ mixing parameter, defined in Eq. (**??**) |
| $\Phi$ | Energy of formation of the ice-liquid interface, $\text{molec}^{1/3}$ J |
| $\Gamma_{\text{w}}$ | Molecular surface excess of at the interface, 1.46 (**??**) |
|  $\mu_{\text{w}}, \mu_{\text{s}}, \mu_{\text{vc}}$ | Chemical potential of water, ice and vicinal water, respectively J |
| $\rho_{\text{w}}, \rho_{\text{i}}$ | Bulk density of liquid water and ice, respectively, $\text{Kg m}^{-3}$ (**?**) |
| $\sigma_{\text{E}}$ | Dimensionless residual entropy |
| $\sigma_{\text{iw}}$ | Ice–liquid interfacial energy $\text{J m}^{-2}$ (**?**) |
| $\theta$ | Contact angle |
| $\zeta$ | Templating factor |
| $\Omega_{\text{g}}$ | Ice germ surface area, $\text{m}^{-2}$ |

* From the data of **?** the following fit was obtained: $S_{\text{c},0} = k_{\text{B}} v_{\text{w}}/v_{\text{w, 0}}(-7.7481 \times 10^{-5}T^2 + 5.5160 \times 10^{-2}T - 6.6716)$ $(\text{J K}^{-1})$ for $T$ between 180 K and 273 K.

[Figure]

**Figure 1.** Diagram representing a thermodynamic path including homogeneous ice nucleation with the same work as heterogeneous freezing.

[Figure]

**Figure 2.** Work of heterogeneous ice nucleation. Color indicates different temperatures.

[Figure]

**Figure 3.** Different representations of immersion freezing. (a) An ice germ (dark blue) forming on an active site (AS) by random collision of water molecules (light blue). (b) Low density regions (dark blue) forming in the vicinity of active sites within a dense liquid phase (light blue).

[Figure]

**Figure 4.**  Freezing temperature as a function of water activity.  Colored lines correspond to $a_{w, het}$  for different values of $\zeta$. Also shown are  the water activities at equilibrium and at the homogeneous freezing threshold, $a_{w, eq}$ and $a_{w, hom}$, respectively, and lines drawn applying constant water ativity shifts, $\Delta a_{w, het}$, of 0.05, 0.15 and 0.20.

[Figure]

**Figure 5.** Critical germ size (left panel) and work of heterogeneous ice nucleation (right panels) for different values of $\zeta$ (color). Black lines correspond to constant  $J_{het} = 10^{6}$ m$^{-2}$ s$^{-1}$.

[Figure]

**Figure 6.** Preexponential factor.  Colored lines indicates different values of $\zeta$. Black lines correspond to results calculated using CNT for different values of the contact angle, $\theta$.

[Figure]

**Figure 8.** Top panels: Heterogeneous ice nucleation rate calculated using a constant shift in $a_w$ (black, dotted, lines) for Leonardite (LEO $\Delta a_{w,\,het} = 0.2703$) and Pawokee Peat (PP, $\Delta a_{w,\,het} = 0.2466$) (top panels). Red, blue and green colors correspond to $a_w$ equal to 1.0, 0.931 and 0.872, respectively, for LEO and 1.0, 0.901 and 0.862 for PP. Shaded area corresponds to $\Delta a_{w,\,het} \pm 0.025$. Markers correspond to experimental measurements reported by **?**; error bars represent an order of magnitude deviation from the reported value. Bottom panels: $J_{het}$ calculated for constant  $\zeta = 0.949$ for LEO and  $\zeta = 0.952$ for PP. The shaded area corresponds to $a_w \pm 0.01$ and  $\zeta \pm 0.0015$.

---

## Referee Report (RR1)

**Review of "On the Thermodynamic and Dynamic Aspects of Immersion Ice Nucleation" by D. Barahona**

August 30, 2018

The paper has improved and the author has properly addressed a large part of my comments. However, I still find the current presentation sometimes difficult to follow. Part of it comes from the technicality of the subject, but I believe that the organization could be improved further to help the reader better understand the line of the argumentation. I have picked some examples, listed below. I would recommend that the author considers my suggestions before the paper can be accepted for publication.

Major points :

1) Distribution of topics between the different sections: On several instances, a topic is brought up by the author but only discussed several pages later. For example, the water activity shift is introduced p 12 but only used then p 20. I would move sect 2.4.1 entirely to the discussion. Similarly, section 3.5 and 3.1 may be merged. The discussion of the preexponential factor could be combined with the kinetic derivation (since, in the end, it is mainly kinetics)

2) Clarification of the notations: I am confused by some of the notations; for instance, in Eq. 38, $\frac{\partial \Delta G}{\partial n^*_{hom}} = 0$ implies that the partial derivative is taken as a function of $n^*_{hom}$. I believe this should read $\left(\frac{\partial \Delta G}{\partial n}\right)_{n=n^*_{hom}} = 0$. In several cases, $n^*_{hom}$ is used while it should just be $n$, at least as far as the reviewer understands.

Other comments:

p 10 : The meaning of the critical temperature $T_c$ should be made clearer. In particular, the absence of metastable equilibrium below $T_c$ should be explained, as in one of the answers to the referees. A textbook reference would also be welcome.

p 15, section 2.4.3: I find the argument presented in that section difficult to follow. My understanding was that $T_c$ should be determined experimentally, but here it is derived analytically "since T c should also correspond to the temperature at which the work of nucleation becomes negligible." (p 10, l 13-14). Could the author elaborate on that? How negligible is the work of nucleation (how large is the minimum)?

Fig. 4: As far as I understand, this figure represents contours of constant $J_{het} = J_{threshold}$

for different values of $\zeta$ (i.e. $T_{ft}(a_w)$ is such that $J_{het}(T_{ft}(a_w), a_w) = J_{threshold}$). Is that correct? If so this should be specified, in particular the value of $J_{threshold}$. Otherwise, a mathematical definition of $T_{ft}(a_w)$ would still be required so that the reader can understand the figure.

---

## Referee Report (RR2)

Review of revised version of
**On the Thermodynamic and Dynamic Aspects of Immersion Ice Nucleation**
by D. Barahona

**General comment:**
The manuscript has improved a lot and my concerns were mostly addressed in an adequate way. However, there are still some issue, which have to be clarified before this manuscript can be accepted for publication.

**Minor points:**

1. The mathematical reformulation of the theory is still hard to follow; at least I had to rewrite the equations and to make the manipulations by myself before I could get through it. Maybe it would be worth to add some more intermediate steps in the derivation. For instance, the step from eq. (34) to eq. (35) took me a while. Maybe some additional derivations could be given in an appendix.

2. I have still concerns with the interpretation of equations (31) and (33). While in general (31) is fine with me, the interpretation of eq. (31) for the case $\zeta \to 1$ is not straightforward. It is clear that $\Lambda_{\mathrm{mix}} = 0$ for $\zeta \to 1$,; however, one have to check, if the quotient $-\frac{\Lambda_{\mathrm{mix}}}{\zeta - 1}$ behave adequately for $\zeta \to 1$, such that the equation is really converging to the mentioned values of $a_{\mathrm{w},*}$, especially to $a_{\mathrm{w,eff}} = 1$. I suggest that this should be checked in details, since the limit is used later.

**Technical comments**
The reference Kärcher (2003) is published in Atmos. Chem. Phys.

---

## Author Response (AR2)

**Response to Comments**

**1 Referee 3**

**Reviewer:** *The paper has improved and the author has properly addressed a large part of my com- ments. However, I still find the current presentation sometimes difficult to follow. Part of it comes from the technicality of the subject, but I believe that the organization could be improved further to help the reader better understand the line of the argumentation. I have picked some examples, listed below. I would recommend that the author considers my suggestions before the paper can be accepted for publication.*

**Response:** Thanks for the positive assessment. Several comments and further equations have been added throughout the paper to better guide the reader on the derivations.

**Major Points**

**Reviewer:** *1) Distribution of topics between the different sections: On several instances, a topic is brought up by the author but only discussed several pages later. For example, the water activity shift is introduced p 12 but only used then p 20. I would move sect 2.4.1 entirely to the discussion. Similarly, section 3.5 and 3.1 may be merged.*

**Response:** This is a good suggestion. The derivation of the water activity shift has been expanded and moved to the end of Section 3, along with the discussion of the $T$ vs $a_w$ relationship and the application to the freezing by humic-like susbtances. This makes the thermodynamic derivation of Section 2 easier to follow.

**Reviewer:** *The discussion of the preexponential factor could be combined with the kinetic derivation (since, in the end, it is mainly kinetics)*

**Response:** This suggestion may distract from the derivation of the nucleation rate. The purpose of the section on the preexponential factor (and the discussion, Section 3) is to analyze separately the thermodynamic and kinetic behavior of $J_{het}$. Immersing such discussion within the mathematical derivation of the equations would make the derivation harder to follow.

**Reviewer:** *2) Clarification of the notations: I am confused by some of the notations; for instance, ... In several cases, $n_{hom}$ is used while it should just be $n$ , at least as far as the reviewer understands.*

**Response:** As the reviewer points out $n$ is appropiate to describe variation

in $\Delta G$ (i.e, the work of cluster formation). However one should refer to the critical sizes $n_{\mathrm{hom}}$ and $n_{\mathrm{het}}$ when describing variation in the works of nucleation, $\Delta G_{\mathrm{hom}}$ and $\Delta G_{\mathrm{het}}$, respectively. This has been clarified throughout the manuscript.

**Other comments:**

**Reviewer:** *3) The meaning of the critical temperature $T_c$ should be made clearer. In particular, the absence of metastable equilibrium below $T_c$ should be explained, as in one of the answers to the referees. A textbook reference would also be welcome.*

**Response:** The thermodynamic reasoning behing the onset of the critical temperarure $T_c$ is already explained after Eq.(25). To address the reviewer's comment the following text has been added after Eq.(26).

"Physically, $T_c$ represents the stability limit of the vicinal water, at which it spontaneously separates into IL and LL regions. At $T < T_{\mathrm{c}}$ the chemical potential of a equimolar solution of IL and LL would be larger than that of a simple mechanical mixture of the two species. Thus it is thermodynamically more favorable for the solution to split into its individual components, i.e., liquid and ice, leading to a stability limit of the system."

**Reviewer:** *p 15, section 2.4.3: I find the argument presented in that section diffcult to follow. My understanding was that $T_c$ should be determined experimentally, but here it is derived analytically "since $T_c$ should also correspond to the temperature at which the work of nucleation becomes negligible." (p 10, l 13-14). Could the author elaborate on that? How negligible is the work of nucleation (how large is the minimum)?*

**Response:** Thanks for bringing this up. Specifically, the criterion used to find $T_c$ is that the reversible work of nucleation (that is, without accounting for the dissipation term, $W_{\mathrm{diss}}$) becomes negligible. This happens when $\Delta G_{\mathrm{hom}}$ is minimum. $W_{\mathrm{diss}}$ is not included since the definition of $\Delta G_{\mathrm{mix}}$ by the regular solution approximation, does not account for such effects. The section has been expanded to clarify the criterion used to find $T_c$.

**Reviewer:** *Fig. 4: As far as I understand, this figure represents contours of constant $J_{het} = J_{threshold}$ for diferent values of $\zeta$... Is that correct? If so this should be specified, in particular the value of $J_{threshold}$. Otherwise, a mathematical definition of $T_{ft}(a_w)$ would still be required so that the reader can understand the figure.*

**Response:** $J_{\mathrm{hom}}$ thresholds are not neccesary to define $a_{\mathrm{w,\ hom}}$. Baker and Baker (2004) and Bullock and Molinero (2013) have separately provided thermodynamic definitions of $a_{\mathrm{w,\ hom}}$, the former as conciding with the maximum compressibility of water. Using their results it is no longer neccessary to define a threshold for $J_{\mathrm{hom}}$ since $a_{\mathrm{w,\ hom}}$ becomes a thermodynamic property of the system. Section 2.4.1 (now 3.5) has been rewritten to reflect this view. The mathematical definition of $T_{\mathrm{f}_t}$ is now provided as Eq.(67). $T_{\mathrm{f}_t}$ is interpreted as the highest temperature where it is likely to observe ice nucleation for a given

thermodynamic state.

**2 Referee 1**

**Reviewer:** *The manuscript has improved a lot and my concerns were mostly addressed in an adequate way. However, there are still some issue, which have to be clarified before this manuscript can be accepted for publication.*

**Response:** I thank the reviewer for the comments on the manuscript.

**Minor Points**

**Reviewer:** *1. The mathematical reformulation of the theory is still hard to follow; at least I had to rewrite the equations and to make the manipulations by myself before I could get through it. Maybe it would be worth to add some more intermediate steps in the derivation. For instance, the step from eq. (34) to eq. (35) took me a while. Maybe some additional derivations could be given in an appendix.*

**Response:** Several comments and further equations have been added throughout the paper to better guide the reader on the derivations. The steps from Eq (34) to (35) (now Eqs. (63) to (66)) are expanded and clarified.

**Reviewer:** *2. I have still concerns with the interpretation of equations (31) and (33). While in general (31) is fine with me, the interpretation of eq. (31) for the case $\zeta \to 1$ is not straightforward. It is clear that mix behave adequately for $\zeta \to 1$, $\Lambda_{mix} = 0$ for $\zeta \to 1$; however, one have to check, if the quotient $\frac{\Lambda_{mix}}{\zeta - 1}$ such that the equation is really converging to the mentioned values of $a_w$ , especially to $a_{w,\ eff} = 1$. I suggest that this should be checked in details, since the limit is used later.*

**Response:** The quotient $\frac{\Lambda_{mix}}{\zeta - 1}$ behaves well for $\zeta = 1$. This is readily shown as follows.

$$\frac{\Lambda_{mix}}{1 - \zeta} = \frac{1}{N(1 - \zeta)} \left[ \zeta \ln(\zeta) + (1 - \zeta) \ln(1 - \zeta) \right] + \frac{2}{N(1 - \zeta)} \frac{T_c}{T} \zeta(1 - \zeta), \quad (1)$$

simplyfing

$$\frac{\Lambda_{mix}}{1 - \zeta} = \frac{1}{N(1 - \zeta)} \left[ \zeta \ln(\zeta) + (1 - \zeta) \ln(1 - \zeta) \right] + \frac{2}{N} \frac{T_c}{T} \zeta, \quad (2)$$

Using $\ln(x) \to (x - 1)$ for $x \to 1$ the last expression can be rewritten as

$$\frac{\Lambda_{mix}}{1 - \zeta} = \frac{1}{N(1 - \zeta)} \left[ -\zeta(1 - \zeta) - (1 - \zeta)\zeta \right] + \frac{2}{N} \frac{T_c}{T} \zeta, \quad (3)$$

From where it can be seen that

$$\lim_{\zeta \to 1} \frac{\Lambda_{\text{mix}}}{1 - \zeta} = \frac{2}{N} \left( \frac{T_{\text{c}}}{T} - 1 \right), \qquad (4)$$

For $N = 6$, $T_{\text{c}} = 211.47$ K (see Table 1) and $T = 273.15$ K, $\lim_{\zeta \to 1} \exp\left( -\frac{\Lambda_{\text{mix}}}{1-\zeta} \right) = 0.93$. Recalling,

$$a_{\text{w, eff}} = \left( \frac{a_{\text{w}}}{a_{\text{w, eq}}^{\zeta}} \right)^{\frac{1}{1-\zeta}} \exp\left( -\frac{\Lambda_{\text{mix}}}{1-\zeta} \right). \qquad (5)$$

Since for $\zeta \to 1$, the second term of the right hand side of Eq. (5) is 0.93, then $a_{\text{w, eff}} = 1$ at some $a_{\text{w, eq}} < a_{\text{w}}$. However since for $\zeta \to 1$ the first term of the right hand side of Eq. (5) is almost singular at $a_{\text{w, eq}} = a_{\text{w}}$ then the value of $a_{\text{w, eq}}$ for which $a_{\text{w, eff}} = 1$ is just infinitesimally lower than $a_{\text{w}}$. Thus for all practical purposes $a_{\text{w, eff}} = 1$ at equilibrium.

The fact that $\lim_{\zeta \to 1} \exp\left( -\frac{\Lambda_{\text{mix}}}{1-\zeta} \right) < 1$ stems from the simple interaction model used to define $\Delta G_{\text{mix}}$ (i.e., the regular solution approximation). Figure 2 suggests that the minimun in $\Delta G_{\text{hom}}$ (hence the critical temperature) depends on $\zeta$. However such dependecy is not supported by the regular solution approximation which has a unique critical point at $\zeta = 0.5$. It is however not a claim of this work that the theory can be used for $\zeta = 1$ as such limit represents equilibrium and ice nucleation is not expected to occur.

The explanation above has been added to the Section.

**References**

[revised manuscript text omitted]
 = \frac{\Delta G_{\rm het}}{3\pi k_{\rm B}T\left(n^*\right)^2}\left(\frac{\Delta G_{\rm het}}{3\pi k_{\rm B}Tn_{\rm het}^2}\right)^{1/2}. \tag{59}$$

On the other hand using Eq. (46) into Eq. (2) we obtain,

$$Z_{\rm d} = \left[\frac{\Delta G_{\rm het}(n^*-2)^{1/3}}{3\pi k_{\rm B}T(n^*)^{7/3}}\frac{\Delta G_{\rm het}(n_{\rm het}-2)^{1/3}}{3\pi k_{\rm B}Tn_{\rm het}^{7/3}}\right]^{1/2} \tag{60}$$

where the subscript "d" indicates that energy dissipation is taken into account. For $n_{\rm het} > 3$ it is easily verifiable that $Z_{\rm d} \approx Z$. Indeed the discrepancy between $Z_{\rm d}$ and $Z$ is only 30% at  $n_{\rm het} = 3$ and much smaller for larger ice germs. However for  $n_{\rm het} = 2$, $Z_{\rm d} = 0$. This issue is rather fundamental and may represent the breaking of the assumption that each germ grows by addition of a single molecule at a time Hence Eq. (59) will be used keeping in mind that for very small ice germs it represents only an approximation.

With the above considerations it is now possible to substitute Eqs. (46), (47), (58) and (59) into Eq. (5) to obtain the heterogeneous ice nucleation rate,

$$J_{\rm het} = \frac{2ZD_\infty\Omega}{3v_w^2}\left(\frac{D_\infty}{D_0}\right)^{\frac{\zeta\sigma_{\rm E}}{1-\zeta\sigma_{\rm E}}}\left[1+\left(\frac{a_w}{a_{\rm w,eq}}\right)^{n_t(1-\zeta)}\right]^{-1}\exp\left(-\frac{[\Delta\mu_i(n_{\rm hom}^*+2)]_{a_{\rm w,eff}}}{2k_BT}\right)\exp\left(-\frac{n_{\rm het}\Delta\mu_i}{2k_BT}\right), \tag{61}$$

[revised manuscript text omitted]

\* From the data of Scala et al. (2000) the following fit was obtained:

$S_{\text{c},0} = k_{\text{B}} \upsilon_{\text{w}}/\upsilon_{\text{w,0}}(-7.7481 \times 10^{-5} T^2 + 5.5160 \times 10^{-2} T - 6.6716)$ $(\text{J}\,\text{K}^{-1})$ for $T$ between 180 K and 273 K.

[Figure]

**Figure 1.** Diagram representing a thermodynamic path including homogeneous ice nucleation with the same work as heterogeneous freezing.

1080

[Figure]

**Figure 2.** Work of heterogeneous ice nucleation. Color indicates different temperatures.

[Figure]

**Figure 3.** Different representations of immersion freezing. (a) An ice germ (dark blue) forming on an active site (AS) by random collision of water molecules (light blue). (b) Low density regions (dark blue) forming in the vicinity of active sites within a dense liquid phase (light blue).

[Figure]

**Figure 4.** Critical germ size (left panel) and work of heterogeneous ice nucleation (right panels) for different values of $\zeta$ (color). Black lines correspond to constant $J_{het} = 10^6$ m$^{-2}$ s$^{-1}$.

[Figure]

**Figure 5.** Preexponential factor. Colored lines indicates different values of $\zeta$. Black lines correspond to results calculated using CNT for different values of the contact angle, $\theta$.

[Figure]

**Figure 6.** Ice nucleation rate calculated using Eq. (61) for different values of $\zeta$ (color). Black lines were calculated using CNT for different values of the contact angle, $\theta$.

[Figure]

**Figure 7.** Thermodynamic freezing temperature as a function of water activity. Colored lines correspond to $T_{ft}(a_w = a_{w,het})$ for different values of $\zeta$. Also shown are the water activities at equilibrium and at the homogeneous freezing threshold, $a_{w,eq}$ and $a_{w,hom}$, respectively, and lines drawn applying constant water ativity shifts, $\Delta a_{w,het}$, of 0.05, 0.15 and 0.20.

[Figure]

**Figure 8.** Top panels: Heterogeneous ice nucleation rate calculated using a constant shift in $a_w$ (black, dotted, lines) for Leonardite (LEO $\Delta a_{w,\,het} = 0.2703$) and Pawokee Peat (PP, $\Delta a_{w,\,het} = 0.2466$) (top panels). Red, blue and green colors correspond to $a_w$ equal to 1.0, 0.931 and 0.872, respectively, for LEO and 1.0, 0.901 and 0.862 for PP. Shaded area corresponds to $\Delta a_{w,\,het} \pm 0.025$. Markers correspond to experimental measurements reported by Rigg et al. (2013); error bars represent an order of magnitude deviation from the reported value. Bottom panels: $J_{het}$ calculated for constant $\zeta = 0.949$ for LEO and $\zeta = 0.952$ for PP. The shaded area corresponds to $a_w \pm 0.01$ and $\zeta \pm 0.0015$.